# EFFICIENT TEST-TIME SCALING FOR SMALL VISION-LANGUAGE MODELS

**Mehmet Onurcan Kaya**[1,2]    **Desmond Elliott**[3,2]    **Dim P. Papadopoulos**[1,2]
[1] Technical University of Denmark    [2] Pioneer Center for AI    [3] University of Copenhagen
monka@dtu.dk,  de@di.ku.dk,  dimp@dtu.dk

⊕ **Project Page:** https://monurcan.github.io/efficient_test_time_scaling

## ABSTRACT

Small Vision-Language Models (VLMs) provide a computationally efficient alternative to larger models, at the cost of weaker generalization abilities and downstream task performance. These shortcomings could be addressed by test-time scaling techniques, but existing methods are typically computationally demanding, contradicting the resource-efficient design goals of small models. To address these limitations, we propose two novel and efficient test-time scaling strategies that leverage the model-internal features rather than external supervision: (i) Test-Time Augmentation (TTAug), which generates multiple augmented inputs and aggregates outputs at the token level without parameter updates, and (ii) Test-Time Adaptation (TTAdapt), which adapts model parameters during inference using consensus-based pseudolabels from TTAug. Through extensive experiments across nine benchmarks, we demonstrate consistent performance improvements while maintaining computational efficiency suitable for resource-constrained environments. The generality of our approach is demonstrated both within models at different scales and across different VLMs without additional tuning.

## 1 INTRODUCTION

Small Vision-Language Models (VLMs) offer computational efficiency and accessibility, yet their performance frequently degrades under domain shift due to inherent biases and limited generalization capabilities (Marafioti et al., 2025; Lu et al., 2025). While test-time scaling methods can, in principle, improve their performance, there are several critical limitations that undermine their practicality for small models in resource-constrained settings.

First, many test-time scaling methods rely on external verification models or computationally intensive reranking strategies, making them unsuitable for deployment on resource-constrained consumer GPUs (Zhang et al., 2024; Singh et al., 2025). This contradicts the resource-efficient design goals of small VLMs. Second, existing approaches that avoid external verifiers, such as sampling multiple candidate responses and aggregating them into a final prediction using the model's internal signals (Wang et al., 2023b; Adiwardana et al., 2020; Chen et al., 2024a), remain unsatisfactory because they typically operate only at the answer level, ignoring local signals for aggregation. Global measures like average confidence obscure token-level fluctuations that signal response quality, and averaging across entire sequences masks reasoning breakdowns at intermediate steps. Moreover, these methods require complete response generation before evaluation, preventing early termination and wasting computation. Finally, many existing methods are restricted to tasks with extractable final answers (e.g., multiple-choice or numerical reasoning), limiting their applicability to open-ended tasks such as visual question answering and captioning (Zhang et al., 2025a; Chen et al., 2024a).

In this paper, we leverage model-internal representations to overcome these limitations. Our goal is to improve the robustness and accuracy of small VLMs at inference time through efficient, lightweight, and practical test-time scaling strategies that require no external models or additional training data. We introduce two methods in a unified framework: Test-Time Augmentation and Test-Time Adaptation. Test-Time Augmentation generates multiple responses by applying input-level augmentations to both images and text. Crucially, it aggregates outputs at the token-level rather than the answer-level,

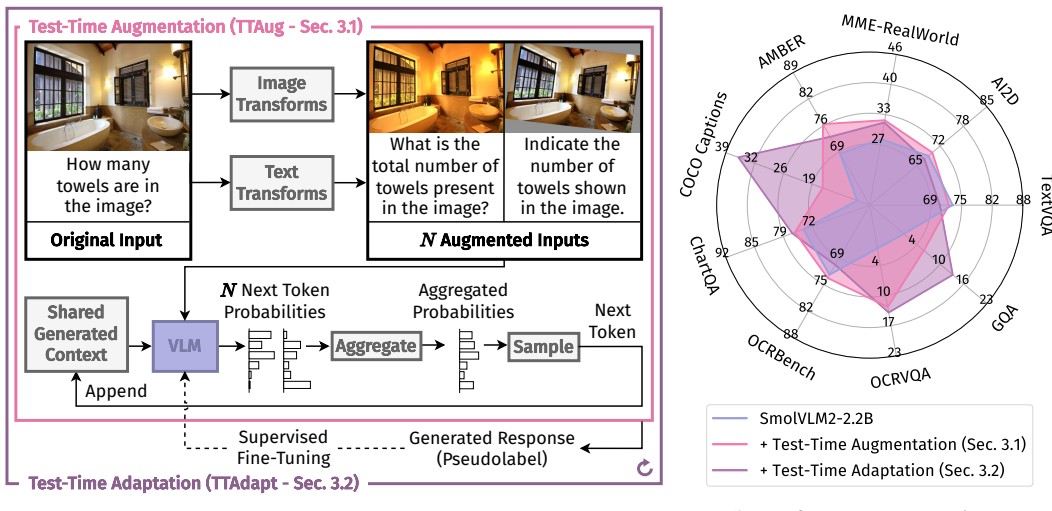

(a) Our efficient test-time scaling framework      (b) Performance comparison

Figure 1: Our framework consists of two main pipelines: (1) *Test-Time Augmentation*: Given an input image and text prompt, we apply various transformations to create multiple augmented versions. VLM processes each augmented input to produce next token probability distributions, which are then aggregated at the token level to generate the final response. (2) *Test-Time Adaptation*: We create pseudolabels through test-time augmentation and fine-tune the VLM parameters, then repeat the process. Both methods demonstrate effectiveness across nine diverse benchmarks as shown in (b).

which allows the model to quickly detect low-quality responses, and allows for more fine-grained exploitation of the model-internal signals. This method requires no parameter updates, making it both simple and efficient. Our second method, Test-Time Adaptation, extends this idea by adapting model parameters during inference. It leverages consensus signals from TTAug as pseudolabels, which guide lightweight fine-tuning on test samples without any labeled data. This enables the model to dynamically adjust to domain-specific characteristics while retaining computational efficiency.

Our approach consistently outperforms existing test-time scaling methods, such as self-consistency (Wang et al., 2023b), sample-and-rank (Adiwardana et al., 2020), self-selector (Chen et al., 2024a; Parmar et al., 2025), and self-synthesizer (Li et al., 2025d; Jiang et al., 2023; Wang et al., 2025a; Li et al., 2025b). Furthermore, these improvements do not come with a heavy computational cost, allowing our approach to be used in resource-constrained settings. Beyond performance gains, our study reveals two important general insights for test-time scaling: (1) generating multiple candidate answers through input augmentations with greedy decoding is more effective than the commonly-used temperature sampling strategy, and (2) token-level aggregation provides stronger signals than aggregating only at the final-answer level. These findings highlight practical principles for scaling VLMs efficiently at inference. Our experiments across nine diverse benchmarks and multiple VLM architectures confirm that these insights translate into consistent improvements and broad generalization, underscoring the effectiveness and generality of our framework.

Our contributions are threefold: (1) We present two efficient test-time scaling methods for small VLMs deployable on consumer GPUs. (2) We provide the first comprehensive analysis of Test-Time Augmentation for VLMs, investigating augmentation strategies, aggregation methods, and optimal aggregation layers. Despite being a simple and easily integrable technique, its application to multimodal settings remains surprisingly underexplored. (3) We introduce the first Test-Time Adaptation method for multimodal language models, whereas prior work on VLM test-time adaptation has focused primarily on CLIP-based models (Liang et al., 2025; Dong et al., 2025; Ji et al., 2025).

## 2 RELATED WORK

**Test-time scaling** is a paradigm in which current large language models increasingly achieve superior performance by allocating substantial computational resources during inference (Zhang et al., 2025a; Ji et al., 2025). A popular test-time scaling strategy is parallel sampling, which generates multiple

outputs simultaneously and aggregates them. However, existing parallel sampling methods face several critical limitations that make them impractical for resource-constrained deployments. Most approaches rely on external verifier models or compute-heavy strategies, making them incompatible with the small model paradigm (Zhang et al., 2024; Singh et al., 2025). We address these limitations by proposing two lightweight but effective methods via test-time augmentation.

**Test-time augmentation (TTAug)** improves model robustness and generalization by averaging predictions across augmented views (Shorten & Khoshgoftaar, 2019). Recent work extends TTAug through learnable policies (Lyzhov et al., 2020; Kim et al., 2020; Shanmugam et al., 2021) by optimizing augmentation selection and weighting. However, these active methods typically require labeled datasets to learn optimal policies, limiting their practical applicability. Prior TTAug research for (multimodal) language models (Mashrur et al., 2022; Kamoda et al., 2023) mainly addresses hallucination detection and robustness, not accuracy improvement, and does not treat TTAug as a systematic test-time scaling method. Our work closes this gap by extending both non-learnable and learnable TTAug strategies to Vision-Language Models (VLMs), systematically evaluating how augmentation design, aggregation, and scaling affect performance across tasks, and leveraging self-supervised objectives from test-time adaptation literature to avoid reliance on labeled data.

**Test-time adaptation (TTAdapt)** is an emerging paradigm for adapting pretrained models to new data batches during inference by updating model weights or inputs to maximize prediction accuracy without ground-truth labels (Xiao & Snoek, 2024). The choice of optimization target and objective is crucial for adaptation effectiveness. In multimodal learning, most prior TTAdapt work focuses on CLIP-based VLMs (Liang et al., 2025; Dong et al., 2025; Ji et al., 2025), with entropy minimization as the optimization strategy (Shu et al., 2022) and widespread use of self-training with pseudolabels. In language models, TTAdapt is less explored (Dong et al., 2025; Ji et al., 2025). Hübotter et al. (2025) require training datasets and is not source-free, while Huang et al. (2025) extend an existing test-time scaling method called self-consistency (Wang et al., 2023b) for better confidence calibration but suffers from the same limitations of applicability and generalization. Akyürek et al. (2025) explore test-time training with methods similar to ours (i.e., aggregated predictions via hierarchical voting and per-instance adaptation); however, their method is specifically designed for the ARC benchmark and lacks broader applicability. Our universal and source-free TTAdapt method overcomes these limitations by leveraging consensus-based pseudolabeling from our TTAug method.

## 3 METHODS

We propose a comprehensive framework for test-time scaling of small Vision-Language Models (VLMs) through two complementary approaches: test-time augmentation (TTAug) and test-time adaptation (TTAdapt). Fig. 1a illustrates our framework, which addresses the fundamental challenge of improving model performance and robustness without requiring additional training data or substantial computational overhead.

### 3.1 TEST-TIME AUGMENTATION (TTAUG)

Our approach leverages input diversity to improve model robustness through systematic aggregation of predictions from semantically equivalent inputs. Given an input consisting of an image $\mathbf{I}$ and text prompt $\mathbf{t}$, we generate a set of $N$ augmented versions $\{(\mathbf{I}_i, \mathbf{t}_i)\}_{i=1}^{N}$ through semantic-preserving transformations (Sec. 4.4). Each transformation preserves the semantic content essential for multimodal understanding while introducing controlled textual and visual diversity (Sec. 4.5 and 4.6).

Our token generation process follows an autoregressive approach where aggregation occurs at each step during generation. Starting with an empty sequence $\mathbf{y} = \{\}$, we iteratively generate tokens. At generation step $j$, for each augmented input $(\mathbf{I}_i, \mathbf{t}_i)$, the VLM computes the probability distribution over the vocabulary $\mathcal{V}$ conditioned on the current shared context:

$$p_{i,j}(v) = p(v|\mathbf{I}_i, \mathbf{t}_i, \mathbf{y}_{<j}) = \text{softmax}(\mathbf{f}_\theta(\mathbf{I}_i, \mathbf{t}_i, \mathbf{y}_{<j})) \tag{1}$$

where $\mathbf{f}_\theta$ represents the VLM with parameters $\theta$, and $\mathbf{y}_{<j} = \{y_1, \ldots, y_{j-1}\}$ denotes the shared sequence of previously generated tokens. We then aggregate the probability distributions across all augmented inputs through token-level averaging:

$$\bar{p}_j(v) = \frac{1}{N} \sum p_{i,j}(v) \tag{2}$$

The next token is selected greedily from this aggregated distribution:

$$y_j = \arg\max_{v \in \mathcal{V}} \bar{p}_j(v) \tag{3}$$

This selected token $y_j$ is then appended to the shared context $\mathbf{y} = \mathbf{y} \cup \{y_j\}$, and the process repeats for the next step. This autoregressive aggregation ensures that each token decision leverages the collective confidence from all augmented views while maintaining a single coherent output sequence.

This token-level aggregation strategy enables the model to leverage local confidence signals from multiple augmented views at each generation step, combining the strengths of different input representations (Sec. 4.3). Moreover, semantic-preserving input perturbations with greedy decoding yield superior diversity than temperature sampling used in prior test-time scaling methods (Sec. 4.2).

## 3.2 Test-Time Adaptation (TTAdapt)

We also introduce a learnable variant that adapts model parameters during inference through iterative pseudolabel generation and fine-tuning. Our TTAdapt method operates without requiring labeled data by leveraging the consensus from TTAug as a supervision signal.

The TTAdapt process optimizes the entire VLM parameter set $\theta$ through consensus-driven supervision in an iterative three-stage loop: (1) generate high-confidence pseudolabels using the current model state with TTAug consensus, (2) fine-tune model parameters using these pseudolabels as supervision through efficient training with gradient checkpointing or parameter-efficient methods, and (3) reset to initial weights before processing each new question to prevent catastrophic forgetting. This iterative process allows the model to progressively adapt to the test distribution while maintaining stability through consensus-based pseudolabeling. See Appendix I.7 for implementation details.

Formally, given an input image-text pair $(\mathbf{I}, \mathbf{t})$, initial model parameters $\theta_0$, and number of adaptation iterations $K$, the TTAdapt process proceeds as outlined in Algorithm 1.

---

**Algorithm 1** Test-Time Adaptation (TTAdapt)

---

**Require:** Input image $\mathbf{I}$, text prompt $\mathbf{t}$, initial parameters $\theta_0$, iterations $K$
1: $\theta \leftarrow \theta_0$                      ▷ Initialize with original weights
2: **for** $k = 1$ to $K$ **do**
3:      $\mathbf{y}^{(k)} \leftarrow \text{TTAug}(\mathbf{I}, \mathbf{t}; \theta)$           ▷ Generate pseudolabel via TTAug
4:      $\theta \leftarrow \arg\min_\theta \left( -\log p(\mathbf{y}^{(k)}|\mathbf{I}, \mathbf{t}; \theta) \right)$          ▷ Update parameters
5: **end for**
6: $\mathbf{y}^* \leftarrow \text{TTAug}(\mathbf{I}, \mathbf{t}; \theta)$           ▷ Generate final adapted response
7: $\theta \leftarrow \theta_0$           ▷ Reset to initial weights for the next question
8: **return** $\mathbf{y}^*$

---

The TTAug method generates multiple predictions for each test input and aggregates them using token-level averaging to create high-confidence pseudolabels. These pseudolabels represent the collective wisdom of the augmented predictions and serve as training targets for model adaptation. By iteratively refining predictions through TTAug consensus and parameter updates, we enable the model to adapt to test-time distribution shifts while preserving its core capabilities. Through this iterative process, we adapt the model parameters to achieve locally-optimal performance for the specific question type encountered during inference (Sec. 4.7).

Our unified framework provides flexibility for different deployment scenarios: TTAug offers immediate improvements without parameter updates, while TTAdapt enables more substantial gains when brief optimization is feasible. We systematically evaluate both approaches across diverse benchmarks and models to understand their effectiveness and computational trade-offs (Sec. 4.8).

## 4 Experiments

We conduct comprehensive experiments to validate the test-time scaling framework presented in the previous section. Each major design decision is explored here, e.g. how can we generate high-quality diverse answers, or should we perform aggregation at the level of the final answer or at

the token-level, using the SmolVLM2-2.2B (Marafioti et al., 2025) model as the baseline. Our experiments encompass 9 benchmarks covering various task types: visual question answering (VQA) including ChartQA (Masry et al., 2022), OCRBench (Liu et al., 2024), OCRVQA (Mishra et al., 2019), GQA (Hudson & Manning, 2019), and TextVQA (Singh et al., 2019); multiple-choice questions (MCQ) with AI2D (Kembhavi et al., 2016) and MME-RealWorld (Zhang et al., 2025b); yes/no questions using AMBER (Wang et al., 2023a); and image captioning with COCO Captions (Lin et al., 2014). We utilize the evaluation protocols provided by VLMEvalKit (Duan et al., 2024) to ensure standardized and reproducible results. For computational efficiency and fair comparison across all methods, we sample 1000 samples from each benchmark using uniform intervals to maintain representative coverage of the original data distribution while enabling extensive ablation studies. The evaluation metric is accuracy for most benchmarks, with ROUGE-L used specifically for COCO Captions. For a comprehensive description of the evaluation metrics, refer to Appendix J. Standard errors for all tables in this section are provided in Appendix K.

### 4.1 COMPARISON WITH OTHER TEST-TIME SCALING METHODS

We compare our TTAug approach against four representative test-time scaling methods from the existing literature that can potentially operate without external model dependencies.

① **Self-Consistency** aggregates candidate answers via majority voting across multiple sampled outputs (Wang et al., 2023b). While effective for tasks where final answers can be parsed, it struggles in creative or open-ended settings where the final answer is not easy to parse.

② **Self-Selector** uses the VLM itself as a verifier to select one response among the candidates (Chen et al., 2024a; Parmar et al., 2025). This approach extends applicability beyond tasks suited to majority voting. See Appendix I.1 for implementation details.

③ **Sample-and-Rank.** Self-Consistency ignores the model's internal signals for selection; majority voting treats all reasoning traces equally, ignoring quality variations (Wang et al., 2025b). Sample-and-Rank (Adiwardana et al., 2020), leverages next-token distribution statistics to assess response quality by selecting the response with the highest log probability, $\arg\max \log p(\mathbf{y})$.

④ **Self-Synthesizer.** The selection of only one answer, as in previous strategies, ignores information from other responses. To combine potentially correct parts from different responses, we use the tested VLM to aggregate responses into one coherent final answer (Li et al., 2025d; Jiang et al., 2023; Wang et al., 2025a; Li et al., 2025b). See Appendix I.2 for implementation details.

⑤ **TTAug (Ours).** Our Token-level aggregation with simple averaging approach aggregates the predictions at each step using a token-level aggregation of the final logits, as defined in Eq. 2.

In this experiment, for our TTAug method, we augment the inputs $N = 8$ times. For all other compared methods, we similarly generate $8$ candidate answers before aggregation.

Tab. 1 demonstrates the superiority of our TTAug method over the existing methods. Interestingly, most existing methods fail to consistently outperform the baseline model across all benchmarks. In contrast, our TTAug method achieves a $+4.1\%$ absolute improvement over the baseline model. Also, our method is more efficient in terms of both runtime and number of output tokens generated. This consistent advantage can be attributed to two key factors. First, by leveraging input perturbations with greedy decoding for diversity inducement, our method generates higher-quality candidate responses than temperature sampling, which is what all other methods rely on. Second, token-level aggregation preserves local confidence signals during generation, enabling more nuanced error correction compared to global

Table 1: Comparison of our TTAug method against existing test-time scaling methods. Our method outperforms all others across accuracy and efficiency metrics.

| | Baseline | Others | | | | Ours |
|---|---|---|---|---|---|---|
| | | ① | ② | ③ | ④ | ⑤ |
| ChartQA | 74.2 | 74.4 | 73.4 | 72.5 | 71.7 | 75.6 |
| OCRBench | 72.9 | 72.6 | 71.9 | 70.2 | 71.9 | 73.4 |
| OCRVQA | 0.0 | 0.0 | 0.0 | 0.0 | 0.2 | 11.8 |
| GQA | 0.0 | 0.0 | 0.0 | 0.0 | 0.0 | 5.8 |
| TextVQA | 73.2 | 72.6 | 71.6 | 69.5 | 72.0 | 72.8 |
| AI2D | 68.5 | 3.1 | 69.2 | 69.1 | 67.4 | 68.8 |
| MME-RW | 27.8 | 26.2 | 26.4 | 27.6 | 27.6 | 31.1 |
| AMBER | 68.7 | 70.4 | 64.5 | 53.5 | 67.8 | 75.4 |
| COCO | 9.1 | 8.2 | 8.4 | 6.2 | 16.7 | 15.9 |
| **Mean** | 43.8 | 36.4 | 42.8 | 41.0 | 43.9 | **47.9** |
| Runtime (s) | 1.43 | 3.73 | 4.18 | 3.74 | 4.46 | 2.99 |
| # Tokens | 8.7 | 74.5 | 77.4 | 74.5 | 82.3 | 70.3 |

The accuracy rows are grouped under "Accuracy ↑" and the last two rows under "Eff. →".

answer-level methods that discard such information. In the following two sections, we separately validate these two critical components of our method.

**Takeaway:** Our TTAug method consistently outperforms existing test-time scaling methods while being significantly more efficient.

## 4.2 Diversity-inducement methods

Generating diverse, high-quality candidate answers is critical for test-time scaling. We compare two approaches for inducing diversity: **Temperature Sampling**, and **Input Perturbations** combined with greedy decoding. Temperature Sampling introduces randomness into the process by sampling from a softened probability distribution, while Input Perturbations applies classic semantic-preserving augmentations to inputs (Sec. 4.5 and 4.6), and then decodes greedily.

Tab. 2 shows that Input Perturbations with Greedy Decoding outperform Temperature Sampling for generating high-quality candidate responses under both the ① Self-Consistency and ② Self-Selector strategies. This approach achieves the largest gains on OCRVQA and GQA, where temperature sampling fails. The theoretical analysis in Appendix A shows that greedy decoding with input perturbations maintains a higher correlation and better alignment with the model training objective, making it more effective for test-time scaling.

**Takeaway:** Input Perturbations with greedy decoding ultimately performs better than the Baseline or Temperature Sampling. This fundamental insight forms the basis of our method throughout the remainder of the paper.

Table 2: Comparison of diversity-inducement methods compared to the Baseline. Input Perturbation outperforms Temperature Sampling.

| | Baseline | Temperature Sampling | | Input Perturbation | |
|---|---|---|---|---|---|
| | | ① | ② | ① | ② |
| ChartQA | 74.2 | 74.4 | 73.4 | 74.8 | 70.9 |
| OCRBench | 72.9 | 72.6 | 71.9 | 72.7 | 73.1 |
| OCRVQA | 0.0 | 0.0 | 0.0 | 12.0 | 4.5 |
| GQA | 0.0 | 0.0 | 0.0 | 7.6 | 3.7 |
| TextVQA | 73.2 | 72.6 | 71.6 | 72.3 | 72.9 |
| AI2D | 68.5 | 3.1 | 69.2 | 3.6 | 66.6 |
| MME-RW | 27.8 | 26.2 | 26.4 | 30.8 | 29.6 |
| AMBER | 68.7 | 70.4 | 64.5 | 72.7 | 67.0 |
| COCO | 9.1 | 8.2 | 8.4 | 21.2 | 13.0 |
| **Mean** | 43.8 | 36.4 | 42.8 | 40.9 | **44.6** |

## 4.3 Aggregation levels

We now compare different aggregation levels for test-time scaling: **Answer-Level** versus **Token-Level** aggregation. Existing test-time scaling methods predominantly employ answer-level aggregation with temperature sampling for diversity inducement (Zhang et al., 2025a). However, given that input perturbations with greedy decoding provide superior diversity inducement, we evaluate answer-level versus token-level aggregation using this improved diversity-inducement method for comparison.

Nevertheless, all of the answer-level aggregation methods have critical limitations. First, global measures like confidence obscure confidence fluctuations at local reasoning steps, which can provide valuable signals for estimating response quality. Averaging across entire sequences masks critical reasoning breakdowns that occur at intermediate steps. Additionally, global measures require generating complete responses before calculation, preventing early stopping of low-quality generations and resulting in computational inefficiency. They generate a constant number of responses per question rather than adaptively distributing computational budget based on response agreement. Moreover, small VLMs also often lack sufficient synthesis capabilities for reliable response combination.

Tab. 3 demonstrates the effectiveness of Token-level aggregation compared to the Answer-level methods.

Table 3: Comparison of Answer-level versus Token-level aggregation methods. Token-level aggregation outperforms all other approaches.

| | Baseline | Answer-Level | | | | Token |
|---|---|---|---|---|---|---|
| | | ① | ② | ③ | ④ | ⑤ |
| ChartQA | 74.2 | 74.8 | 70.9 | 61.1 | 72.8 | 75.6 |
| OCRBench | 72.9 | 72.7 | 73.1 | 60.9 | 71.1 | 73.4 |
| OCRVQA | 0.0 | 12.0 | 4.5 | 0.2 | 3.3 | 11.8 |
| GQA | 0.0 | 7.6 | 3.7 | 0.0 | 0.0 | 5.8 |
| TextVQA | 73.2 | 72.3 | 72.9 | 61.6 | 71.6 | 72.8 |
| AI2D | 68.5 | 3.6 | 66.6 | 69.9 | 68.0 | 68.8 |
| MME-RW | 27.8 | 30.8 | 29.6 | 29.0 | 29.2 | 31.1 |
| AMBER | 68.7 | 72.7 | 67.0 | 58.9 | 75.8 | 75.4 |
| COCO | 9.1 | 21.2 | 13.0 | 8.6 | 29.5 | 15.9 |
| **Mean** | 43.8 | 40.9 | 44.6 | 38.9 | 46.8 | **47.9** |

This consistent advantage validates our hypothesis that token-level aggregation preserves valuable local confidence information that global answer-level methods discard. Particularly notable are the

improvements on OCRVQA, GQA, and COCO, where the baseline model struggles, indicating that token-level aggregation effectively leverages augmentation diversity to recover from initial prediction failures. The method's ability to outperform answer-level approaches, such as Self-Synthesizer, despite their access to the full model's reasoning capabilities, underscores the fundamental advantage of preserving local confidence signals during generation rather than attempting post-hoc response combination. Appendix B provides a mathematical analysis of this phenomenon. Appendix C presents experiments using different Token-level aggregation methods, including entropy-weighted averaging, majority voting, and most confident token. Finally, Appendix D shows that aggregation of earlier layer outputs can produce better results for some tasks.

**Takeaway:** Token-level aggregation consistently outperforms Answer-level aggregation. This validates our test-time augmentation method as a more practical alternative to existing test-time scaling approaches that rely on Answer-level approaches based on Selection or Synthesis. We use Token-level aggregation with simple averaging at the final logits for all subsequent experiments.

## 4.4 NUMBER OF AUGMENTATIONS

We study how performance scales with the number of augmented inputs to understand the optimal balance between computational cost and accuracy. Augmentation counts range from 1 (baseline) to 64 with simple averaging aggregation. This analysis clarifies diminishing returns in test-time scaling and provides practical guidance for deployment scenarios with varying computational budgets.

Fig. 2 reveals diverse scaling behaviors across benchmarks, reflecting task-specific characteristics. Benchmarks showing monotonic improvement with saturation (OCRVQA, AMBER, MME-RealWorld) follow established test-time scaling patterns (Snell et al., 2025; Brown et al., 2025; Wu et al., 2024), with performance increasing steadily before plateau-

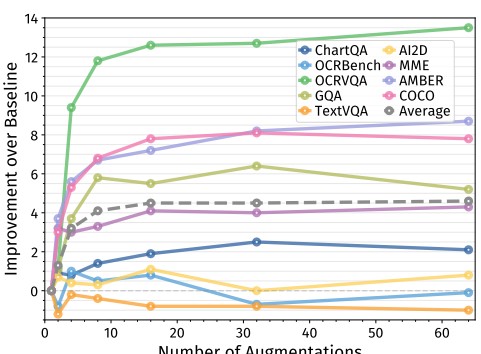

Figure 2: Performance scaling as a function of the number of augmentations. Performance gains generally plateau after 16 augmentations.

ing. In contrast, several benchmarks exhibit non-monotonic curves (ChartQA, COCO, GQA) where performance peaks at intermediate augmentation counts before declining due to the consistency-diversity tradeoff (Geiping et al., 2023). This decline probably occurs because excessive augmentation introduces outlier predictions and semantic drift that degrade aggregated signal quality, as simple token-level averaging assumes equal validity across augmented predictions. Mixed behaviors (OCR-Bench, TextVQA, AI2D) show irregular patterns with task-specific characteristics.

**Takeaway:** The average performance curve (dashed line) indicates peak performance at 16 augmentations, which we adopt for subsequent experiments. This translates to a peak GPU memory usage of 8.75 GB ($1.9\times$ increase from 4.60 GB baseline) and an inference time of 4.77 s per query ($3.33\times$ increase from 1.43 s baseline), when using parallel batch inference on an NVIDIA A100 GPU. For detailed computational overhead analysis across different augmentation counts, refer to Appendix E.

## 4.5 TEXT AUGMENTATION METHODS

We now compare different text augmentation strategies to understand the trade-offs between quality, practicality, and computational overhead in our resource-constrained setting.

⓪ **Image-Only** uses classical image augmentations (Sec. 4.6) without text augmentation, serving as a control. ①–④ apply the same image augmentations along with their respective text strategies.

① **AugGPT** uses ChatGPT (Achiam et al., 2023) to generate high-quality paraphrases, to evaluate the ability of high-capacity finetuned paraphraser models distilled for our scenario (Dai et al., 2025). This provides high-quality paraphrasing augmentation using state-of-the-art external models, but it is not practical as it requires external models in resource-constrained deployment scenarios.

② **Self-Paraphrasing** uses the LLM of the VLM to paraphrase the input prompt. Since small VLMs cannot reliably do this in one shot, we split the prompt into sentences and paraphrase each

with structured generation to obtain a fixed number of variants. The final paraphrased prompt is the concatenation of these outputs. This approach maintains consistency with the target model's internal linguistic patterns while remaining self-contained. See Appendix I.3 for implementation details.

③ **Classical Augmentation** uses simple and fast semantic-preserving augmentations sequentially and randomly with minimal cost (Ma, 2019; Aepli & Sennrich, 2022). Keyboard errors simulate common typing mistakes by replacing characters with nearby keys leveraging one-key distance to generate realistic character substitutions. Word splitting introduces spacing variations within compound words. Word deletion removes individual words. Sentence reordering swaps adjacent sentences.

**Consistency Enforcement** is applied by appending the original prompt after each augmented version, structured as "In other words," followed by the original prompt, mirroring the alpha blending technique in AugMix (Hendrycks et al., 2020). We report ablation study results without consistency enforcement using the classical augmentation method in the ④ column of Tab. 4; all other columns (①, ②, ③) employ consistency enforcement technique.

Tab. 4 shows that self-paraphrasing achieves superior performance by leveraging model-aligned augmentations, as the model's own weights influence how prompts are generated, resulting in augmentations that exhibit superior alignment with the model's internal representations. This approach creates linguistic patterns within the training manifold, leading to better-calibrated confidence estimates during token-level aggregation. Consistency enforcement proves critical for semantic coherence, with large drops observed in the ablation study, though notable exceptions occur in GQA and MME-RealWorld where diversity outweighs consistency. Classical augmentations remain competitive with minimal computational overhead, making them the most practical choice for resource-constrained deployment. Their similar performance to self-paraphrasing suggests simple perturbations provide sufficient diversity for our purposes.

Table 4: Comparison of text augmentation strategies. Self-Paraphrasing ② and Classical Augmentations ③ consistently perform best.

| | Baseline | ⓪ | ① | ② | ③ | ④ |
|---|---|---|---|---|---|---|
| ChartQA | 74.2 | 74.7 | 76.9 | 76.6 | 76.1 | 71.4 |
| OCRBench | 72.9 | 73.3 | 73.5 | 72.8 | 73.7 | 70.6 |
| OCRVQA | 0.0 | 0.0 | 2.6 | 0.0 | 12.6 | 0.0 |
| GQA | 0.0 | 0.0 | 0.0 | 0.0 | 5.5 | 31.2 |
| TextVQA | 73.2 | 74.2 | 73.5 | 74.0 | 72.4 | 63.9 |
| AI2D | 68.5 | 69.8 | 69.9 | 68.4 | 69.6 | 63.9 |
| MME-RW | 27.8 | 26.6 | 30.0 | 25.9 | 31.9 | 32.1 |
| AMBER | 68.7 | 64.7 | 68.8 | 72.9 | 75.9 | 60.0 |
| COCO | 9.1 | 8.4 | 20.6 | 46.2 | 16.9 | 13.2 |
| **Mean** | 43.8 | 43.5 | 46.2 | **48.5** | 48.3 | 45.1 |

**Takeaway:** Self-paraphrasing ≻ Classical ≻ AugGPT. Consistency enforcement is critical for reliable performance. For the remaining experiments, we use classical augmentation with consistency enforcement to balance accuracy and efficiency.

## 4.6 IMAGE AUGMENTATION METHODS

We evaluate three different image augmentation strategies to understand their effectiveness for multimodal test-time scaling: classical transformations, established methods, and generative approaches.

⓪ **Text-Only** uses classical text augmentation (Sec. 4.5) without image augmentation, serving as a control. ①–③ apply the same text augmentation along with their respective image strategies.

① **Classical Augmentations** apply traditional computer vision transformations including brightness/contrast adjustments, rotation, blurring, noise injection, and geometric distortions, shown useful in other vision-language tasks (Vendrov et al., 2016). We test three augmentation intensity levels: Ⓛ **Low** (conservative), Ⓜ **Medium** (moderate), and Ⓗ **High** (aggressive) to explore the diversity-consistency trade-off. See Appendix I.4 for detailed implementation specifications.

② **AugMix** (Hendrycks et al., 2020) employs a mixing strategy that combines multiple augmentation chains with convex combinations, originally designed for robustness in image classification tasks.

③ **Generative Augmentations** use FLUX.1-dev (Labs et al., 2025) to create semantically similar but visually distinct image variants. However, this approach excludes text-containing images to

prevent OCR corruption. Also, it requires external diffusion models, making it impractical for resource-constrained deployments. See Appendix I.5 for detailed implementation specifications.

Tab. 5 reveals several key insights about image augmentation strategies. Classical augmentations with high and low strengths marginally outperform medium strength augmentations. This non-monotonic relationship reflects the fundamental diversity-consistency trade-off: low strength preserves semantic coherence but provides limited diversity; high strength introduces beneficial variance without excessive semantic drift (Geiping et al., 2023); while medium strength falls into a suboptimal region where augmentations disrupt model confidence without compensating diversity benefits.

AugMix performs competitively but falls short of classical methods, suggesting that the principled mixing strategy designed for unimodal classification may not align with the token-level aggregation in VLMs. Generative augmentations underperform despite their semantic richness, primarily because text-containing images must be excluded, reducing the effective augmentation coverage.

For a modality-wise decomposition of TTAug performance gains, see Appendix F; for representative samples of augmented inputs with classical methods and corresponding outputs, see Appendix L.

**Takeaway:** Classical high/low strength augmentations outperform AugMix and generative approaches; with medium strength falling into a suboptimal diversity-consistency trade-off. Thus, we use high-strength classical augmentations for all subsequent experiments.

Table 5: Comparison across different image augmentation strategies. Classical Augmentations Ⓛ, Ⓗ perform the best.

| | Baseline | ⓪ | Ⓛ | Ⓜ | Ⓗ | ② | ③ |
|---|---|---|---|---|---|---|---|
| ChartQA | 74.2 | 75.8 | 77.0 | 76.4 | 76.1 | 74.1 | 75.7 |
| OCRBench | 72.9 | 73.1 | 73.7 | 73.3 | 73.7 | 72.4 | 65.3 |
| OCRVQA | 0.0 | 13.5 | 12.1 | 10.6 | 12.6 | 12.9 | 12.0 |
| GQA | 0.0 | 2.0 | 4.1 | 3.7 | 5.5 | 3.1 | 2.5 |
| TextVQA | 73.2 | 73.0 | 72.6 | 73.3 | 72.4 | 72.4 | 71.6 |
| AI2D | 68.5 | 68.1 | 69.1 | 69.0 | 69.6 | 68.9 | 67.0 |
| MME-RW | 27.8 | 31.6 | 31.8 | 32.5 | 31.9 | 32.1 | 31.1 |
| AMBER | 68.7 | 77.3 | 77.0 | 75.9 | 75.9 | 77.3 | 76.2 |
| COCO | 9.1 | 19.0 | 17.8 | 17.1 | 16.9 | 17.8 | 18.0 |
| **Mean** | 43.8 | 48.2 | 48.3 | 48.0 | **48.3** | 47.9 | 46.6 |

## 4.7 Test-time adaptation methods

While TTAug provides improvements, test-time adaptation (TTAdapt) extends this framework by optimizing learnable components during inference. Unlike conventional test-time scaling that generates and selects among multiple candidate responses, our approach directly optimizes model behavior using self- or semi-supervised objectives. We investigate two different adaptation strategies targeting distinct components of the aggregation pipeline, each with unique optimization objectives.

① **Aggregation Weights Optimization** learns adaptive token-wise weights $w_{i,j}$ to replace the uniform averaging scheme in Eq. 2. At each generation step $j$, we initialize learnable parameters as $\mathbf{w}_j \in \mathbb{R}^N$ and optimize them through gradient descent to minimize the marginal entropy $H(\bar{p}_j) = -\sum_{v \in \mathcal{V}} \bar{p}_j(v) \log \bar{p}_j(v)$ of the weighted aggregated distribution by performing multiple micro-steps per token to achieve convergence. This approach requires minimal computational overhead with a compact computational graph, making it suitable for real-time deployment. Marginal entropy minimization represents the dominant optimization paradigm in test-time adaptation for CLIP-based models (Shu et al., 2022; Liang et al., 2025). We include this method as an ablation study and as a computationally efficient alternative to our main adaptation approach. See Appendix I.6 for details.

② **Model Parameter Adaptation** implements the iterative pseudolabel generation and fine-tuning framework detailed in Sec. 3.2.

Tab. 6 shows clear performance differences among adaptation methods with distinct efficiency-performance trade-offs. Aggregation weights optimization performs on par with TTAug, mainly improving benchmarks that require precise confidence calibration (e.g., AMBER, TextVQA), where adaptive weighting highlights high-quality predictions. This supports findings that TTAdapt via marginal entropy minimization is not more effective than TTAug for CLIP-based VLMs (Farina et al., 2024). Its average performance matches TTAug, but it notably fixes simple averaging's underperformance on TextVQA, outperforming the baseline model on all benchmarks. Model parameter adaptation delivers the strongest overall gains, particularly excelling on COCO. Given its superior performance, we refer to model parameter adaptation as TTAdapt throughout this paper.

However, performance occasionally degrades on specialized benchmarks with strong baseline capabilities, indicating that aggressive parameter adaptation can disrupt carefully calibrated domain-specific knowledge. This pattern suggests that adaptation intensity should be task-dependent, conservative for well-calibrated domains where the base model already performs well, and more aggressive for challenging distributions where consensus-based supervision provides reliable guidance.

**Takeaway:** Model adaptation achieves superior gains through consensus-based learning. Aggregation weight optimization provides an efficient middle ground with minimal computational overhead.

Table 6: Performance comparison of test-time adaptation strategies. Model parameter adaptation ② yields the best performance.

|  | Baseline | TTAug | ① | ② |
|---|---|---|---|---|
| ChartQA | 74.2 | 76.1 | 76.1 | 76.7 |
| OCRBench | 72.9 | 73.7 | 73.0 | 70.5 |
| OCRVQA | 0.0 | 12.6 | 11.9 | 13.8 |
| GQA | 0.0 | 5.5 | 5.2 | 13.5 |
| TextVQA | 73.2 | 72.4 | 74.2 | 70.5 |
| AI2D | 68.5 | 69.6 | 69.7 | 67.4 |
| MME-RW | 27.8 | 31.9 | 30.9 | 31.4 |
| AMBER | 68.7 | 75.9 | 76.9 | 72.8 |
| COCO | 9.1 | 16.9 | 16.4 | 35.9 |
| **Mean** | 43.8 | 48.3 | 48.3 | **50.3** |

## 4.8 CROSS-MODEL GENERALIZATION

Finally, we test our method's generalization to other VLMs by applying the SmolVLM2-2.2B configuration (greedy decoding, 16 classical augmentations, token-level averaging) to diverse architectures and parameter scales. See Appendix H for more details.

Fig. 3 shows performance gains across model families and parameter scales. The best performance gains are found for SmolVLM2-2.2B, but we find consistent improvements across different architectures and scales. TTAug prevents error propagation through token-level aggregation, which provides fundamental advantages regardless of model specifics. Key findings include: (1) Although optimal hyperparameters vary across models due to differences in training data, architecture, and training augmentations, our framework generalizes well and provides improvements. However, no universal set of hyperparameters exists that optimally serves all models. Hyperparameter transferability is inherently limited due to model-specific characteristics, including training data biases, architectural differences, and training

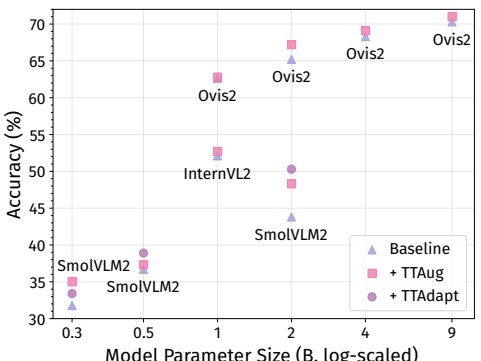

Figure 3: Improvements across different models, demonstrating cross-model generalization.

augmentation strategies. (2) TTAug effectiveness does not simply correlate with model size but rather with model family and architectural similarity. This challenges our initial expectation that TTAug would primarily benefit smaller models by mitigating biases (with larger models being more robust). Instead, improvements appear more dependent on similarity to our hyperparameter optimization target. Hyperparameter transferability is stronger within model families sharing similar architectures and training procedures, as optimal hyperparameters depend on dataset biases, training augmentations, and architectural inductive biases. Also, models with similar parameter counts exhibit better hyperparameter transferability, suggesting that model capacity also influences optimal augmentation strategies. Even with suboptimal hyperparameters, our methods yield robust improvements, though dedicated tuning is recommended for best results. See Appendix G for more detailed results.

**Takeaway:** Despite hyperparameters being optimized for SmolVLM2-2.2B, our methods provide consistent improvements across diverse models, though transferability varies by family and size.

## 5 CONCLUSION

We propose two efficient test-time scaling methods, Test-Time Augmentation and Test-Time Adaptation. Our comprehensive experiments demonstrate that both methods consistently improve performance by outperforming existing test-time scaling approaches with minimal overhead, making them suitable for resource-constrained environments. Our work provides a systematic way to tune hyperparameters for a given model, though optimal strategies remain task- and model-dependent.

## ACKNOWLEDGMENTS

Dim P. Papadopoulos was supported by the DFF Sapere Aude Starting Grant "ACHILLES". Desmond Elliott was supported by the European Union's Horizon 2020 research and innovation program under grant agreement No. 101135671 (TrustLLM) and a research grant (VIL53122) from Villum Fonden. This work was supported by the Pioneer Centre for AI, DNRF grant number P1. We would like to thank Marco Schouten, Thanos Delatolas and Stella Frank for proofreading and insightful discussions.

## REPRODUCIBILITY STATEMENT

To ensure the replicability of our findings, we release our code, allowing the research community to reproduce our results and build upon our contributions. Our experimental setup exclusively employs publicly accessible models, ensuring that all resources are readily obtainable by other researchers. We provide comprehensive details regarding all prompts and hyperparameters utilized across our experiments in Appendix I. Additionally, Appendix J contains thorough descriptions of the benchmarks and evaluation metrics employed in our study. All evaluation benchmarks utilized in this work are established and widely-used standards within the field. We include references to these resources to facilitate easy access for interested researchers. Our commitment to transparency extends beyond code release, as we meticulously detail every aspect of our experimental methodology to enable faithful reproduction of our work.

## AUTHOR STATEMENT ON THE USE OF LARGE LANGUAGE MODELS

During the preparation of this paper, large language models were used solely for minor grammar and language polishing. They were not used for research ideation, experiment design, analysis, or writing of scientific content. All research contributions are the sole responsibility of the authors.

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

# A    THEORETICAL ANALYSIS OF DIVERSITY-INDUCEMENT METHODS

Formally, let each candidate response $y$ have a latent quality $Q(y)$. The model also assigns an internal signal, such that the confidence score, $S(y)$, which is used for candidate selection. In practice, since the true quality $Q(y)$ is unknown at test time, the practical selector chooses the candidate

$$y^* = \arg\max_{y \in \mathcal{Y}} S(y).$$

We can approximate the joint distribution of $(Q, S)$ as a bivariate distribution. This distribution has means $\mu_Q$ and $\mu_S$, variances $\sigma_Q^2$ and $\sigma_S^2$, and correlation $\rho = \mathrm{Corr}(Q, S)$. The expected quality of the selected candidate can then be expressed as:

$$\mathbb{E}[Q(y^*)] \approx \mu_Q + \rho \sigma_Q k_N,$$

where $k_N = \int_{-\infty}^{\infty} N z\, \varphi(z)\, \Phi(z)^{N-1}\, dz$ is the expected maximum of $N$ standard normal variables. Here, $\varphi(z)$ is the standard normal probability density function, and $\Phi(z)$ is the standard normal cumulative distribution function. Notably, $k_N$ grows slowly as the candidate pool size $N$ increases.

Temperature sampling generates candidates with high variance in quality, $\sigma_Q$. However, these samples are often drawn from low-likelihood regions, where the model's internal confidence $S(y)$ is poorly aligned with the true quality $Q(y)$. As a result, the correlation $\rho$ between $Q$ and $S$ is small, which leads to weak scaling as more candidates are added.

In contrast, input perturbations combined with greedy decoding produce candidates with lower variance but higher mean quality $\mu_Q$. More importantly, the correlation $\rho$ is stronger, because these responses remain on the likelihood manifold where the model was trained to assign high confidence. This difference arises from the training objective of language models: next-token prediction under maximum likelihood estimation. During training, the model is optimized for greedy decoding, and temperature sampling is not simulated (e.g., there is no Gumbel-softmax trick in training), making temperature sampling less natural for the model.

Furthermore, language models are often miscalibrated, especially after post-training (Achiam et al., 2023). This miscalibration further reduces the correlation $\rho$ for candidates from temperature sampling.

Under confidence-based selection, the product $\rho \sigma_Q$ is provably larger for greedy decoding with input perturbations than for temperature sampling. This establishes greedy decoding with augmented inputs as a superior mechanism for generating diverse candidates in test-time scaling.

# B    THEORETICAL ANALYSIS OF TOKEN-LEVEL AGGREGATION VS. ANSWER-LEVEL AGGREGATION

Consider generating a response of length $T$ tokens. Let $p_t$ denote the probability of the base model generating the correct token at step $t$ given the correct prefix, with $0 < p_{\min} \le p_t \le p_{\max} < 1$.

**Token-level selection.** At each step $t$, $N \ge 2$ candidate tokens are generated. A selector with accuracy $s_t$ (probability of selecting the correct token if available) yields correctness probability $q_t = s_t \left[1 - (1 - p_t)^N\right]$. Thus, the overall correctness probability is

$$P_{\text{token}} = \prod_{t=1}^{T} q_t.$$

**Answer-level selection.** $N$ independent responses are generated. A selector with accuracy $s$ (probability of selecting the fully correct response if available) yields a correctness probability given by

$$P_{\text{answer}} = s \left[1 - \left(1 - \prod_{t=1}^{T} p_t\right)^N\right].$$

**Theorem.** Assume there exists $\delta > 0$ such that $q_t \ge (1 + \delta)p_t$ for all $t$. Then for sufficiently large $T$, token-level selection achieves a higher expected correctness probability, $P_{\text{token}} > P_{\text{answer}}$.

**Proof.** From the assumption $q_t \geq (1 + \delta)p_t$:

$$P_{\text{token}} \geq (1 + \delta)^T \prod_{t=1}^{T} p_t = (1 + \delta)^T P_{\text{correct}}$$

where $P_{\text{correct}} = \prod_{t=1}^{T} p_t$. For answer-level selection:

$$P_{\text{answer}} \leq s \cdot N \cdot P_{\text{correct}}$$

since $1 - (1 - x)^N \leq Nx$ for $x \in [0, 1]$. Comparing the two:

$$\frac{P_{\text{token}}}{P_{\text{answer}}} \geq \frac{(1 + \delta)^T P_{\text{correct}}}{sN P_{\text{correct}}} = \frac{(1 + \delta)^T}{sN}.$$

Since $\delta > 0$, $(1 + \delta)^T$ grows exponentially with $T$, while $sN$ is constant. Therefore, for sufficiently large $T$:

$$\frac{(1 + \delta)^T}{sN} > 1 \implies P_{\text{token}} > P_{\text{answer}}.$$

**Feasibility of $q_t \geq (1 + \delta)p_t$.** The condition holds if:

$$s_t \geq (1 + \delta)\frac{p_t}{1 - (1 - p_t)^N}$$

Since $1 - (1 - p_t)^N > p_t$ for $N \geq 2$ and $p_t < 1$, the right-hand side $< 1$. Thus, there exists $s_t < 1$ satisfying the inequality. For typical $p_t \in (0.5, 0.99)$ and $N \geq 2$, reasonable $s_t$ ($\approx 0.7 - 0.95$) suffice.

**Conclusion.** Token-level selection achieves superior performance because it corrects errors immediately at each generation step, preventing error propagation through the sequence. The per-token improvement factor $(1 + \delta)$ compounds multiplicatively across steps. In contrast, answer-level selection suffers from exponential decay in correctness probability ($\prod p_t$) and provides only constant-factor improvement ($sN$) through response selection.

This exponential scaling with sequence length means that token-level aggregation provides a rapidly growing advantage as responses become longer, making it especially effective for reasoning tasks such as chain-of-thought and thinking models. In these settings, each token represents a step in the reasoning process, so the ability to correct errors at every step prevents error accumulation and leads to much higher overall correctness compared to answer-level selection, whose benefits do not scale with sequence length.

Also, the superiority of increased granularity aligns with empirical observations that process reward models outperform outcome reward models (Lightman et al., 2023), and reasoning step-wise approaches like step-level self-evaluation (Xie et al., 2023) and REBASE (Wu et al., 2024) surpass answer-level methods. However, these reasoning step-wise strategies remain limited to problems where reasoning steps can be clearly defined and still fall short of token-level granularity. But, they exemplify a general trend: increased granularity yields better performance in test-time scaling.

For autoregressive generation with imperfect selectors, token-level selection achieves higher expected correctness than answer-level selection when the token selectors provide consistent multiplicative improvement over base probabilities and the response length is sufficiently large. The critical advantage comes from per-step error correction that mitigates compounding errors.

## C  AGGREGATION METHODS

We compare different token-level aggregation methods for test-time augmentation.

**Simple averaging** uniformly weights all augmented predictions by computing the arithmetic mean of probability distributions across all augmented inputs, as in Eq. 2, $\bar{p}_j(v) = \frac{1}{N} \sum_{i=1}^{N} p_{i,j}(v)$.

**Entropy-weighted averaging** assigns higher weights to more confident predictions by computing the entropy $H_{i,j} = -\sum_{v \in \mathcal{V}} p_{i,j}(v) \ln p_{i,j}(v)$ for each augmented input $i$ at step $j$, deriving weights

$w_{i,j} = e^{-H_{i,j}} / \sum_{k=1}^{N} e^{-H_{k,j}}$ through softmax over negative entropy, and aggregating as $\bar{p}_j(v) = \sum_{i=1}^{N} w_{i,j} p_{i,j}(v)$ (Chun et al., 2022).

**Majority voting** aggregates predictions by selecting the token that receives the most votes across augmented inputs. For each vocabulary token $v$ at step $j$, we compute the vote count $c_j(v) = \sum_{i=1}^{N} \mathbb{I}[\arg\max_{u \in \mathcal{V}} p_{i,j}(u) = v]$, where $\mathbb{I}[\cdot]$ is the indicator function. The final token is selected as $y_j = \arg\max_{v \in \mathcal{V}} c_j(v)$, choosing the vocabulary token with the highest vote count across all augmented predictions (Farina et al., 2024).

**Most confident token** method selects the token with the highest predicted probability across all augmented inputs, $y_j = \arg\max_{i,v} p_{i,j}(v)$. Since the predicted probability offers a noisy proxy for confidence as shown by Guo et al. (2017), this approach effectively chooses the most confident token across all augmentations (Hendrycks & Gimpel, 2017).

The experimental results in Tab. 7 reveal that averaging-based methods consistently outperform discrete voting approaches, challenging the widespread adoption of majority voting in established test-time scaling methods like self-consistency (Wang et al., 2023b). This performance hierarchy reflects fundamental differences in handling prediction uncertainty and model calibration: averaging-based approaches leverage continuous probability distributions from all augmented inputs, preserving valuable confidence information that discrete methods discard, while the majority voting and the most confident selection rely on discrete decisions from poorly calibrated predictions (Achiam et al., 2023). Simple averaging demonstrates superior robustness compared to entropy-weighted averaging, suggesting that equal weighting provides better stability than confidence-based weighting given the miscalibration issues in language models. But, confidence-based weighting can be beneficial when the model's internal confidence aligns well with true prediction quality.

Table 7: **Comparison of token-level aggregation methods for test-time augmentation.**

|  | No TTA | Most Conf. | Maj. Vote | EW Av. | Simple Av. |
|---|---|---|---|---|---|
| ChartQA | 74.2 | 73.6 | 74.8 | 76.6 | 75.6 |
| OCRBench | 72.9 | 72.0 | 72.2 | 73.4 | 73.4 |
| OCRVQA | 0.0 | 3.5 | 9.0 | 11.4 | 11.8 |
| GQA | 0.0 | 6.1 | 3.4 | 4.3 | 5.8 |
| TextVQA | 73.2 | 70.5 | 71.5 | 73.3 | 72.8 |
| AI2D | 68.5 | 68.7 | 68.7 | 68.8 | 68.8 |
| MME-RW | 27.8 | 29.5 | 30.4 | 31.0 | 31.1 |
| AMBER | 68.7 | 72.3 | 71.4 | 74.6 | 75.4 |
| COCO | 9.1 | 14.2 | 18.4 | 14.6 | 15.9 |
| **Mean** | 43.8 | 45.6 | 46.6 | 47.6 | 47.9 |

**Takeaway:** Averaging-based aggregation outperforms discrete selection methods, with simple averaging achieving the best overall performance. Continuous probability aggregation preserves valuable uncertainty information that discrete voting methods discard.

## D    AGGREGATION IN EARLY LAYERS

To understand the optimal point for feature aggregation within the model architecture, we systematically evaluate aggregation at different transformer layers rather than exclusively at the final output logits. Instead of averaging probability distributions from the final layer, we aggregate hidden representations from intermediate layers and continue forward propagation through the remaining layers using the aggregated features.

Formally, for aggregation at layer $\ell$, we compute the averaged hidden states $\bar{\mathbf{h}}_{\ell,j} = \frac{1}{N} \sum_{i=1}^{N} \mathbf{h}_{i,\ell,j}$ across all $N$ augmented inputs at generation step $j$, then feed this aggregated representation through layers $\ell+1$ to $L$ to produce the final token probabilities. This approach investigates whether early semantic representations or late linguistic features provide superior aggregation targets for multimodal understanding.

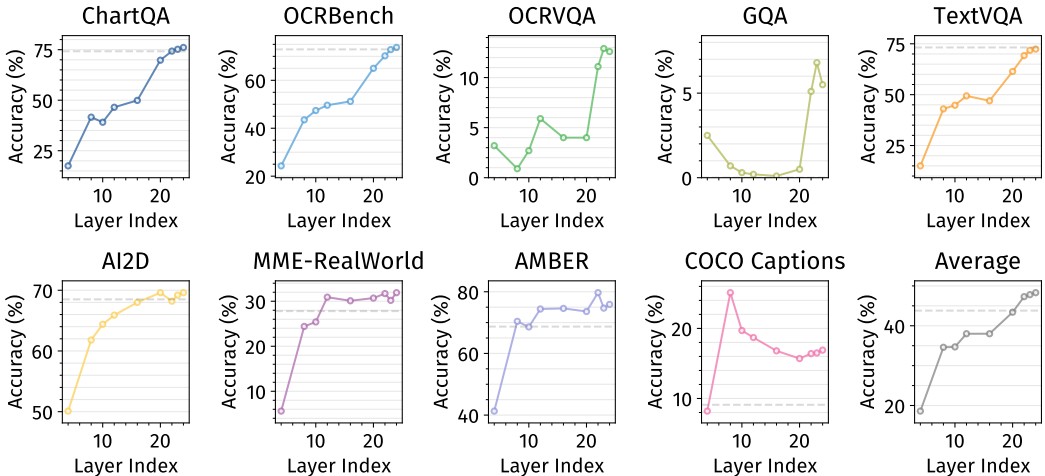

Figure 4: **Performance across aggregation layers.** Each subplot shows accuracy as a function of the transformer layer where feature aggregation occurs. Different benchmarks exhibit distinct optimal aggregation points: later layers favor language-heavy tasks (ChartQA, TextVQA), while earlier layers benefit visual reasoning tasks (OCRVQA, GQA).

The experimental results in Fig. 4 reveal task-dependent variations in optimal aggregation layers, exposing fundamental differences in how VLMs process multimodal information across different reasoning types. Three distinct patterns emerge that reflect the hierarchical nature of multimodal understanding in transformer architectures.

**Late-layer preference for linguistic reasoning.** Language-heavy benchmarks, including ChartQA, OCRBench, and TextVQA, consistently achieve optimal performance when aggregating at later layers (layers 18-24), with monotonic improvement as aggregation approaches the final output. This pattern aligns with established findings from logit lens analysis (Nostalgebraist, 2020), where later layers increasingly specialize in linguistic refinement and task-specific formatting. Recent work by Chuang et al. (2024) demonstrates that factual knowledge progressively accumulates in higher transformer layers, with later layers exhibiting stronger factual representations than earlier ones when contrasted through layer-wise decoding strategies. This hierarchical knowledge encoding suggests that deeper layers contain more refined and task-specific information essential for accurate linguistic reasoning. For tasks requiring precise text extraction and numerical reasoning, the specialized linguistic representations in deeper layers provide more reliable aggregation targets than earlier semantic features.

**Early-layer advantage for visual reasoning.** Conversely, visually-intensive benchmarks like OCRVQA and GQA demonstrate superior performance when aggregating at earlier layers (layers 6-12), with performance degrading as aggregation moves toward final layers. This counterintuitive finding reflects the model's information processing hierarchy: early layers capture rich multimodal semantic representations before aggressive compression into linguistic tokens. Recent work on visual information steering by Li et al. (2025c) reveals that visual information gradually attenuates through transformer layers, with genuine visual tokens losing prominence as language priors dominate in deeper layers. This *gradual visual information loss* phenomenon explains why early aggregation preserves critical visual details that become diluted in later layers optimized for autoregressive text generation. The early excitation pattern observed in multimodal models (Li et al., 2025c) further supports this finding, showing that semantically meaningful visual tokens achieve peak activation in penultimate or earlier layers rather than the final output layer. For tasks requiring complex visual understanding and spatial reasoning, these early semantic representations retain critical visual details that are progressively lost in later transformer layers.

**Task-specific optimal points.** Benchmarks like AI2D, AMBER, and COCO Captions exhibit intermediate optimal points around layers 10-16, suggesting these tasks benefit from balanced multimodal-linguistic representations. This intermediate optimum reflects the complex interplay between visual understanding and linguistic expression required for these tasks. The non-monotonic

patterns observed in several benchmarks indicate that aggregation timing must carefully balance semantic richness against linguistic specificity. This finding resonates with the token ranking dynamics identified by Li et al. (2025c), who demonstrate that different token types (genuine visual vs. hallucinated linguistic) achieve peak confidence at different layer depths, suggesting that optimal aggregation strategies should account for the hierarchical emergence of multimodal information processing patterns.

The observed layer preferences can be attributed to fundamental architectural properties of VLMs and align with recent discoveries about information flow in transformer-based multimodal models. Early layers primarily encode multimodal semantic relationships and spatial structures, while later layers increasingly focus on autoregressive text generation and task-specific output formatting (Tenney et al., 2019). This hierarchical specialization creates a trade-off: early aggregation preserves rich semantic diversity, but may introduce inconsistencies in linguistic expression, while late aggregation ensures coherent text generation, but may lose crucial semantic nuances. The dynamic contrastive decoding work of Chuang et al. (2024) provides additional theoretical support for our findings, demonstrating that factual knowledge evolves systematically across transformer layers, with different types of information reaching peak reliability at distinct layer depths. Our layer-dependent aggregation results extend these insights to the multimodal domain, revealing that visual and linguistic information follow distinct developmental trajectories through the network architecture.

From a theoretical perspective, these findings suggest that optimal aggregation requires matching the representational granularity to the task demands. Visual reasoning tasks benefit from the semantic spaces of early layers, where diverse augmented views can provide complementary visual interpretations. Conversely, linguistic tasks require the refined representations of later layers, where augmented inputs converge toward consistent textual expressions.

The practical implications are significant for deployment optimization. Rather than universally aggregating at final layers, practitioners can achieve substantial improvements by selecting task-appropriate aggregation points. This layer-aware aggregation strategy could be implemented adaptively, with the aggregation layer selected based on task classification or learned through validation performance. However, the computational overhead of this approach remains modest, as early aggregation actually reduces computation by bypassing later layers for individual augmented inputs.

Notably, the average performance trend shows late-layer aggregation as generally superior, but this global pattern obscures important task-specific exceptions where early aggregation provides substantial benefits. This finding challenges the common assumption that final-layer representations are universally optimal for test-time scaling and suggests that hierarchical aggregation strategies could unlock further improvements in multimodal understanding.

**Takeaway:** Optimal aggregation layers depend critically on task type: language-heavy tasks benefit from late-layer aggregation that preserves linguistic refinement, while visual reasoning tasks achieve superior performance through early-layer aggregation that retains semantic richness. Task-adaptive layer selection can provide substantial improvements over universal late-layer aggregation.

## E  COMPUTATIONAL OVERHEAD OF TTAUG

A critical consideration for deploying TTAug on resource-constrained devices is the computational overhead introduced by processing multiple augmented inputs. We analyze two implementation strategies that offer different trade-offs between memory usage and inference latency, enabling practitioners to select the most suitable approach based on their hardware constraints and requirements.

**Parallel implementation.** In the parallel strategy, all $N$ augmented inputs are processed simultaneously within a single forward pass by concatenating them into a larger batch. This approach maximizes GPU utilization and minimizes wall-clock time by leveraging parallel computation capabilities. The memory overhead scales linearly with the number of augmentations, as the model must store activations for all inputs concurrently. Peak memory consumption increases substantially due to the need to maintain intermediate representations for the entire augmented batch during forward propagation.

**Sequential implementation.** The sequential approach processes each augmented input independently in separate forward passes, accumulating token-level probability distributions for subsequent

aggregation. While this strategy significantly reduces peak memory requirements by processing only one augmentation at a time, it incurs higher latency due to the sequential nature of computation. The modest memory increase observed in sequential processing primarily stems from the accumulation of key-value cache states across multiple forward passes, which must be retained for faster generation. Note that without such a key-value caching mechanism, the sequential implementation can run on any platform capable of supporting the baseline small model, ensuring broad accessibility.

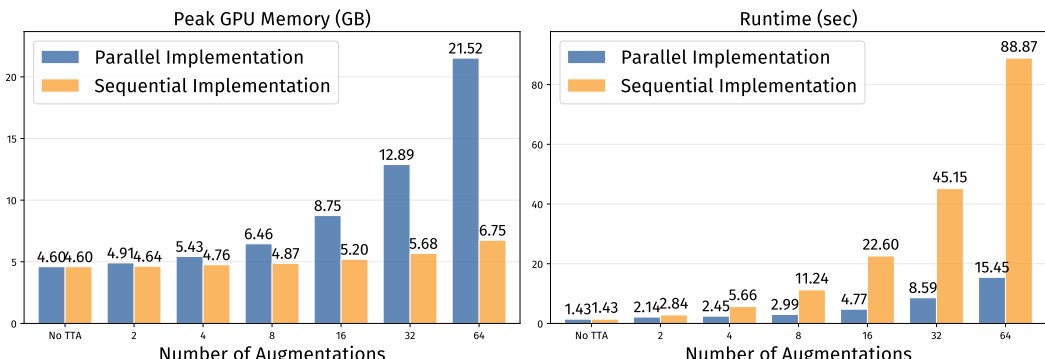

Figure 5: Overhead in peak GPU memory usage and runtime for different numbers of augmentations, comparing parallel and sequential implementation strategies.

The experimental results in Fig. 5 demonstrate distinct scaling behaviors for the two strategies, measured on an NVIDIA A100 GPU. Parallel implementation exhibits substantial memory overhead that grows approximately linearly with the number of augmentations, while sequential implementation maintains relatively constant memory usage with only minor increases due to key-value cache accumulation. Conversely, runtime overhead follows the opposite pattern: parallel processing achieves near-constant inference time regardless of augmentation count, while sequential processing incurs linear time penalties proportional to the number of augmentations.

These complementary trade-offs enable flexible deployment across diverse hardware configurations. For applications with abundant GPU memory but strict latency constraints, parallel implementation provides optimal performance. Conversely, memory-constrained environments benefit from sequential processing, which maintains feasible memory footprints at the cost of increased inference time. Practitioners can select the appropriate strategy based on their specific resource limitations and performance requirements, with both approaches representing practical extremes of the memory-latency trade-off spectrum.

While our computational overhead analysis was conducted exclusively on NVIDIA A100 GPUs, the observed patterns are highly transferable across different hardware platforms. Peak memory requirements remain platform-agnostic, determined by model architecture and batch size rather than specific hardware. Similarly, the scaling behaviors and relative trade-offs between parallel and sequential strategies exhibit consistent patterns across diverse configurations, confirming that the provided analysis is sufficient for practitioners' reference when deploying on different platforms.

**Takeaway:** Parallel implementation minimizes latency but requires substantial memory, while sequential implementation conserves memory at the cost of increased runtime. The choice between strategies depends on hardware constraints and application priorities.

## F  MULTIMODAL AUGMENTATION DECOMPOSITION

To understand the individual contributions of different modality-specific augmentations to our TTAug framework, we conduct an ablation study that isolates the effects of text-only, image-only, and combined multimodal augmentations. This analysis addresses a fundamental question in multimodal test-time scaling: whether the benefits of joint augmentation can be decomposed into additive components from individual modalities, or whether multimodal synergies introduce non-linear interactions that exceed the sum of single-modal improvements.

We design three experimental conditions to systematically evaluate modality-specific contributions. In the *text-only* condition, we apply classical textual augmentations while keeping the input image identical across all augmented samples. Conversely, the *image-only* condition applies classical visual transformations while maintaining identical text prompts. The *both* condition applies augmentations to both modalities simultaneously, representing our full TTAug framework. This decomposition enables us to quantify the relative importance of each modality and assess whether multimodal interactions produce emergent benefits beyond simple additive effects.

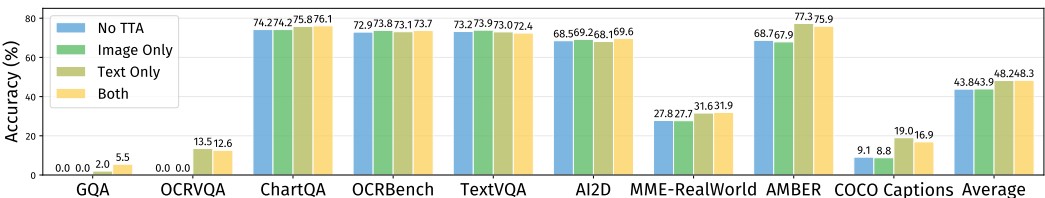

Figure 6: Performance comparison across different augmentation strategies showing the relative contributions of text-only, image-only, and combined multimodal augmentations. Each benchmark demonstrates different sensitivity patterns to modality-specific augmentations, with text augmentations consistently providing larger improvements than image augmentations across most tasks.

The experimental results in Fig. 6 reveal several critical insights about multimodal augmentation dynamics. First, combined multimodal augmentation consistently outperforms both single-modality approaches across all benchmarks, demonstrating the value of joint augmentation strategies. However, the magnitude of improvement varies substantially across different task types, suggesting that multimodal synergies are task-dependent rather than universally additive.

Second, text-only augmentation emerges as the dominant contributor to performance gains, substantially outperforming image-only augmentation across most benchmarks. This asymmetry is particularly pronounced on language-heavy benchmarks, where textual diversity appears more critical for robust understanding than visual transformations.

Third, our analysis reveals that the combined effect exhibits non-linear characteristics that cannot be predicted by simply summing the individual contributions of text-only and image-only augmentations. On several benchmarks, the joint augmentation achieves improvements that exceed the arithmetic sum of single-modality gains, indicating positive synergistic interactions between visual and textual diversity. This non-linearity suggests that multimodal augmentation creates richer semantic spaces that enhance the model's ability to extract consistent signals across diverse input representations.

The observed modality asymmetry can be attributed to several fundamental architectural and representational factors inherent to multimodal language models. First, multimodal language models typically employ heavily compressed visual representations to maintain computational efficiency, often reducing high-resolution images to low-dimensional feature vectors through aggressive pooling or patch-based tokenization (Marafioti et al., 2025). These compression operations inherently filter out fine-grained visual details that our image augmentations target, rendering subtle transformations like brightness adjustments or minor rotations largely imperceptible to the model's internal representations. Consequently, visual augmentations operate in a severely constrained semantic space where meaningful diversity is difficult to achieve.

Second, our findings align with recent interpretability research demonstrating that when one modality dominates the reasoning process, variations in the subordinate modality become largely irrelevant to model outputs (Ben Melech Stan et al., 2024). In many of our benchmarks, the textual component carries the primary semantic load, specifying the question type, reasoning requirements, and output format, while the visual component provides supplementary information. This inherent task structure naturally amplifies the impact of textual diversity while diminishing the influence of visual variations.

Third, the token-level architecture of multimodal language models creates an additional bias toward textual processing. Since both visual and textual inputs are eventually projected into a shared token space for text generation, the model's training predominantly optimizes for linguistic coherence and next-token prediction accuracy. This architectural choice inherently favors modalities that directly influence the language generation process, explaining why textual augmentations, which directly

modify the prompt structure and linguistic context, yield more substantial improvements than visual transformations that must traverse multiple encoding layers before affecting token-level decisions.

The observed modality asymmetry has important implications for practical deployment. Since text augmentation provides disproportionate benefits while requiring minimal computational overhead compared to image processing, resource-constrained applications might prioritize textual diversity generation over complex visual transformations. However, the non-additive nature of multimodal interactions suggests that completely eliminating visual augmentation would sacrifice valuable synergistic effects, supporting our unified approach that leverages both modalities while emphasizing textual diversity. Future work might explore augmentation strategies that operate directly in the compressed visual feature space or develop modality-aware weighting schemes that account for task-specific dominance patterns.

**Takeaway:** Combined multimodal augmentation outperforms single-modality approaches through non-linear synergistic effects. Text augmentations contribute more substantially than image augmentations, but their combination produces emergent benefits that exceed simple additive predictions.

## G  DETAILED RESULTS FOR DIFFERENT MODELS

Table 8: Performance comparison across SmolVLM2 family models (256M, 500M, 2.2B parameters) with no TTA, TTAug, and TTAdapt approaches.

| | SmolVLM2-256M | | | SmolVLM2-500M | | | SmolVLM2-2.2B | | |
|---|---|---|---|---|---|---|---|---|---|
| | No TTA | TT Aug | TT Adapt | No TTA | TT Aug | TT Adapt | No TTA | TT Aug | TT Adapt |
| ChartQA | 65.1 | 59.4 | 55.1 | 64.1 | 64.8 | 65.5 | 74.2 | 76.1 | 76.7 |
| OCRBench | 56.7 | 53.3 | 50.3 | 61.0 | 60.0 | 57.6 | 72.9 | 73.7 | 70.5 |
| OCRVQA | 0.2 | 0.4 | 0.3 | 0.0 | 4.6 | 5.2 | 0.0 | 12.6 | 13.8 |
| GQA | 0.1 | 5.8 | 18.4 | 0.0 | 0.0 | 0.9 | 0.0 | 5.5 | 13.5 |
| TextVQA | 47.8 | 45.1 | 40.1 | 59.9 | 58.0 | 57.7 | 73.2 | 72.4 | 70.5 |
| AI2D | 37.0 | 35.4 | 34.0 | 56.6 | 55.3 | 52.1 | 68.5 | 69.6 | 67.4 |
| MME-RW | 21.0 | 21.4 | 20.7 | 27.6 | 27.6 | 27.2 | 27.8 | 31.9 | 31.4 |
| AMBER | 29.5 | 53.3 | 43.0 | 55.3 | 56.1 | 52.8 | 68.7 | 75.9 | 72.8 |
| COCO | 29.0 | 40.6 | 38.5 | 6.2 | 9.2 | 31.6 | 9.1 | 16.9 | 35.9 |
| **Mean** | 31.8 | 35.0 | 33.4 | 36.7 | 37.3 | 38.9 | 43.8 | 48.3 | 50.3 |

Table 9: TTAug performance across Ovis2 model family (1B, 2B, 4B, 9B) and InternVL2-1B.

| | Ovis2-1B | | Ovis2-2B | | Ovis2-4B | | Ovis2-9B | | InternVL2-1B | |
|---|---|---|---|---|---|---|---|---|---|---|
| | No TTA | TT Aug | No TTA | TT Aug | No TTA | TT Aug | No TTA | TT Aug | No TTA | TT Aug |
| ChartQA | 80.4 | 81.6 | 86.6 | 85.9 | 87.6 | 87.8 | 87.4 | 87.9 | 72.1 | 72.1 |
| OCRBench | 88.8 | 84.9 | 87.3 | 86.0 | 91.2 | 89.2 | 89.2 | 87.2 | 75.7 | 75.1 |
| OCRVQA | 74.3 | 70.5 | 76.7 | 73.1 | 80.2 | 76.9 | 79.3 | 78.7 | 43.3 | 42.0 |
| GQA | 30.0 | 54.3 | 34.5 | 58.7 | 40.5 | 55.7 | 59.4 | 64.2 | 52.0 | 51.3 |
| TextVQA | 79.2 | 77.2 | 78.8 | 79.5 | 83.5 | 83.9 | 83.1 | 84.0 | 69.6 | 67.6 |
| AI2D | 76.5 | 73.3 | 81.9 | 82.2 | 84.9 | 84.5 | 87.1 | 87.2 | 52.8 | 52.6 |
| MME-RW | 35.5 | 35.6 | 38.6 | 40.5 | 45.7 | 44.1 | 45.7 | 46.5 | 13.5 | 13.3 |
| AMBER | 76.1 | 73.8 | 84.9 | 85.9 | 87.4 | 87.4 | 87.3 | 89.8 | 72.6 | 75.7 |
| COCO | 22.7 | 13.7 | 17.3 | 13.1 | 14.0 | 12.5 | 13.8 | 13.3 | 17.2 | 24.6 |
| **Mean** | 62.6 | 62.8 | 65.2 | 67.2 | 68.3 | 69.1 | 70.3 | 71.0 | 52.1 | 52.7 |

Note that TTAdapt method is not implemented for Ovis2 and InternVL model families due to practical constraints; the Unsloth library does not currently support those model families yet.

Table 10: Evaluation on diverse baseline models for reference. These models are evaluated without our methods just to establish performance baselines across different architectures. Baseline results are shown in Fig. 7.

| | Pali Gemma | xGen -MM | LLaVA -OV | Molmo -D | Idefics 2 | Janus -Pro |
|---|---|---|---|---|---|---|
| ChartQA | 40.7 | 65.0 | 72.3 | 85.8 | 31.6 | 31.0 |
| OCRBench | 61.4 | 55.5 | 61.2 | 66.3 | 63.4 | 58.9 |
| OCRVQA | 61.2 | 70.7 | 69.5 | 44.9 | 0.0 | 2.5 |
| GQA | 61.5 | 60.2 | 62.5 | 55.1 | 0.0 | 13.7 |
| TextVQA | 70.7 | 72.8 | 60.8 | 81.5 | 72.6 | 55.0 |
| AI2D | 67.9 | 73.5 | 78.2 | 80.7 | 72.2 | 67.5 |
| MME-RW | 25.4 | 35.1 | 31.1 | 36.8 | 34.3 | 23.4 |
| AMBER | 84.9 | 82.1 | 84.4 | 85.0 | 85.4 | 74.8 |
| COCO | 45.9 | 15.7 | 13.9 | 12.1 | 24.4 | 18.0 |
| **Mean** | 57.7 | 59.0 | 59.3 | 60.9 | 42.7 | 38.3 |

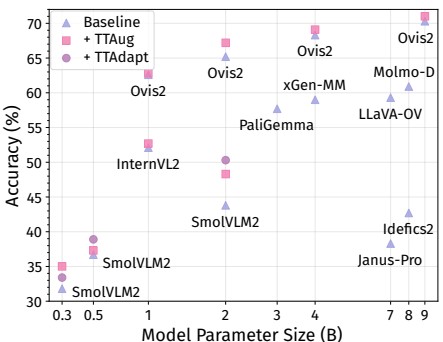

Figure 7: **Performance improvements across different models.** Each point represents a different model-strategy pair; x-axis shows model parameter size (B) using asinh scaling, and y-axis shows accuracy (%).

## H    SMALL VISION-LANGUAGE MODELS

The Transformer architecture (Vaswani et al., 2017) revolutionized language modeling, enabling models like BERT (Devlin et al., 2019) through bidirectional pretraining and GPT (Radford et al., 2019; Brown et al., 2020) via autoregressive generation. These foundational advances led to large-scale models such as GPT-3 (Brown et al., 2020) with human-like text generation abilities. More recent developments have emphasized efficiency, with LLaMA (Touvron et al., 2023) demonstrating that smaller, well-trained models can outperform earlier, larger counterparts. Open-source families including Qwen (Bai et al., 2023), InternLM (Team, 2023), and Gemma (Team, 2024) further expanded access to capable language models. In the multimodal domain, CLIP (Radford et al., 2021) introduced contrastive vision-language pretraining, facilitating strong zero-shot visual understanding. This inspired the integration of vision encoders with LLMs to produce multimodal large language models, such as GPT-4V (Achiam et al., 2023), LLaVA (Liu et al., 2023), Qwen-VL (Bai et al., 2023), and InternVL (Chen et al., 2024b). Notably, Molmo (Deitke et al., 2025) provides transparency by releasing full training data and evaluation protocols. Recently, the emergence of small vision-language models or multimodal small language models, models under 10B parameters, has shifted attention toward efficient, accessible architectures suitable for edge deployment. Examples include Ovis2 (Lu et al., 2025), InternVL2 (Chen et al., 2024b), Janus-Pro (Chen et al., 2025), Idefics2 (Laurençon et al., 2024), LLaVA-OneVision (Li et al., 2025a), Molmo (Deitke et al., 2025), XGen-MM (Xue et al., 2024), PaliGemma (Beyer et al., 2024), and the SmolVLM family (Marafioti et al., 2025), with models as small as 256M parameters. These compact models achieve competitive performance on vision-language benchmarks while significantly reducing computational cost, making them attractive for real-world, resource-constrained applications. They offer compelling advantages for practical deployment: they enable inference on consumer GPUs and edge devices, support privacy-preserving local processing, and demonstrate superior cost-performance ratios for specialized tasks (Belcak et al., 2025). But, their limited parameter capacity makes them particularly vulnerable to domain shifts, various biases, and distribution mismatches at inference time.

## I    IMPLEMENTATION DETAILS

### I.1    SELF-SELECTOR

Self-selector uses the tested VLM itself as a verifier to select one response among the candidates (Chen et al., 2024a; Parmar et al., 2025). We enforce the VLM to choose between available indices ranging from 0 to the number of augmentations. Since small VLMs are not capable of reliably following this constrained output behavior through prompt engineering alone, we employ structured generation techniques to guarantee valid responses. We use the Outlines library (Willard & Louf, 2023) for structured generation. We use the prompt given below:

> **Prompt**
>
> "{input_question}"
> Different people answered this question in different ways. Select the best response from these candidate answers:
> {responses}
> Just return the index of the best response. Return an integer between 0 and {n_aug}.

## I.2 SELF-SYNTHESIZER

Self-synthesizer method uses the tested VLM to aggregate responses into one coherent final answer (Li et al., 2025d; Jiang et al., 2023; Wang et al., 2025a; Li et al., 2025b). We use the prompt given below:

> **Prompt**
>
> "{input_question}"
> Different people answered this question in different ways. Combine these responses into a single, coherent and accurate answer:
> {responses}
> Just return the final answer.

## I.3 SELF-PARAPHRASING

Self-paraphrasing uses the text backbone of the tested VLM to paraphrase the input prompt. Since the model is not good enough to do this in one shot, we split the prompt into sentences and feed each sentence to the model to paraphrase using structured generation to get a fixed number of paraphrases. After that, we concatenate all paraphrased sentences to get the final paraphrased prompt. This approach maintains consistency with the target model's internal linguistic patterns while remaining self-contained. We use the prompt given below:

> **Prompt**
>
> You are an expert paraphraser.
> Your task is to paraphrase input text without changing its meaning. Keep the details and core content. Generate {n_aug} paraphrased versions.
> Return your output as a JSON object with the key "paraphrases", mapped to a list of {n_aug} unique paraphrased versions.
> Now, paraphrase the following text:

Since small VLMs are not capable of paraphrasing complex long prompts reliably in one shot, we first split the input text into individual sentences using spaCy (Honnibal et al., 2020) for sentence splitting. We then paraphrase each sentence independently. Also, since small VLMs are not capable of reliably following a constrained output behavior, outputting the exact number of paraphrases, through prompt engineering alone, we employ structured generation techniques to guarantee valid responses. We use a JSON schema that enforces an output with exactly desired number of paraphrases.

After obtaining paraphrases for each sentence independently, we compute the Cartesian product across all sentence-level paraphrase sets to generate diverse combinations of the complete prompt. This approach produces final paraphrased prompts by systematically combining different paraphrased versions of each sentence, ensuring both local sentence-level diversity and global prompt-level variation while maintaining semantic consistency.

## I.4 CLASSICAL IMAGE AUGMENTATIONS

We implement classical image augmentations using the Albumentations library (Buslaev et al., 2020). For each input image, we randomly select three transformations from our predefined set and apply them sequentially through a composed transformation pipeline. This random selection approach ensures diverse augmentation combinations while maintaining computational efficiency. The predefined sets for different augmentation strengths are given below.

**High**

```
A.RandomBrightnessContrast(p=0.6),
A.SafeRotate(limit=20, p=0.6, border_mode=cv2.BORDER_CONSTANT,
↪  fill=144),
A.GaussianBlur(blur_limit=(3, 7), p=0.6),
A.CLAHE(p=0.5),
A.RandomGamma(p=0.6),
A.HueSaturationValue(p=0.6),
A.RandomScale(scale_limit=0.1, p=0.6),
A.RGBShift(p=0.6),
A.MedianBlur(blur_limit=3, p=0.6),
A.ImageCompression(quality_range=(85, 95), p=0.45),
A.Sharpen(p=0.6),
A.PlanckianJitter(),
A.RandomFog(alpha_coef=0.15),
A.RandomToneCurve(),
A.Emboss(),
A.GridDistortion(),
A.Perspective(scale=0.05, fit_output=True),
A.GridDropout(ratio=0.25, random_offset=True, fill=144, p=0.66),
A.CoarseDropout(fill=144, p=0.7),
```

**Medium**

```
A.RandomBrightnessContrast(brightness_limit=0.2, contrast_limit=0.2),
A.SafeRotate(limit=15, border_mode=cv2.BORDER_CONSTANT, fill=144),
A.GaussianBlur(blur_limit=(3, 7), p=0.5),
A.CLAHE(clip_limit=3.0, p=0.4),
A.RandomGamma(gamma_limit=(80, 120), p=0.5),
A.HueSaturationValue(hue_shift_limit=15, sat_shift_limit=15,
↪  val_shift_limit=15, p=0.5),
A.RandomScale(scale_limit=0.08, p=0.5),
A.RGBShift(r_shift_limit=15, g_shift_limit=15, b_shift_limit=15),
A.MedianBlur(blur_limit=3, p=0.5),
A.ImageCompression(quality_range=(85, 95), p=0.35),
A.Sharpen(alpha=(0.2, 0.5), lightness=(0.6, 1.0), p=0.5),
A.PlanckianJitter(p=0.5),
A.RandomFog(alpha_coef=0.1, p=0.3),
A.RandomToneCurve(scale=0.2, p=0.5),
A.Emboss(alpha=(0.2, 0.5), strength=(0.5, 0.7), p=0.5),
A.GridDistortion(num_steps=5, distort_limit=0.2, p=0.5),
A.Perspective(scale=0.03, fit_output=True, p=0.5),
A.GridDropout(ratio=0.25, random_offset=True, fill=144, p=0.6),
A.CoarseDropout(fill=144, p=0.5),
```

> **Low**
>
> ```
> A.RandomBrightnessContrast(brightness_limit=0.1, contrast_limit=0.1,
> ↪   p=0.3),
> A.SafeRotate(limit=10, p=0.3, border_mode=cv2.BORDER_CONSTANT,
> ↪   fill=144),
> A.GaussianBlur(blur_limit=(3, 5), p=0.3),
> A.CLAHE(clip_limit=2.0, p=0.3),
> A.RandomGamma(gamma_limit=(90, 110), p=0.3),
> A.HueSaturationValue(hue_shift_limit=10, sat_shift_limit=10,
> ↪   val_shift_limit=10, p=0.3),
> A.RandomScale(scale_limit=0.05, p=0.3),
> A.RGBShift(r_shift_limit=10, g_shift_limit=10, b_shift_limit=10,
> ↪   p=0.3),
> A.MedianBlur(blur_limit=3, p=0.3),
> A.ImageCompression(quality_range=(85, 95), p=0.25),
> A.Sharpen(alpha=(0.1, 0.3), lightness=(0.7, 1.0), p=0.3),
> A.PlanckianJitter(p=0.3),
> A.RandomFog(alpha_coef=0.05, p=0.2),
> A.RandomToneCurve(scale=0.1, p=0.3),
> A.Emboss(alpha=(0.1, 0.3), strength=(0.3, 0.5), p=0.3),
> A.GridDistortion(num_steps=5, distort_limit=0.1, p=0.3),
> A.Perspective(scale=0.02, fit_output=True, p=0.3),
> ```

### I.5 GENERATIVE IMAGE AUGMENTATIONS

Generative augmentations use FLUX.1-dev (Labs et al., 2025) to create semantically similar but visually distinct image variants. We employ an image-to-image pipeline that, unlike traditional flow matching which starts from random noise, begins denoising from a fixed intermediate timestep with a noisy version of the input image. This approach preserves semantic similarity to the original while introducing visual diversity through the prompt "realistic image."

However, even recent generative models struggle with creating images containing textual elements (Bosheah & Bilicki, 2025). Therefore, our approach excludes text-containing images to prevent OCR corruption, using Tesseract for text detection (Smith, 2007). Two key hyperparameters control the generation process: strength (chosen as 0.25) determines the initial denoising timestep, lower values preserve more of the original image structure, and guidance scale (chosen as 3.0) controls the classifier-free guidance parameter.

While this method produces diverse and consistent image variations, it requires external diffusion models and significant computation budget. It is not practical for resource-constrained deployments.

### I.6 AGGREGATION WEIGHTS OPTIMIZATION

Aggregation weights optimization learns adaptive token-wise weights $w_{i,j}$ to replace the uniform averaging scheme in Eq. 2. At each generation step $j$, we initialize learnable parameters as $\mathbf{w}_j \in \mathbb{R}^N$ and optimize them through gradient descent to minimize the marginal entropy $H(\bar{p}_j) = -\sum_{v \in \mathcal{V}} \bar{p}_j(v) \log \bar{p}_j(v)$ of the weighted aggregated distribution. The optimization employs AdamW with adaptive learning rates and gradient clipping for stability, performing multiple micro-steps per token to achieve convergence. This approach requires minimal computational overhead with a compact computational graph, making it suitable for real-time deployment.

**Optimization Parameters.** We use the AdamW optimizer with an initial learning rate of $1 \times 10^{-2}$ and weight decay of $1 \times 10^{-4}$. The aggregation weights $\mathbf{w}_j$ are initialized uniformly as $w_{i,j} = 1/N$ where $N$ is the number of augmentations. We perform 20 optimization micro-steps per token generation step to ensure convergence of the entropy minimization objective. We reinitialize the aggregation weights before processing each new question to ensure independent optimization across different inputs.

**Gradient Clipping.** To maintain training stability, we apply gradient clipping with a maximum norm of 1.0. This prevents gradient explosion during the iterative optimization process.

**Numerical Stability.** We add a small epsilon value of $1 \times 10^{-12}$ to the logarithm computation in the entropy calculation to prevent numerical instabilities when probabilities approach zero. The softmax temperature is kept at the default value of 1.0. At each optimization step, we apply softmax normalization to the raw learnable parameters to ensure the weights sum to 1: $w_{i,j} = \frac{\exp(\theta_{i,j})}{\sum_{k=1}^{N} \exp(\theta_{k,j})}$ where $\theta_{i,j}$ are the raw learnable parameters.

**Computational Efficiency.** The optimization process uses detached probability distributions from the forward pass to prevent gradients from flowing back through the entire model, maintaining the compact computational graph.

## I.7 MODEL PARAMETER ADAPTATION

Model parameter adaptation (TTAdapt) performs iterative fine-tuning during inference using pseudolabels generated from TTAug consensus. The method employs full parameter fine-tuning with gradient checkpointing for memory efficiency and implements a three-stage iterative loop: pseudolabel generation, parameter updates, and weight reset between questions.

**Training Configuration.** We use the AdamW optimizer in the Unsloth (AI et al., 2025) library. The learning rate is set to $2 \times 10^{-6}$ with a cosine learning rate scheduler and 5 warmup steps. We apply weight decay of 0.01 for regularization and perform 6 training steps per pseudolabel iteration with a batch size of 64 and gradient accumulation steps of 2.

**Iterative Adaptation Process.** We perform 3 pseudolabel iterations per question. Each iteration generates pseudolabels using the current model state with TTAug consensus (average aggregation), then fine-tunes the model parameters using these pseudolabels as supervision. The final iteration generates the output without additional training to prevent overfitting.

**Resetting Weights.** A fundamental challenge in continual test-time adaptation is catastrophic forgetting (Niu et al., 2022; Wang et al., 2022), where models suffer severe performance degradation on original training samples after adaptation. During sample-by-sample adaptation to test streams, models can lose important information through unsupervised learning, causing rare domains to disappear while abundant ones dominate. One solution involves episodic adaptation, which means restarting from the original model for each sample rather than continual learning. Thus, in our method, model parameters are reset to their initial state before processing each new question to prevent catastrophic forgetting.

## I.8 IMPLEMENTATION DETAILS FOR TTAUG

TTAug implementation requires careful integration with the model's generation pipeline to enable efficient token-level aggregation while preserving KV caching and other optimization features. We achieve this through dynamic method patching that intercepts the sampling process without disrupting the underlying generation mechanics.

Monkey patching is critical for KV cache compatibility. We override the model's `_sample` method to inject our aggregation logic while maintaining compatibility with existing optimizations. The patched method preserves the original sampling interface but intercepts logits before token selection to perform aggregation across augmented inputs:

The modified sampling method extracts logits from multiple augmented forward passes, applies the specified aggregation strategy (uniform averaging, learned weights, or entropy optimization), and returns aggregated token selections. This approach enables seamless integration with existing generation pipelines, including beam search, nucleus sampling, and temperature scaling.

Our implementation leverages KV caching by processing augmented inputs in batches and sharing cached key-value pairs across the prefix tokens. The aggregation computation adds minimal overhead as it operates only on the final logits rather than intermediate representations, maintaining the model's inference speed while enabling test-time adaptation.

The patched method maintains full compatibility with the Transformer (Wolf et al., 2020) library's generation utilities, preserving advanced sampling techniques such as top-k, top-p, and temperature scaling. The aggregation occurs at the logit level before these sampling strategies are applied, ensuring that the enhanced diversity from TTAug benefits from sophisticated decoding procedures.

## J EVALUATION METRICS DETAILS

We provide detailed mathematical formulations for the evaluation metrics used across all benchmarks in our study. We carefully selected nine benchmarks from VLMEvalKit (Duan et al., 2024) to ensure representative, reliable, and reproducible evaluation while maintaining computational feasibility for our extensive ablation studies. Our selection prioritizes benchmarks with objective evaluation metrics (visual question answering, multiple-choice questions, yes/no questions, and captioning tasks) over LLM-as-a-judge approaches, which suffer from model bias and lack reproducibility. We exclude text-dominant benchmarks as well as specialized benchmarks focused on specific domains. The selected benchmarks represent diverse visual reasoning capabilities.

### J.1 EXACT STRING MATCHING (OCRVQA, GQA)

For datasets requiring exact string correspondence, we define the accuracy metric as:

$$\text{Accuracy} = \frac{1}{N} \sum_{i=1}^{N} \mathbb{I}[\hat{y}_i = y_i] \tag{4}$$

where $\hat{y}_i$ is the predicted answer and $y_i$ is the ground truth answer.

### J.2 VQA SCORE WITH INTER-ANNOTATOR AGREEMENT (TEXTVQA)

Following the standard VQA evaluation protocol that accounts for multiple valid answers and inter-annotator variability:

$$\text{VQA Score} = \frac{1}{N} \sum_{i=1}^{N} \frac{1}{|\mathcal{A}_i|} \sum_{k=1}^{|\mathcal{A}_i|} \min \left( 1, \frac{1}{3} \sum_{\substack{j=1 \\ j \neq k}}^{|\mathcal{A}_i|} \mathbb{I}[\hat{y}_i = y_{i,j}] \right) \tag{5}$$

where $\mathcal{A}_i$ is the set of ground truth answers for question $i$, $y_{i,j}$ represents the $j$-th ground truth answer, and $k$ indexes through each ground truth answer to simulate the leave-one-out evaluation process. For each answer $y_{i,k}$, we count how many of the remaining annotators ($j \neq k$) would agree with a prediction matching that answer. The factor $\frac{1}{3}$ reflects the standard VQA scoring that considers an answer correct if at least 3 out of $|\mathcal{A}_i|$ annotators agree. In order to be consistent with "human accuracies", machine accuracies are averaged over all $\binom{|\mathcal{A}_i|}{|\mathcal{A}_i|-1}$ sets of human annotators with leave-one-out evaluation process.

### J.3 RELAXED STRING MATCHING (CHARTQA)

For numerical and chart-based questions requiring approximate matching:

$$\text{Relaxed Accuracy} = \frac{1}{N} \sum_{i=1}^{N} \max_{j \in \mathcal{A}_i} \mathcal{R}(\hat{y}_i, y_{i,j}) \tag{6}$$

where $\mathcal{A}_i$ is the set of acceptable answers for question $i$, and $\mathcal{R}(\hat{y}, y)$ is defined as:

$$\mathcal{R}(\hat{y}, y) = \begin{cases} \mathbb{I}\left[ \frac{|v_{\hat{y}} - v_y|}{|v_y|} \leq 0.05 \right] & \text{if both are numeric} \\ \mathbb{I}[\hat{y} = y] & \text{otherwise} \end{cases} \tag{7}$$

where $v_{\hat{y}}$ and $v_y$ represent the numerical values extracted from the predicted and ground truth answers, respectively.

### J.4 SUBSTRING CONTAINMENT MATCHING (OCRBENCH)

OCRBench evaluates text recognition performance using substring containment matching:

$$\text{Accuracy} = \frac{1}{N} \sum_{i=1}^{N} \max_{j \in \mathcal{A}_i} \mathbb{I}[y_{i,j} \subseteq \hat{y}_i] \tag{8}$$

where $\mathcal{A}_i$ represents the set of acceptable answers for question $i$, and $\subseteq$ denotes substring containment.

### J.5 MULTIPLE-CHOICE AND YES/NO EXTRACTION (MME-REALWORLD, AI2D, AMBER)

For multiple-choice and yes/no questions, we extract the choice label from predictions and perform exact matching:

$$\text{Accuracy} = \frac{1}{N} \sum_{i=1}^{N} \mathbb{I}[l_i = c_i] \tag{9}$$

where $c_i \in \{A, B, C, D, ...\}$ or $\{yes, no\}$ is the correct choice label for question $i$, and $l_i$ is the extracted choice label from the predicted answer $\hat{y}_i$.

### J.6 ROUGE-L EVALUATION (COCO CAPTIONS)

We evaluate captioning quality using ROUGE-L, which measures the longest common subsequence between predicted and reference captions:

$$\text{ROUGE-L} = \frac{2 \cdot P_{\text{LCS}} \cdot R_{\text{LCS}}}{P_{\text{LCS}} + R_{\text{LCS}}} \tag{10}$$

where the precision and recall are defined as:

$$P_{\text{LCS}} = \frac{|\text{LCS}(\hat{y}, y)|}{|\hat{y}|} \tag{11}$$

$$R_{\text{LCS}} = \frac{|\text{LCS}(\hat{y}, y)|}{|y|} \tag{12}$$

and $\text{LCS}(\hat{y}, y)$ computes the longest common subsequence between the predicted caption $\hat{y}$ and reference caption $y$, with $|\cdot|$ denoting sequence length.

For multiple reference captions, we compute ROUGE-L against each reference and take the maximum score:

$$\text{ROUGE-L}_{\text{multi}} = \max_{j \in \mathcal{R}_i} \text{ROUGE-L}(\hat{y}_i, y_{i,j}) \tag{13}$$

where $\mathcal{R}_i$ is the set of reference captions for image $i$.

### J.7 IMPLEMENTATION NOTES

All text preprocessing follows consistent normalization procedures: (1) converting to lowercase, (2) stripping leading and trailing whitespace, (3) replacing multiple consecutive spaces with single spaces, and (4) removing newline characters where appropriate. For mathematical expressions in OCRBench, additional preprocessing removes all whitespace to handle formatting variations. For ChartQA relaxed matching, numerical values are extracted by handling percentage symbols (converting "X%" to X/100) and parsing floating-point numbers. For multiple-choice extraction in MME-RealWorld and AI2D, choice labels are identified using regular expressions that match single uppercase letters (A-Z) appearing in isolation or with minimal surrounding punctuation. For detailed implementation specifics and evaluation protocols, refer to VLMEvalKit (Duan et al., 2024).

## K STANDARD ERROR VALUES

This appendix presents the standard error values corresponding to all experimental results reported in the main text tables. We calculate all reported standard errors using the empirical variance of the observed per-sample scores $s_1, \ldots, s_n$. Let $\bar{s} = \frac{1}{n} \sum_i s_i$ denote the sample mean. This average accuracy is the value reported in the main text tables. Following the Central Limit Theorem, the corresponding standard error is estimated as

$$\text{SE}_{\text{C.L.T.}} = \sqrt{\text{Var}(s)/n} = \sqrt{\left(\frac{1}{n-1} \sum_i (s_i - \bar{s})^2\right)/n}.$$

Table 11: Comparison of our TTAug method against existing test-time scaling methods. Our method outperforms all others across accuracy and efficiency metrics.

(a) Mean accuracy values

| | Baseline | Others | | | | Ours |
|---|---|---|---|---|---|---|
| | | ① | ② | ③ | ④ | ⑤ |
| ChartQA | 74.2 | 74.4 | 73.4 | 72.5 | 71.7 | 75.6 |
| OCRBench | 72.9 | 72.6 | 71.9 | 70.2 | 71.9 | 73.4 |
| OCRVQA | 0.0 | 0.0 | 0.0 | 0.0 | 0.2 | 11.8 |
| GQA | 0.0 | 0.0 | 0.0 | 0.0 | 0.0 | 5.8 |
| TextVQA | 73.2 | 72.6 | 71.6 | 69.5 | 72.0 | 72.8 |
| AI2D | 68.5 | 3.1 | 69.2 | 69.1 | 67.4 | 68.8 |
| MME-RW | 27.8 | 26.2 | 26.4 | 27.6 | 27.6 | 31.1 |
| AMBER | 68.7 | 70.4 | 64.5 | 53.5 | 67.8 | 75.4 |
| COCO | 9.1 | 8.2 | 8.4 | 6.2 | 16.7 | 15.9 |
| **Mean** | 43.8 | 36.4 | 42.8 | 41.0 | 43.9 | **47.9** |

(b) Standard error values

| | Baseline | Others | | | | Ours |
|---|---|---|---|---|---|---|
| | | ① | ② | ③ | ④ | ⑤ |
| ChartQA | 1.4 | 1.4 | 1.4 | 1.4 | 1.4 | 1.4 |
| OCRBench | 1.4 | 1.4 | 1.4 | 1.4 | 1.4 | 1.4 |
| OCRVQA | 0.0 | 0.0 | 0.0 | 0.0 | 0.1 | 1.0 |
| GQA | 0.0 | 0.0 | 0.0 | 0.0 | 0.0 | 0.7 |
| TextVQA | 1.3 | 1.3 | 1.4 | 1.4 | 1.3 | 1.3 |
| AI2D | 1.5 | 0.5 | 1.5 | 1.5 | 1.5 | 1.5 |
| MME-RW | 1.4 | 1.4 | 1.4 | 1.4 | 1.4 | 1.5 |
| AMBER | 1.5 | 1.4 | 1.5 | 1.6 | 1.5 | 1.4 |
| COCO | 0.1 | 0.1 | 0.1 | 0.1 | 0.4 | 0.3 |
| **Mean** | 0.4 | 0.3 | 0.4 | 0.4 | 0.4 | 0.4 |

Table 12: Comparison of diversity-inducement methods compared to the Baseline. Input Perturbation outperforms Temperature Sampling.

(a) Mean accuracy values

| | Baseline | Temperature Sampling | | Input Perturbation | |
|---|---|---|---|---|---|
| | | ① | ② | ① | ② |
| ChartQA | 74.2 | 74.4 | 73.4 | 74.8 | 70.9 |
| OCRBench | 72.9 | 72.6 | 71.9 | 72.7 | 73.1 |
| OCRVQA | 0.0 | 0.0 | 0.0 | 12.0 | 4.5 |
| GQA | 0.0 | 0.0 | 0.0 | 7.6 | 3.7 |
| TextVQA | 73.2 | 72.6 | 71.6 | 72.3 | 72.9 |
| AI2D | 68.5 | 3.1 | 69.2 | 3.6 | 66.6 |
| MME-RW | 27.8 | 26.2 | 26.4 | 30.8 | 29.6 |
| AMBER | 68.7 | 70.4 | 64.5 | 72.7 | 67.0 |
| COCO | 9.1 | 8.2 | 8.4 | 21.2 | 13.0 |
| **Mean** | 43.8 | 36.4 | 42.8 | 40.9 | **44.6** |

(b) Standard error values

| | Baseline | Temperature Sampling | | Input Perturbation | |
|---|---|---|---|---|---|
| | | ① | ② | ① | ② |
| ChartQA | 1.4 | 1.4 | 1.4 | 1.4 | 1.4 |
| OCRBench | 1.4 | 1.4 | 1.4 | 1.4 | 1.4 |
| OCRVQA | 0.0 | 0.0 | 0.0 | 1.0 | 0.7 |
| GQA | 0.0 | 0.0 | 0.0 | 0.8 | 0.6 |
| TextVQA | 1.3 | 1.3 | 1.4 | 1.3 | 1.3 |
| AI2D | 1.5 | 0.5 | 1.5 | 0.6 | 1.5 |
| MME-RW | 1.4 | 1.4 | 1.4 | 1.5 | 1.4 |
| AMBER | 1.5 | 1.4 | 1.5 | 1.4 | 1.5 |
| COCO | 0.1 | 0.1 | 0.1 | 0.4 | 0.3 |
| **Mean** | 0.4 | 0.3 | 0.4 | 0.4 | 0.4 |

Table 13: Comparison of Answer-level versus Token-level aggregation methods. Token-level aggregation outperforms all other approaches.

(a) Mean accuracy values

| | Baseline | Answer-level | | | | Token |
|---|---|---|---|---|---|---|
| | | ① | ② | ③ | ④ | ⑤ |
| ChartQA | 74.2 | 74.8 | 70.9 | 61.1 | 72.8 | 75.6 |
| OCRBench | 72.9 | 72.7 | 73.1 | 60.9 | 71.1 | 73.4 |
| OCRVQA | 0.0 | 12.0 | 4.5 | 0.2 | 3.3 | 11.8 |
| GQA | 0.0 | 7.6 | 3.7 | 0.0 | 0.0 | 5.8 |
| TextVQA | 73.2 | 72.3 | 72.9 | 61.6 | 71.6 | 72.8 |
| AI2D | 68.5 | 3.6 | 66.6 | 69.9 | 68.0 | 68.8 |
| MME-RW | 27.8 | 30.8 | 29.6 | 29.0 | 29.2 | 31.1 |
| AMBER | 68.7 | 72.7 | 67.0 | 58.9 | 75.8 | 75.4 |
| COCO | 9.1 | 21.2 | 13.0 | 8.6 | 29.5 | 15.9 |
| **Mean** | 43.8 | 40.9 | 44.6 | 38.9 | 46.8 | **47.9** |

(b) Standard error values

| | Baseline | Answer-level | | | | Token |
|---|---|---|---|---|---|---|
| | | ① | ② | ③ | ④ | ⑤ |
| ChartQA | 1.4 | 1.4 | 1.4 | 1.5 | 1.4 | 1.4 |
| OCRBench | 1.4 | 1.4 | 1.4 | 1.5 | 1.4 | 1.4 |
| OCRVQA | 0.0 | 1.0 | 0.7 | 0.1 | 0.6 | 1.0 |
| GQA | 0.0 | 0.8 | 0.6 | 0.0 | 0.0 | 0.7 |
| TextVQA | 1.3 | 1.3 | 1.3 | 1.5 | 1.4 | 1.3 |
| AI2D | 1.5 | 0.6 | 1.5 | 1.5 | 1.5 | 1.5 |
| MME-RW | 1.4 | 1.5 | 1.4 | 1.4 | 1.4 | 1.5 |
| AMBER | 1.5 | 1.4 | 1.5 | 1.6 | 1.4 | 1.4 |
| COCO | 0.1 | 0.4 | 0.3 | 0.1 | 0.6 | 0.3 |
| **Mean** | 0.4 | 0.4 | 0.4 | 0.4 | 0.4 | 0.4 |

Table 14: Comparison of text augmentation strategies. Self-Paraphrasing ② and Classical Augmentations ③ consistently perform best.

(a) Mean accuracy values

| | Baseline | ⓪ | ① | ② | ③ | ④ |
|---|---|---|---|---|---|---|
| ChartQA | 74.2 | 74.7 | 76.9 | 76.6 | 76.1 | 71.4 |
| OCRBench | 72.9 | 73.3 | 73.5 | 72.8 | 73.7 | 70.6 |
| OCRVQA | 0.0 | 0.0 | 2.6 | 0.0 | 12.6 | 0.0 |
| GQA | 0.0 | 0.0 | 0.0 | 0.0 | 5.5 | 31.2 |
| TextVQA | 73.2 | 74.2 | 73.5 | 74.0 | 72.4 | 63.9 |
| AI2D | 68.5 | 69.8 | 69.9 | 68.4 | 69.6 | 63.9 |
| MME-RW | 27.8 | 26.6 | 30.0 | 25.9 | 31.9 | 32.1 |
| AMBER | 68.7 | 64.7 | 68.8 | 72.9 | 75.9 | 60.0 |
| COCO | 9.1 | 8.4 | 20.6 | 46.2 | 16.9 | 13.2 |
| **Mean** | 43.8 | 43.5 | 46.2 | **48.5** | 48.3 | 45.1 |

(b) Standard error values

| | Baseline | ⓪ | ① | ② | ③ | ④ |
|---|---|---|---|---|---|---|
| ChartQA | 1.4 | 1.4 | 1.3 | 1.3 | 1.3 | 1.4 |
| OCRBench | 1.4 | 1.4 | 1.4 | 1.4 | 1.4 | 1.4 |
| OCRVQA | 0.0 | 0.0 | 0.5 | 0.0 | 1.0 | 0.0 |
| GQA | 0.0 | 0.0 | 0.0 | 0.0 | 0.7 | 1.5 |
| TextVQA | 1.3 | 1.3 | 1.3 | 1.3 | 1.3 | 1.4 |
| AI2D | 1.5 | 1.5 | 1.5 | 1.5 | 1.5 | 1.5 |
| MME-RW | 1.4 | 1.4 | 1.4 | 1.4 | 1.5 | 1.5 |
| AMBER | 1.5 | 1.5 | 1.5 | 1.4 | 1.4 | 1.5 |
| COCO | 0.1 | 0.1 | 0.4 | 0.5 | 0.3 | 0.2 |
| **Mean** | 0.4 | 0.4 | 0.4 | 0.4 | 0.4 | 0.4 |

Table 15: Comparison across different image augmentation strategies. Classical Augmentations Ⓛ, Ⓗ perform the best.

(a) Mean accuracy values

| | Baseline | ⓪ | Ⓛ | Ⓜ | Ⓗ | ② | ③ |
|---|---|---|---|---|---|---|---|
| ChartQA | 74.2 | 75.8 | 77.0 | 76.4 | 76.1 | 74.1 | 75.7 |
| OCRBench | 72.9 | 73.1 | 73.7 | 73.3 | 73.7 | 72.4 | 65.3 |
| OCRVQA | 0.0 | 13.5 | 12.1 | 10.6 | 12.6 | 12.9 | 12.0 |
| GQA | 0.0 | 2.0 | 4.1 | 3.7 | 5.5 | 3.1 | 2.5 |
| TextVQA | 73.2 | 73.0 | 72.6 | 73.3 | 72.4 | 72.4 | 71.6 |
| AI2D | 68.5 | 68.1 | 69.1 | 69.0 | 69.6 | 68.9 | 67.0 |
| MME-RW | 27.8 | 31.6 | 31.8 | 32.5 | 31.9 | 32.1 | 31.1 |
| AMBER | 68.7 | 77.3 | 77.0 | 75.9 | 75.9 | 77.3 | 76.2 |
| COCO | 9.1 | 19.0 | 17.8 | 17.1 | 16.9 | 17.8 | 18.0 |
| **Mean** | 43.8 | 48.2 | 48.3 | 48.0 | **48.3** | 47.9 | 46.6 |

(b) Standard error values

| | Baseline | ⓪ | Ⓛ | Ⓜ | Ⓗ | ② | ③ |
|---|---|---|---|---|---|---|---|
| ChartQA | 1.4 | 1.4 | 1.3 | 1.3 | 1.3 | 1.4 | 1.4 |
| OCRBench | 1.4 | 1.4 | 1.4 | 1.4 | 1.4 | 1.4 | 1.5 |
| OCRVQA | 0.0 | 1.1 | 1.0 | 1.0 | 1.0 | 1.1 | 1.0 |
| GQA | 0.0 | 0.4 | 0.6 | 0.6 | 0.7 | 0.5 | 0.5 |
| TextVQA | 1.3 | 1.3 | 1.3 | 1.3 | 1.3 | 1.3 | 1.4 |
| AI2D | 1.5 | 1.5 | 1.5 | 1.5 | 1.5 | 1.5 | 1.5 |
| MME-RW | 1.4 | 1.5 | 1.5 | 1.5 | 1.5 | 1.5 | 1.5 |
| AMBER | 1.5 | 1.3 | 1.3 | 1.4 | 1.4 | 1.3 | 1.3 |
| COCO | 0.1 | 0.3 | 0.3 | 0.3 | 0.3 | 0.3 | 0.3 |
| **Mean** | 0.4 | 0.4 | 0.4 | 0.4 | 0.4 | 0.4 | 0.4 |

Table 16: Performance comparison of test-time adaptation strategies. Model parameter adaptation ② yields the best performance.

(a) Mean accuracy values

| | Baseline | TTAug | ① | ② |
|---|---|---|---|---|
| ChartQA | 74.2 | 76.1 | 76.1 | 76.7 |
| OCRBench | 72.9 | 73.7 | 73.0 | 70.5 |
| OCRVQA | 0.0 | 12.6 | 11.9 | 13.8 |
| GQA | 0.0 | 5.5 | 5.2 | 13.5 |
| TextVQA | 73.2 | 72.4 | 74.2 | 70.5 |
| AI2D | 68.5 | 69.6 | 69.7 | 67.4 |
| MME-RW | 27.8 | 31.9 | 30.9 | 31.4 |
| AMBER | 68.7 | 75.9 | 76.9 | 72.8 |
| COCO | 9.1 | 16.9 | 16.4 | 35.9 |
| **Mean** | 43.8 | 48.3 | 48.3 | **50.3** |

(b) Standard error values

| | Baseline | TTAug | ① | ② |
|---|---|---|---|---|
| ChartQA | 1.4 | 1.3 | 1.3 | 1.3 |
| OCRBench | 1.4 | 1.4 | 1.4 | 1.4 |
| OCRVQA | 0.0 | 1.0 | 1.0 | 1.1 |
| GQA | 0.0 | 0.7 | 0.7 | 1.1 |
| TextVQA | 1.3 | 1.3 | 1.3 | 1.4 |
| AI2D | 1.5 | 1.5 | 1.5 | 1.5 |
| MME-RW | 1.4 | 1.5 | 1.5 | 1.5 |
| AMBER | 1.5 | 1.4 | 1.3 | 1.4 |
| COCO | 0.1 | 0.3 | 0.3 | 0.4 |
| **Mean** | 0.4 | 0.4 | 0.4 | 0.4 |

## L QUALITATIVE RESULTS

Both classical text augmentations (Sec. 4.5) and classical image augmentations with high strength (Sec. 4.6) are applied, with 16 augmentations per sample. Thus, the shown cases correspond to samples underlying the quantitative results in Sec. 4.7.

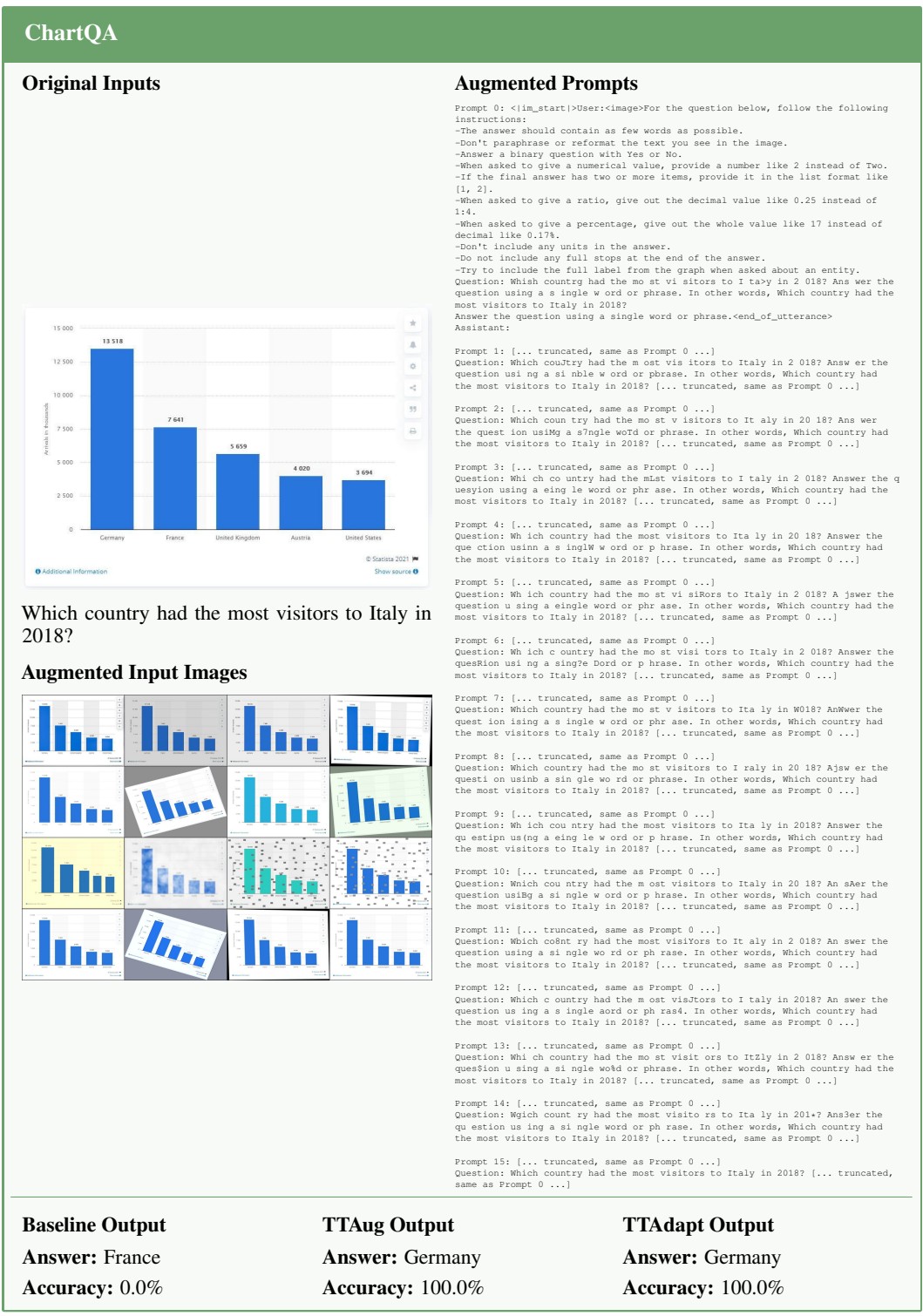

## OCRBench

**Original Inputs**

**Augmented Prompts**

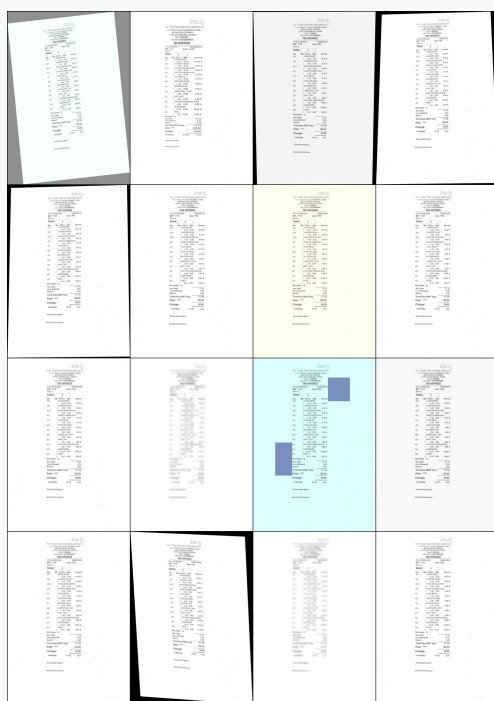

what is the total amount of this receipt? Answer this question using the text in the image directly.

**Augmented Input Images**

Prompt 0: <|im_start|>User:<image>what the amount thks receipt? this question the text the imagw directly. In other words, what is the total amount of this receipt? Answer this question using the text in the image directly.
Give a very brief answer.<end_of_utterance>
Assistant:

Prompt 1: <|im_start|>User:<image>what is the total amount of? AGswer this the in image directly. In other words, what is the total amount of this receipt? Answer this question using the text in the image directly.
Give a very brief answer.<end_of_utterance>
Assistant:

Prompt 2: <|im_start|>User:<image>wNat is the toyal amount? this question using in the image directly. In other words, what is the total amount of this receipt? Answer this question using the text in the image directly.
Give a very brief answer.<end_of_utterance>
Assistant:

Prompt 3: <|im_start|>User:<image>whWt total amount of receipt? AnsAer this text in the image directly. In other words, what is the total amount of this receipt? Answer this question using the text in the image directly.
Give a very brief answer.<end_of_utterance>
Assistant:

Prompt 4: <|im_start|>User:<image>what is the smount of this? quesGion using the text in image. In other words, what is the total amount of this receipt? Answer this question using the text in the image directly.
Give a very brief answer.<end_of_utterance>
Assistant:

Prompt 5: <|im_start|>User:<image>what is the total amount receipt? question the text the image di5ectly. In other words, what is the total amount of this receipt? Answer this question using the text in the image directly.
Give a very brief answer.<end_of_utterance>
Assistant:

Prompt 6: <|im_start|>User:<image>what is total amount of? Answer this question in the image dieectly. In other words, what is the total amount of this receipt? Answer this question using the text in the image directly.
Give a very brief answer.<end_of_utterance>
Assistant:

Prompt 7: <|im_start|>User:<image>is the total of tbis receipt? Answer thJs question using text in the. In other words, what is the total amount of this receipt? Answer this question using the text in the image directly.
Give a very brief answer.<end_of_utterance>
Assistant:

Prompt 8: <|im_start|>User:<image>is the total amount of this reVeipt? Answer questiLn text the image directly. In other words, what is the total amount of this receipt? Answer this question using the text in the image directly.
Give a very brief answer.<end_of_utterance>
Assistant:

Prompt 9: <|im_start|>User:<image>what the total aJount of this rdceipt? using the the image directly. In other words, what is the total amount of this receipt? Answer this question using the text in the image directly.
Give a very brief answer.<end_of_utterance>
Assistant:

Prompt 10: <|im_start|>User:<image>is amouHt of this? Answer th * question text in the image directly. In other words, what is the total amount of this receipt? Answer this question using the text in the image directly.
Give a very brief answer.<end_of_utterance>
Assistant:

Prompt 11: <|im_start|>User:<image>what is the total amount of this receipt? thix using in iKage. In other words, what is the total amount of this receipt? Answer this question using the text in the image directly.
Give a very brief answer.<end_of_utterance>
Assistant:

Prompt 12: <|im_start|>User:<image>is total amounY of this? Answer this question using text in the. In other words, what is the total amount of this receipt? Answer this question using the text in the image directly.
Give a very brief answer.<end_of_utterance>
Assistant:

Prompt 13: <|im_start|>User:<image>is the amount of receipt? ques5ion using the text in the direVtly. In other words, what is the total amount of this receipt? Answer this question using the text in the image directly.
Give a very brief answer.<end_of_utterance>
Assistant:

Prompt 14: <|im_start|>User:<image>wYat the of this receipt? thie question the in the image directly. In other words, what is the total amount of this receipt? Answer this question using the text in the image directly.
Give a very brief answer.<end_of_utterance>
Assistant:

Prompt 15: <|im_start|>User:<image>what is the total amount of this receipt? Answer this question using the text in the image directly.
Give a very brief answer.<end_of_utterance>
Assistant:

**Baseline Output**

**Answer:** 100.00

**Accuracy:** 0.0%

**TTAug Output**

**Answer:** 71.10

**Accuracy:** 100.0%

**TTAdapt Output**

**Answer:** 71.10

**Accuracy:** 100.0%

## OCRVQA

**Original Inputs**

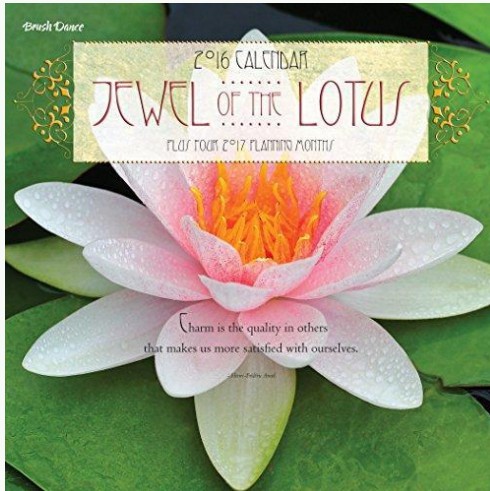

Who is the author of this book?

**Augmented Input Images**

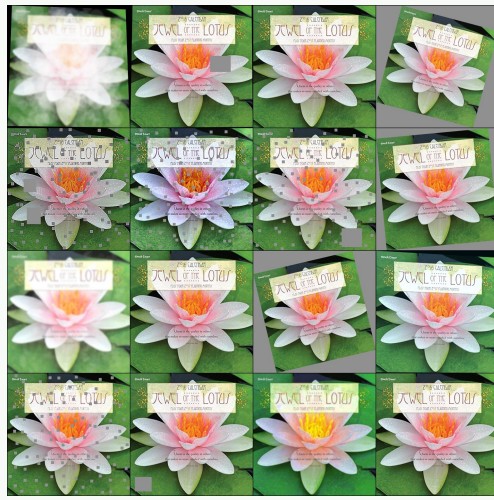

**Augmented Prompts**

Prompt 0: <|im_start|>User:<image>Answer the question using a single word or phGase. Who is the author of thiC book? In other words, Who is the author of this book?
Answer the question using a single word or phrase.
Give a very brief answer.<end_of_utterance>
Assistant:

Prompt 1: <|im_start|>User:<image>qnswer the question using a single word or pTrase. Who is the author of this book? In other words, Who is the author of this book?
Answer the question using a single word or phrase.
Give a very brief answer.<end_of_utterance>
Assistant:

Prompt 2: <|im_start|>User:<image>Answer the question usjng a single word or phraDe. Who is the author of this book? In other words, Who is the author of this book?
Answer the question using a single word or phrase.
Give a very brief answer.<end_of_utterance>
Assistant:

Prompt 3: <|im_start|>User:<image>Answer the question usinT a single Sord or phrase. Who is the author of this book? In other words, Who is the author of this book?
Answer the question using a single word or phrase.
Give a very brief answer.<end_of_utterance>
Assistant:

Prompt 4: <|im_start|>User:<image>Answer the question using a Eingle word or (hrase. Who is the author of this book? In other words, Who is the author of this book?
Answer the question using a single word or phrase.
Give a very brief answer.<end_of_utterance>
Assistant:

Prompt 5: <|im_start|>User:<image>Answer the question using a sinNle word or phrase. Who is the author of tgis book? In other words, Who is the author of this book?
Answer the question using a single word or phrase.
Give a very brief answer.<end_of_utterance>
Assistant:

Prompt 6: <|im_start|>User:<image>Answer the quest(on using a s&ngle word or phrase. Who is the author of this book? In other words, Who is the author of this book?
Answer the question using a single word or phrase.
Give a very brief answer.<end_of_utterance>
Assistant:

Prompt 7: <|im_start|>User:<image>AnsweF the question ^sing a single word or phrase. Who is the author of this book? In other words, Who is the author of this book?
Answer the question using a single word or phrase.
Give a very brief answer.<end_of_utterance>
Assistant:

Prompt 8: <|im_start|>User:<image>Answer the question uAing a single word or phrase. Who is the authoT of this book? In other words, Who is the author of this book?
Answer the question using a single word or phrase.
Give a very brief answer.<end_of_utterance>
Assistant:

Prompt 9: <|im_start|>User:<image>AnsweF the qHestion using a single word or phrase. Who is the author of this book? In other words, Who is the author of this book?
Answer the question using a single word or phrase.
Give a very brief answer.<end_of_utterance>
Assistant:

Prompt 10: <|im_start|>User:<image>Answe$ the question using a single word or phrase. Who is the author of this Hook? In other words, Who is the author of this book?
Answer the question using a single word or phrase.
Give a very brief answer.<end_of_utterance>
Assistant:

Prompt 11: <|im_start|>User:<image>Answer the question using a single word or phrase. Who is the autho# of thks book? In other words, Who is the author of this book?
Answer the question using a single word or phrase.
Give a very brief answer.<end_of_utterance>
Assistant:

Prompt 12: <|im_start|>User:<image>Answer the question using a single word or ph#ase. Who is the author of %his book? In other words, Who is the author of this book?
Answer the question using a single word or phrase.
Give a very brief answer.<end_of_utterance>
Assistant:

Prompt 13: <|im_start|>User:<image>AnAwer the question using a single word or phrase. Who is the autTor of this book? In other words, Who is the author of this book?
Answer the question using a single word or phrase.
Give a very brief answer.<end_of_utterance>
Assistant:

Prompt 14: <|im_start|>User:<image>Answer the question using a s7ngle word or phrQse. Who is the author of this book? In other words, Who is the author of this book?
Answer the question using a single word or phrase.
Give a very brief answer.<end_of_utterance>
Assistant:

Prompt 15: <|im_start|>User:<image>Who is the author of this book?
Answer the question using a single word or phrase.
Give a very brief answer.<end_of_utterance>
Assistant:

| Baseline Output | TTAug Output | TTAdapt Output |
|---|---|---|
| **Answer:** Brushy. | **Answer:** Brush Dance. | **Answer:** Brush Dance. |
| **Accuracy:** 0.0% | **Accuracy:** 100.0% | **Accuracy:** 100.0% |

## GQA

**Original Inputs**

**Augmented Prompts**

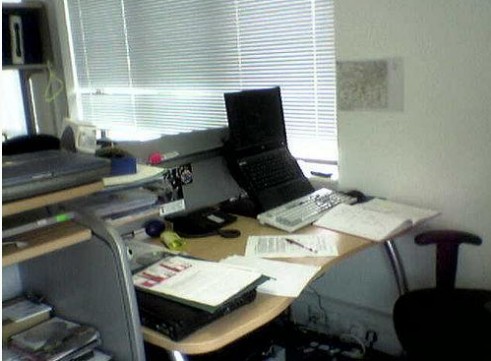

What's in front of the window?

**Augmented Input Images**

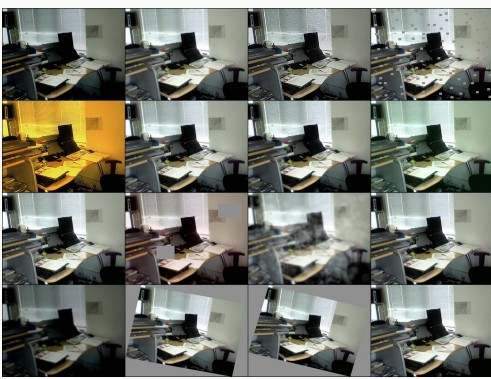

```
Prompt 0: <|im_start|>User:<image>W hat ' s in feont of the win dow? Answer
the que stiin u sing a si ngle wo rd or phrWse. In other words, What's in
front of the window?
Answer the question using a single word or phrase.<end_of_utterance>
Assistant:

Prompt 1: <|im_start|>User:<image>Wh at ' s in fro nt of the window? Ansder
the qu estlon usi ng a sing le word or ph rqse. In other words, What's in
front of the window?
Answer the question using a single word or phrase.<end_of_utterance>
Assistant:

Prompt 2: <|im_start|>User:<image>W hat ' s in fro nt of the win dow? Abswer
the question uxing a sihg le wo rd or phra se. In other words, What's in
front of the window?
Answer the question using a single word or phrase.<end_of_utterance>
Assistant:

Prompt 3: <|im_start|>User:<image>Wh at ' s in fro nt of the wi ndow? AnXwer
the queAtion us ing a si ngle w)rd or phr ase. In other words, What's in
front of the window?
Answer the question using a single word or phrase.<end_of_utterance>
Assistant:

Prompt 4: <|im_start|>User:<image>Wh at ' s in fr ont of the wi ndod? AnsAer
the queWt ion using a s ingle wo rd or phrase. In other words, What's in
front of the window?
Answer the question using a single word or phrase.<end_of_utterance>
Assistant:

Prompt 5: <|im_start|>User:<image>W hat ' s in f romt of the window? Answ er
the qu estion us ing a sin gle Sord or lhrase. In other words, What's in
front of the window?
Answer the question using a single word or phrase.<end_of_utterance>
Assistant:

Prompt 6: <|im_start|>User:<image>W hat ' s in fro nt of the windo#? Answer
the ques tion using a s&ng le wo rd or p hGase. In other words, What's in
front of the window?
Answer the question using a single word or phrase.<end_of_utterance>
Assistant:

Prompt 7: <|im_start|>User:<image>Wmat ' s in fro nt of the wi ndow? Ans wer
the quest ion usigg a sin gle Sord or ph rase. In other words, What's in
front of the window?
Answer the question using a single word or phrase.<end_of_utterance>
Assistant:

Prompt 8: <|im_start|>User:<image>What ' s in f ront of the !ind ow? Answer
the Wuesti on usi ng a single wo rd or p hrQse. In other words, What's in
front of the window?
Answer the question using a single word or phrase.<end_of_utterance>
Assistant:

Prompt 9: <|im_start|>User:<image>Wh at ' s in fr0nt of the wi ndow? An swer
the qu estKon using a sin gle w9rd or phr ase. In other words, What's in
front of the window?
Answer the question using a single word or phrase.<end_of_utterance>
Assistant:

Prompt 10: <|im_start|>User:<image>Wh at ' s in front of the qind ow? Ans
wer the q Testion usi ng a s ingle w ord or phraAe. In other words, What's
in front of the window?
Answer the question using a single word or phrase.<end_of_utterance>
Assistant:

Prompt 11: <|im_start|>User:<image>Wh at ' s in f%ont of the win dow? Ajswer
the que stion usi ng a simg le word or p hrase. In other words, What's in
front of the window?
Answer the question using a single word or phrase.<end_of_utterance>
Assistant:

Prompt 12: <|im_start|>User:<image>W hat ' s in front of the wi ndow? Ans
wer the questi on usinN a s ingIe w ord or phraEe. In other words, What's in
front of the window?
Answer the question using a single word or phrase.<end_of_utterance>
Assistant:

Prompt 13: <|im_start|>User:<image>W hat ' s in front of the winE ow? Answer
the qu 4stion using a sing le w ord or p Urase. In other words, What's in
front of the window?
Answer the question using a single word or phrase.<end_of_utterance>
Assistant:

Prompt 14: <|im_start|>User:<image>dhat ' s in fro nt of the win dow? An
sw2r the Aues tion using a s ingle word or phr ase. In other words, What's
in front of the window?
Answer the question using a single word or phrase.<end_of_utterance>
Assistant:

Prompt 15: <|im_start|>User:<image>What's in front of the window?
Answer the question using a single word or phrase.<end_of_utterance>
Assistant:
```

**Baseline Output**

**Answer:** Blinds.

**Accuracy:** 0.0%

**TTAug Output**

**Answer:** Desk.

**Accuracy:** 100.0%

**TTAdapt Output**

**Answer:** Desk.

**Accuracy:** 100.0%

## TextVQA

**Original Inputs**

**Augmented Prompts**

Prompt 0: <|im_start|>User:<image>Answer the following question about the image using as few words as possible. Follow these additional instructions:
-Always answer a binary question with Yes or No.
-When asked what time it is, reply with the time seen in the image.
-Do not put any full stops at the end of the answer.
-Do not put quotation marks around the answer.
-An answer with one or two words is favorable.
-Do not apply common sense knowledge. The answer can be found in the image.
Question: which of ese b ooks re adapted? que stion usi ng a single word or phra. In other words, which of these books was recently adapted by netflix?
Answer the question using a single word or phrase.<end_of_utterance>
Assistant:

Prompt 1: [... truncated, same as Prompt 0 ...]
Question: w of these boo was recen adapted by netflix? the ques tion using sing le word. In other words, which of these books was recently adapted by netflix?
Answer the question using a single word or phrase.<end_of_utterance>
Assistant:

Prompt 2: [... truncated, same as Prompt 0 ...]
Question: of t hese was cently apted by netflix? question using a sing le wo or ase. In other words, which of these books was recently adapted by netflix?
Answer the question using a single word or phrase.<end_of_utterance>
Assistant:

Prompt 3: [... truncated, same as Prompt 0 ...]
Question: of these was recen adapted netfl? A nswer questi on us ing a word or se. In other words, which of these books was recently adapted by netflix?
Answer the question using a single word or phrase.<end_of_utterance>
Assistant:

Prompt 4: [... truncated, same as Prompt 0 ...]
Question: of t hese boo was by? An swer the question usi ng a single word or se. In other words, which of these books was recently adapted by netflix?
Answer the question using a single word or phrase.<end_of_utterance>
Assistant:

Prompt 5: [... truncated, same as Prompt 0 ...]
Question: which of t hese was adapted netflix? the ques tion ng a word phrase. In other words, which of these books was recently adapted by netflix?
Answer the question using a single word or phrase.<end_of_utterance>
Assistant:

Prompt 6: [... truncated, same as Prompt 0 ...]
Question: ch th ese b ooks was adapted by tflix? Answer question sin gle w ord or. In other words, which of these books was recently adapted by netflix?
Answer the question using a single word or phrase.<end_of_utterance>
Assistant:

Prompt 7: [... truncated, same as Prompt 0 ...]
Question: ich t books recently by netf lix? Ans wer the q using le word or phrase. In other words, which of these books was recently adapted by netflix?
Answer the question using a single word or phrase.<end_of_utterance>
Assistant:

Prompt 8: [... truncated, same as Prompt 0 ...]
Question: whi ch of the se books was adapted by? A nswer the sing a wo hrase. In other words, which of these books was recently adapted by netflix?
Answer the question using a single word or phrase.<end_of_utterance>
Assistant:

Prompt 9: [... truncated, same as Prompt 0 ...]
Question: of t bo oks was recently by? using a si ngle wo rd or phra se. In other words, which of these books was recently adapted by netflix?
Answer the question using a single word or phrase.<end_of_utterance>
Assistant:

Prompt 10: [... truncated, same as Prompt 0 ...]
Question: whi ch books recent ly adapted by ix? wer the qu using a le word phrase. In other words, which of these books was recently adapted by netflix?
Answer the question using a single word or phrase.<end_of_utterance>
Assistant:

Prompt 11: [... truncated, same as Prompt 0 ...]
Question: ich se books was pted by netflix? Answer the ion using a wo or phra se. In other words, which of these books was recently adapted by netflix?
Answer the question using a single word or phrase.<end_of_utterance>
Assistant:

Prompt 12: [... truncated, same as Prompt 0 ...]
Question: w t hese books was recently adapted n? Answer que stion using a wo or ase. In other words, which of these books was recently adapted by netflix?
Answer the question using a single word or phrase.<end_of_utterance>
Assistant:

Prompt 13: [... truncated, same as Prompt 0 ...]
Question: wh was rece ntly ad apted netflix? A nswer using a s word or ph rase. In other words, which of these books was recently adapted by netflix?
Answer the question using a single word or phrase.<end_of_utterance>
Assistant:

Prompt 14: [... truncated, same as Prompt 0 ...]
Question: which of the books by netfl ix? Answer the questi using a single ord or phr. In other words, which of these books was recently adapted by netflix?
Answer the question using a single word or phrase.<end_of_utterance>
Assistant:

Prompt 15: [... truncated, same as Prompt 0 ...]
Question: which of these books was recently adapted by netflix?
Answer the question using a single word or phrase.<end_of_utterance>
Assistant:

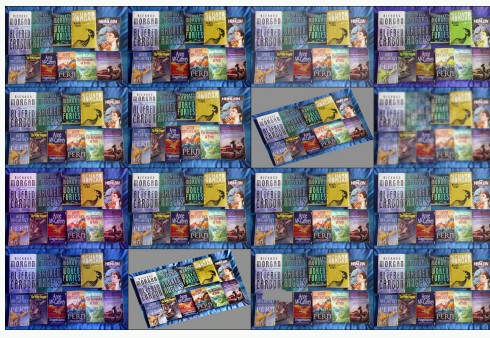

which of these books was recently adapted by net-flix?

**Augmented Input Images**

| | | |
|---|---|---|
| **Baseline Output** | **TTAug Output** | **TTAdapt Output** |
| **Answer:** broken angels | **Answer:** altered carbon | **Answer:** altered carbon |
| **Accuracy:** 0.0% | **Accuracy:** 100.0% | **Accuracy:** 100.0% |

## AI2D

**Original Inputs**

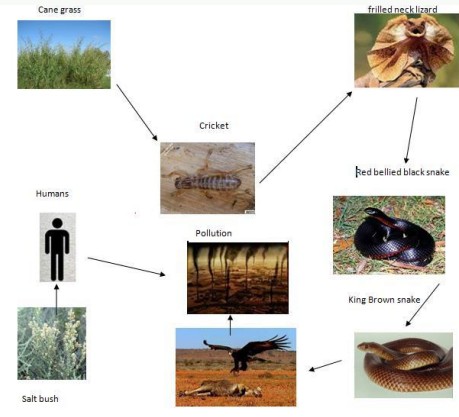

What would happen if the cricket population decreased? The choices are listed below:
(A) lizards would decrease
(B) eagle would increase
(C) king brown snake would increast
(D) salt bush would decrease

**Augmented Input Images**

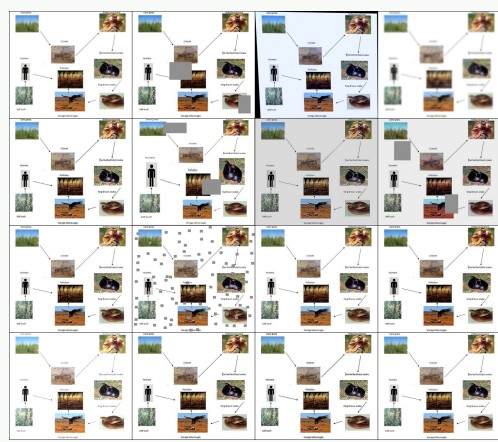

**Augmented Prompts**

```
Prompt 0: <|im_start|>User:<image>Ques tion: What wo^ld h appen if the c
ricket popula tion deSreased? In other words, Question: What would happen if
the cricket population decreased?Options:
A. lizards would decrease
B. eagle would increase
C. king brown snake would increast
D. salt bush would decrease
Answer with the letter.<end_of_utterance>
Assistant: Answer:

Prompt 1: <|im_start|>User:<image>Que stion: W hat wo uld happen if the cri
cket p)pulation decr2ased? In other words, [... truncated, same as Prompt 0
...]

Prompt 2: <|im_start|>User:<image>Question: Wh at woulS happen if the cr
icke^ pop ulation decrea sed? In other words, [... truncated, same as Prompt
0 ...]

Prompt 3: <|im_start|>User:<image>Question: What wo uld h wppen if the cri
cket p opulation Eecreased? In other words, [... truncated, same as Prompt 0
...]

Prompt 4: <|im_start|>User:<image>Qu3s tion: Dhat wo uld hap pen if the cri
cket population decreased? In other words, [... truncated, same as Prompt 0
...]

Prompt 5: <|im_start|>User:<image>Quwstion: WhQt wou ld happen if the cr
icket populati on d ecreased? In other words, [... truncated, same as Prompt
0 ...]

Prompt 6: <|im_start|>User:<image>Que sti+n: W hat wou;d happen if the cri
cket populati on decreased? In other words, [... truncated, same as Prompt 0
...]

Prompt 7: <|im_start|>User:<image>Question: Wuat would hap) en if the crick
et populati on decreas ed? In other words, [... truncated, same as Prompt 0
...]

Prompt 8: <|im_start|>User:<image>Question: W hat wPuld happ en if the cr7c
ket population decrea sed? In other words, [... truncated, same as Prompt 0
...]

Prompt 9: <|im_start|>User:<image>Quest ion: W hat wouId happen if the cr
icket pop7lation decr eased? In other words, [... truncated, same as Prompt
0 ...]

Prompt 10: <|im_start|>User:<image>QueC tion: Wh at wou ld h xppen if the
cricket population decreased? In other words, [... truncated, same as Prompt
0 ...]

Prompt 11: <|im_start|>User:<image>Quest ion: What wo uld h appen if the
crifket populayion dec reased? In other words, [... truncated, same as
Prompt 0 ...]

Prompt 12: <|im_start|>User:<image>QuFst ion: What wo uld happen if the
cricket pop u;ation decra sed? In other words, [... truncated, same as
Prompt 0 ...]

Prompt 13: <|im_start|>User:<image>Qu estJon: W hat wou ld happen if the c
ricket Lopulation decreased? In other words, [... truncated, same as Prompt
0 ...]

Prompt 14: <|im_start|>User:<image>Question: W hat w oulE happ en if the
cricket popula ti)n decreased? In other words, [... truncated, same as
Prompt 0 ...]

Prompt 15: <|im_start|>User:<image>Question: What would happen if the
cricket population decreased?Options:
A. lizards would decrease
B. eagle would increase
C. king brown snake would increast
D. salt bush would decrease
Answer with the letter.<end_of_utterance>
Assistant: Answer:
```

| **Baseline Output** | **TTAug Output** | **TTAdapt Output** |
|---|---|---|
| **Answer:** C | **Answer:** A | **Answer:** A |
| **Accuracy:** 0.0% | **Accuracy:** 100.0% | **Accuracy:** 100.0% |

## MME-RealWorld

**Original Inputs**

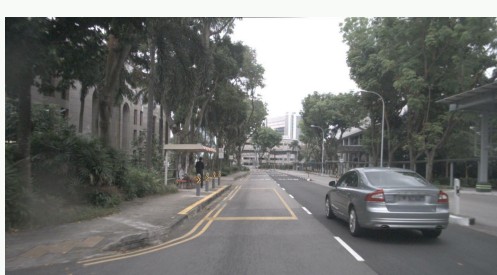

This image shows the front view of the ego car. What is the future state of the black pants pedestrian in the middle? The choices are listed below:
(A) Turn left.
(B) Stationary.
(C) Keep going straight.
(D) Turn right.
(E) The image does not feature the object.

**Augmented Input Images**

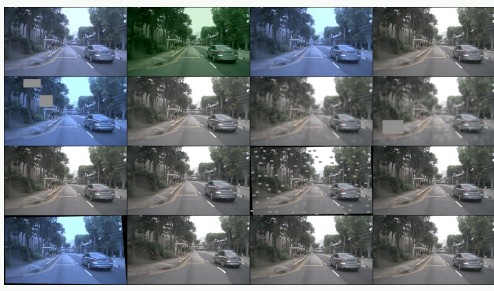

**Augmented Prompts**

```
Prompt 0: <|im_start|>User:<image>This image shows the front view of the ego car. What is the
future state of black pants pedestrian middle? The are listed below: (A) Turn. (B) Stationary.
(C) Keep going straight. (D) Turn right. (E) The image does not feature the object. Respond
only letter (A, B, C, D, or E) of the correct option. Select the best answer to the above
multiple - choice question based on the image. The answer: In other words, This image shows
the front view of the ego car. What is the future state of the black pants pedestrian in the
middle? The choices are listed below:
(A) Turn left.
(B) Stationary.
(C) Keep going straight.
(D) Turn right.
(E) The image does not feature the object.
Select the best answer to the above multiple-choice question based on the image. Respond with
only the letter (A, B, C, D, or E) of the correct option.
The best answer is:<end_of_utterance>
Assistant:
```

```
Prompt 1: <|im_start|>User:<image>This image shows front view of the ego. The choices are
listed: (A) Turn. (B ). What is future state of the black pants in the middle? (C) Keep going
straight. (D) Turn right. (E) The does not feature object. the best answer to the above
multiple - choice question based on the image. The best answer is: Respond with only the
letter (A, B, C, D, or E) of the correct option. In other words, [... truncated, same as
Prompt 0 ...]
```

```
Prompt 2: <|im_start|>User:<image>This image the view the car. What is the state of the black
pants pedestrian in the middle? The choices are listed below: (A) Turn left. (C) Keep going
straight. (B) Stationary. () The image does not feature the object. () Turn right. Respond
with only the letter (A, B, C, D, or E) of the option. the best answer to above multiple -
choice question based on the image. The best answer is: In other words, [... truncated, same
as Prompt 0 ...]
```

```
Prompt 3: <|im_start|>User:<image>The is: What is the future state of the black pants
pedestrian in the middle? (B ). The choices are listed below: () Turn left. (C) Keep going
straight. () right. Select the best answer to the above multiple - choice question based on
the image. (E) image does not feature the object. Respond with only the letter (, , C, D, E)
of the correct option. This image shows the front view of the ego car. In other words, [...
truncated, same as Prompt 0 ...]
```

```
Prompt 4: <|im_start|>User:<image>What is the future state of the black pants pedestrian in
the? This shows the front view of the ego car. The choices are listed below: (A) left. (C)
Keep going straight. (B ). () Turn right. (E) The image does not feature the object. the
answer to the multiple - choice question based on the image. The best answer is: Respond only
the letter (, B, C, D, or E) of the option. In other words, [... truncated, same as Prompt 0
...]
```

```
Prompt 5: <|im_start|>User:<image>This image shows the front view of the ego car. What the
future state of the black pedestrian in the middle? The choices are listed below: () Turn
left. (C) Keep going straight. (B ). () The image does not feature the object. (D) Turn right.
Select the best answer to above multiple - choice question on the image. The answer is:
Respond with only the letter (A, , C, D, or) of the correct option. In other words, [...
truncated, same as Prompt 0 ...]
```

```
Prompt 6: <|im_start|>User:<image>The best answer is: The choices are listed below: (A) Turn
left. What is the state the black pedestrian in the middle? (B) Stationary. (C) Keep going.(D)
Turn right. the best answer to the above multiple - choice question based on the image. (E)
The image does not feature the object. Respond with only the letter (A, , C, , or) of the
correct option. This image shows front view of the ego. In other words, [... truncated, same
as Prompt 0 ...]
```

```
Prompt 7: <|im_start|>User:<image>This image shows the front view of the ego car. What is the
future state of the black pants in the? (B) Stationary. The choices are listed below: (A)
Turn. (D) Turn right. (C) Keep going straight. (E) The image does not the object. Respond with
only the letter (A, B, C, D, or) of correct. Select the best answer the above multiple -
choice question based the image. The best answer: In other words, [... truncated, same as
Prompt 0 ...]
```

```
Prompt 8: <|im_start|>User:<image>This image shows the front view the ego car. The choices are
listed below: (A) Turn left. (B) Stationary. What is the future state of black pedestrian in
middle? (C) Keep going. (D) Turn right. Select the best answer above multiple - choice
question based on the image. (E) The image does feature the object. Respond with only the
letter (A, B, , D, or E) of the option. The best answer is: In other words, [... truncated,
same as Prompt 0 ...]
```

```
Prompt 9: <|im_start|>User:<image>The best answer: This image shows front of the ego car. The
choices are below: (A) Turn left. (B) Stationary. (C) Keep going straight. (D) Turn. () The
image does not feature the. Respond only the letter (A, B, C, D, or E) of the correct option.
Select best answer to the above multiple - choice question based on the image. What is the
future state of black pants pedestrian in the middle? In other words, [... truncated, same as
Prompt 0 ...]
```

```
Prompt 10: <|im_start|>User:<image>The best answer: What the future state of the black
pedestrian in the middle? The choices listed below: (A) Turn left. (B) Stationary. (C) Keep
straight. Select the best answer to the above multiple - choice question based on the image.
(D) Turn right. (E) The image does not feature the object. Respond with only letter (A, B, C,
D, E) the correct option. This image shows the front of the ego car. In other words, [...
truncated, same as Prompt 0 ...]
```

```
Prompt 11: <|im_start|>User:<image>This image the front view of. What is the future state of
the black pants pedestrian in the middle? (A) Turn left. (B) Stationary. (C) Keep going
straight. (D) right. Select the best to the above - choice based on the image. () The image
does not feature the object. with only the letter (A, B, C, D, or E) of the correct option.
The best answer is: In other words, [... truncated, same as Prompt 0 ...]
```

```
Prompt 12: <|im_start|>User:<image>This shows the view of the ego car. What is future state of
the pants pedestrian in the middle? The choices below: (A) Turn left. (C) Keep going straight.
(B) Stationary. (D) Turn right. (E) does not feature the object. Respond with only the letter
(A, B, C, D, or E) of the correct option. The best answer is: Select the best answer to above
multiple - choice based on the image. In other words, [... truncated, same as Prompt 0 ...]
```

```
Prompt 13: <|im_start|>User:<image>This image shows the front view of ego car. is the state of
the black pants in the middle? The choices listed below: (A) Turn. (B) Stationary. (C) Keep
going straight. () Turn. (E) The image does not feature the object. Respond with only the
letter (, B, C, D, or E) of the correct option. Select the best answer to the multiple -
choice question based on the image. The best answer is: In other words, [... truncated, same
as Prompt 0 ...]
```

```
Prompt 14: <|im_start|>User:<image>The best answer is: What is the future state the black
pants pedestrian in the middle? The choices listed below: (A) Turn. (B) Stationary. (D) right.
(C) Keep going straight. Select the best answer to the above multiple - choice question based
on the image. (E) The image does not feature the object. Respond with the letter (, ,, , or E)
of the correct option. This image shows the front view of ego car. In other words, [...
truncated, same as Prompt 0 ...]
```

```
Prompt 15: <|im_start|>User:<image>This image shows the front view of the ego car. What is the
future state of the black pants pedestrian in the middle? The choices are listed below:
(A) Turn left.
(B) Stationary.
(C) Keep going straight.
(D) Turn right.
(E) The image does not feature the object.
Select the best answer to the above multiple-choice question based on the image. Respond with
only the letter (A, B, C, D, or E) of the correct option.
The best answer is:<end_of_utterance>
Assistant:
```

| **Baseline Output** | **TTAug Output** | **TTAdapt Output** |
|---|---|---|
| **Answer:** E | **Answer:** B | **Answer:** B |
| **Accuracy:** 0.0% | **Accuracy:** 100.0% | **Accuracy:** 100.0% |

## AMBER

**Original Inputs**

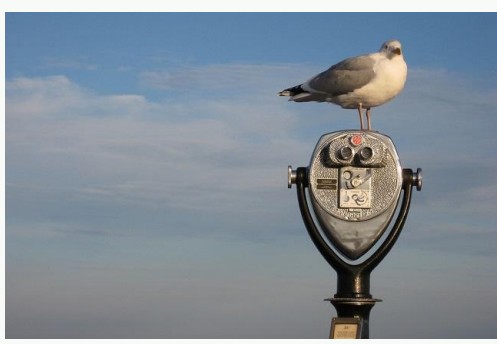

Does the pigeon stand in this image?

**Augmented Input Images**

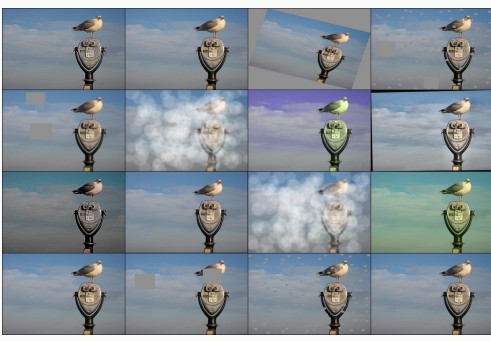

**Augmented Prompts**

Prompt 0: <|im_start|>User:<image>Do es the p igeon sta nd in this image? In other words, Does the pigeon stand in this image?<end_of_utterance> Assistant:

Prompt 1: <|im_start|>User:<image>D oes the pigeon sta nd in th is image? In other words, Does the pigeon stand in this image?<end_of_utterance> Assistant:

Prompt 2: <|im_start|>User:<image>D oes the pigeon s tand in th is image? In other words, Does the pigeon stand in this image?<end_of_utterance> Assistant:

Prompt 3: <|im_start|>User:<image>Does the pige on sta nd in th is image? In other words, Does the pigeon stand in this image?<end_of_utterance> Assistant:

Prompt 4: <|im_start|>User:<image>Does the p igeon sta nd in th is image? In other words, Does the pigeon stand in this image?<end_of_utterance> Assistant:

Prompt 5: <|im_start|>User:<image>D oes the pige on stand in th is image? In other words, Does the pigeon stand in this image?<end_of_utterance> Assistant:

Prompt 6: <|im_start|>User:<image>Do es the pigeon s tand in t his image? In other words, Does the pigeon stand in this image?<end_of_utterance> Assistant:

Prompt 7: <|im_start|>User:<image>Does the pig eon st and in this ima ge? In other words, Does the pigeon stand in this image?<end_of_utterance> Assistant:

Prompt 8: <|im_start|>User:<image>Does the pig eon stand in t his im age? In other words, Does the pigeon stand in this image?<end_of_utterance> Assistant:

Prompt 9: <|im_start|>User:<image>Do es the pigeon st and in th is image? In other words, Does the pigeon stand in this image?<end_of_utterance> Assistant:

Prompt 10: <|im_start|>User:<image>D oes the p igeon st and in this image? In other words, Does the pigeon stand in this image?<end_of_utterance> Assistant:

Prompt 11: <|im_start|>User:<image>Does the pigeon s tand in t his i mage? In other words, Does the pigeon stand in this image?<end_of_utterance> Assistant:

Prompt 12: <|im_start|>User:<image>Do es the pig eon stand in t his image? In other words, Does the pigeon stand in this image?<end_of_utterance> Assistant:

Prompt 13: <|im_start|>User:<image>Do es the pige on stand in th is image? In other words, Does the pigeon stand in this image?<end_of_utterance> Assistant:

Prompt 14: <|im_start|>User:<image>Does the pige on st and in t his image? In other words, Does the pigeon stand in this image?<end_of_utterance> Assistant:

Prompt 15: <|im_start|>User:<image>Does the pigeon stand in this image?<end_of_utterance> Assistant:

**Baseline Output**
**Answer:** No, the pigeon is perched on top of the coin return machine.
**Accuracy:** 0.0%

**TTAug Output**
**Answer:** Yes
**Accuracy:** 100.0%

**TTAdapt Output**
**Answer:** Yes
**Accuracy:** 100.0%

## COCO Captions

**Original Inputs**

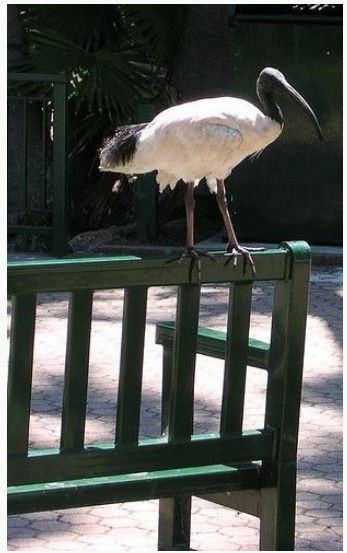

Please describe this image in general. Directly provide the description, do not include prefix like "This image depicts".

**Augmented Input Images**

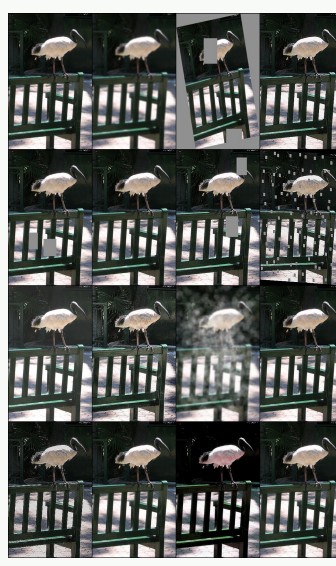

**Augmented Prompts**

Prompt 0: <|im_start|>User:<image>Please describe th is ima ge in ge Beral. Di rectly pgovide the description, do not include pre fix li<e " Th is i mage depicts ". In other words, Please describe this image in general. Directly provide the description, do not include prefix like "This image depicts".<end_of_utterance>
Assistant:

Prompt 1: <|im_start|>User:<image>Plea se describe this ima ge in ge nerap. Directly p rPvide the fescriprion, do not include prefix li ke " This i mage depi cts ". In other words, [... truncated, same as Prompt 0 ...]

Prompt 2: <|im_start|>User:<image>Ple ase desFribe th is ima ge in general. Directly proDide the des cription, do not include prefix l ike " T his i mxge depicts ". In other words, [... truncated, same as Prompt 0 ...]

Prompt 3: <|im_start|>User:<image>llease describe th is image in ge neral. Directly p rovide the des cription, do not include predix li ke " Th is im age dep&cts ". In other words, [... truncated, same as Prompt 0 ...]

Prompt 4: <|im_start|>User:<image>P lease des cribe t his image in general. Di rectly provide the descrip4Uon, do not i ncludF prsfix like " Th is image d epicts ". In other words, [... truncated, same as Prompt 0 ...]

Prompt 5: <|im_start|>User:<image>Pleaae de scribe t his ima ge in generzl. DirectlG provi de the description, do not include pr efix l ike " Th is image depicts ". In other words, [... truncated, same as Prompt 0 ...]

Prompt 6: <|im_start|>User:<image>Pleade descr ibe this ima ge in gene ral. Di$ec tly provide the description, do not inc lude pref8x like " Th is image dep icts ". In other words, [... truncated, same as Prompt 0 ...]

Prompt 7: <|im_start|>User:<image>Pl ease describe this image in ge neral. Sirect ly provide the Ceqcription, do not include p$ef ix like " T his i mage depi cts ". In other words, [... truncated, same as Prompt 0 ...]

Prompt 8: <|im_start|>User:<image>P lease describe this i mage in gen eral. Directl5 peovide the descr iption, do not i nclude prefix like " ThiE im age depic ts ". In other words, [... truncated, same as Prompt 0 ...]

Prompt 9: <|im_start|>User:<image>Plewse des cribe t his Omage in general. Dire ctly provi de the des cription, do not include pr efix like " This ima ge depicrs ". In other words, [... truncated, same as Prompt 0 ...]

Prompt 10: <|im_start|>User:<image>Plesse de scrube th is image in gene ral. Di rectly pro vide the description, do not inc lude prefix like " This im age Sepicts ". In other words, [... truncated, same as Prompt 0 ...]

Prompt 11: <|im_start|>User:<image>Pl ease descriGe this image in ge neral. Directly p%ovide the description, do not incl ude pr efix li ke " Th9s ima ge dep icts ". In other words, [... truncated, same as Prompt 0 ...]

Prompt 12: <|im_start|>User:<image>Olea se describe th is image in g eneral. Rirectly provide the WeCcription, do not inclu de pre fix l ike " T his image depicts ". In other words, [... truncated, same as Prompt 0 ...]

Prompt 13: <|im_start|>User:<image>Please describe t his im age in general. Dir ect.y provi de the descr((tion, do not inclu de prefix l ike " Th is image depictW ". In other words, [... truncated, same as Prompt 0 ...]

Prompt 14: <|im_start|>User:<image>PleasD describe t his image in gen etal. Directly provi de the descri Ltion, do not in clude prefix like " Th is ima ge depicts ". In other words, [... truncated, same as Prompt 0 ...]

Prompt 15: <|im_start|>User:<image>Please describe this image in general. Directly provide the description, do not include prefix like "This image depicts".<end_of_utterance>
Assistant:

**Baseline Output**

**Answer:** The image features a white bird with a long, curved beak and black legs standing on a green bench. The bird appears to be a type of ibis, characterized by its long, slender neck and beak. The bench is made of wood and has a slatted design, with the bird perched on the top slat. The background includes a paved area with a few scattered leaves and some greenery, suggesting an outdoor setting, possibly a park or a zoo. The bird is facing to the right, and its body is oriented towards the left side of the image.

**Accuracy (%):**
ROUGE-L: 15.59, BLEU-1: 12.24, BLEU-2: 7.94, BLEU-3: 5.08, BLEU-4: 3.43, CIDEr: 0.00

**TTAug Output**

**Answer:** A white bird with a long curved beak is standing on a green bench. The bird is looking to the left.

**Accuracy (%):**
ROUGE-L: 45.02,
BLEU-1: 52.38,
BLEU-2: 36.19,
BLEU-3: 27.45,
BLEU-4: 21.89,
CIDEr: 12.62

**TTAdapt Output**

**Answer:** A white bird with a long beak and black legs is standing on a green bench.

**Accuracy (%):**
ROUGE-L: 53.20,
BLEU-1: 62.50,
BLEU-2: 40.82,
BLEU-3: 32.93,
BLEU-4: 27.23,
CIDEr: 60.98

