# OpenReview forum: "Efficient Test-Time Scaling for Small Vision-Language Models"
_ICLR.cc/2026/Conference — ICLR 2026 Poster_

### Official Review · Reviewer_9zi6 · 2025-10-15

**Soundness:** 3
**Presentation:** 3
**Contribution:** 3
**Rating:** 6
**Confidence:** 3

**Summary:**

This paper proposes two efficient test-time scaling strategies, Test-Time Augmentation and Test-Time Adaptation, for improving the generalization performance of existing methods. Test-Time Augmentation aggregates the outputs of multiple augmented inputs at the token level without parameter updates. Test-Time Adaptation further improves the performance of Test-Time Adaptation by using it’s outputs as pseudolabels and updating model parameters. The effectiveness of the two methods is verified by extensive experiments.

**Strengths:**

* This paper proposes a new approach for aggregating multiple outputs at the token level.

* The experimental results are sufficient.

**Weaknesses:**

* The explanations on some of experimental results seems insufficient. For instances, in Table 1-5, the value of performance evaluation is zero on some datasets. It is better to explain such results.

* There is a lack of comparison on the computational cost across different methods. Test-Time Adaptation must update model parameters and thus requires more computational resources than Test-Time Adaptation and similar methods. The authors can provide some theoretical analyses or empirical results.

**Questions:**

* In Table2, if we exclude the dataset AI2D, it seems that answer-level aggregation with Self-Consistency is not worse than token-level aggregation with simple averaging. Besides, for the same dataset, answer-level aggregation with Sample-and-Rank is also better than token-level aggregation with simple averaging. We can observe the same results on the GQA datasets and answer-level aggregation with Self-Synthesizer. Therefore, it seems that token-level aggregation does not consistently outperforms answer-level aggregation. Could the authors explain those experimental results?

* How can the proposed methods implement early-stopping for low-quality generations?

---

> ### Author Response · Authors · 2025-11-20
>
> Thank you for your valuable feedback.
>
>
>
> ## Weakness 1: Insufficient Explanation of Zero Benchmarks
> > "The explanations on some of experimental results seems insufficient. For instances, in Table 1-5, the value of performance evaluation is zero on some datasets. It is better to explain such results."
>
>
>
> We believe there may be a misunderstanding regarding these zero performance values. **The zero scores represent baseline performance limitations** where the SmolVLM model completely fails on certain challenging tasks, as demonstrated by qualitative examples in our last appendix. These zeros also appear for other test-time scaling methods and suboptimal hyperparameter configurations across our ablation studies and comparisons. Importantly, with the optimized components, our methods demonstrate consistent improvements even on these challenging benchmarks where baselines fail entirely. As shown in Table 6, our TTAug method successfully recovers performance from complete failure, improving OCR-VQA accuracy from 0.0% to 12.6% and GQA accuracy from 0.0% to 5.5%, while TTAdapt achieves even larger gains (OCR-VQA: 0.0% → 13.8%, GQA: 0.0% → 13.5%). These results highlight our method's effectiveness in scenarios where traditional test-time scaling approaches provide no benefit.
>
>
>
>
>
> ## Weakness 2: Efficiency Comparison with Other Methods
> > "There is a lack of comparison on the computational cost across different methods. Test-Time Adaptation must update model parameters and thus requires more computational resources than Test-Time Adaptation and similar methods. The authors can provide some theoretical analyses or empirical results."
>
>
> We appreciate this important question regarding computational efficiency comparisons. While our previous analysis focused on TTAug overhead in "Appendix E - Computational Overhead of TTAug", **we have now added comprehensive computational cost results for other test-time scaling methods (in Section 4.1)**.
>
> |   |  Baseline | Self-Consistency | Self-Selector | Sample-and-Rank | Self-Synthesizer | TTAug (Ours) |
> |---|---|---|---|---|---|---|
> | Runtime (s) | 1.43 | 3.73 |  4.18  | 3.74 | 4.46 | 2.99 |
> | # Tokens | 8.76 | 74.47 | 77.39 | 74.47 | 82.28 | 70.31 |
>
>
> Our efficiency claims are relative to existing test-time scaling approaches: answer-level methods require full generation of candidate responses without early stopping mechanisms, while Self-Selector and Self-Synthesizer methods necessitate dual model passes for synthesis or selection from candidate pools, making them computationally even more expensive.
>
>
> Regarding TTAdapt efficiency,  TTAdapt is explicitly designed as a "when compute permits" option. Our work explores the full Pareto frontier of the computational efficiency-accuracy trade-off, allowing practitioners to choose the most suitable configuration for their platform constraints. Despite requiring parameter updates, TTAdapt remains efficient through several design choices: we use LoRA adapters to minimize optimizer state memory costs and employ efficient implementations with gradient checkpointing. We also provide an alternative aggregation weights optimization method for test-time adaptation, which improves the baseline accuracy across all benchmarks. The computational overhead is justified by substantial accuracy gains on challenging benchmarks. Note that we provide comprehensive implementation details in Appendices I.6 and I.7 to enable efficient deployment. This exemplifies the fundamental principle of test-time compute methods: strategically investing additional computational resources to achieve more accurate responses.

---

> ### Author Response · Authors · 2025-11-20
>
> ## Question 1: Superiority of Our TTAug Method
> > "In Table2, if we exclude the dataset AI2D, it seems that answer-level aggregation with Self-Consistency is not worse than token-level aggregation with simple averaging. Besides, for the same dataset, answer-level aggregation with Sample-and-Rank is also better than token-level aggregation with simple averaging. We can observe the same results on the GQA datasets and answer-level aggregation with Self-Synthesizer. Therefore, it seems that token-level aggregation does not consistently outperforms answer-level aggregation. Could the authors explain those experimental results?"
>
> We appreciate this important clarification question. Table 2 represents a controlled ablation study rather than a direct comparison between methods in their original forms. To isolate the effects of our novel diversity inducement mechanism from our token-level aggregation approach, we intentionally provided existing methods with access to our superior candidate generation strategy. However, this is a contribution they do not possess in their original implementations. This experimental design allows us to demonstrate two key insights: Table 1 establishes the superiority of our diversity inducement method, while Table 2 validates our aggregation approach even when competitors benefit from our improved diverse candidate generation mechanism.
>
> To address your concern about our method's superiority, **we have now included a comprehensive comparison table that explicitly demonstrates TTAug's advantages against existing test-time scaling methods in their original forms**. This new table clearly shows our method's consistent improvements across all benchmarks, making our contributions more transparent and accessible to readers. As one can see, our TTAug method outperforms all existing test-time scaling methods, confirming its overall superiority.
>
>
>
> |   |  Baseline | Self-Consistency | Self-Selector | Sample-and-Rank | Self-Synthesizer | TTAug (Ours) |
> |---|---|---|---|---|---|---|
> | ChartQA | 74.2 | 74.4 | 73.4 | 72.5 | 71.7 | 75.6 |
> | OCRBench | 72.9 | 72.6 | 71.9 | 70.2 | 71.9 | 73.4 |
> | OCRVQA | 0.0 | 0.0 | 0.0 | 0.0 | 0.2 | 11.8 |
> | GQA | 0.0 | 0.0 | 0.0 | 0.0 | 0.0 | 5.8 |
> | TextVQA | 73.2 | 72.6 | 71.6 | 69.5 | 72.0 | 72.8 |
> | AI2D | 68.5 | 3.1 | 69.2 | 69.1 | 67.4 | 68.8 |
> | MME-RW | 27.8 | 26.2 | 26.4 | 27.6 | 27.6 | 31.1 |
> | AMBER | 68.7 | 70.4 | 64.5 | 53.5 | 67.8 | 75.4 |
> | COCO | 9.1 | 8.2 | 8.4 | 6.2 | 16.7 | 15.9 |
> | **Mean** | 43.8 | 36.4 | 42.8 | 41.0 | 43.9 | **47.9** |
>
>
>
>
>
> ## Question 2: Early-Stopping for Low-Quality Generations
> > "How can the proposed methods implement early-stopping for low-quality generations?"
>
>
> **Our TTAug method incorporates an implicit pruning mechanism that naturally prevents low-quality generation trajectories.** By leveraging collective wisdom from all augmented inputs at each token position, we inherently avoid generating low-quality sequences without requiring explicit early-stopping heuristics.
>
> Traditional answer-level test-time scaling methods implement early-stopping through explicit heuristics, such as terminating candidate generation when the likelihood drops below a threshold. However, these approaches rely on hand-crafted stopping criteria. In contrast, our method provides a more principled solution where early stopping emerges naturally through agreement-based confidence at the token level. This is fundamentally impossible with answer-level aggregation because: (i) it must generate complete sequences before aggregation can occur, (ii) it prevents early stopping. Token-level aggregation permits per-token confidence monitoring, enabling early stopping similar to stepwise verification. This limitation was articulated in our paper:
>
> > "Nevertheless, all of these answer-level aggregation methods have critical limitations. First, global measures like confidence obscure confidence fluctuations at local reasoning steps, which can provide valuable signals for estimating response quality. Averaging across entire sequences masks critical reasoning breakdowns that occur at intermediate steps. Additionally, global measures require generating complete responses before calculation, preventing early stopping of low-quality generations and resulting in computational inefficiency."
>
> Note that we have now included a comprehensive comparison table showing our method's advantages against existing test-time scaling methods, including token-efficiency and runtime benefits beyond accuracy improvements. Please refer to the revised manuscript for complete details.
>
>
> ---
>
>
> We hope to have addressed your questions and concerns. We would greatly appreciate it if you could reconsider the score based on our response and results.

---

> > ### Comment · Reviewer_9zi6 · 2025-11-23
> > **I thank the authors for detailed rebuttal that have addressed all of my concerns.**
> >
> > I will determine whether to maintain or raise the score after the discussions with other reviewers.

---

### Official Review · Reviewer_6M4C · 2025-10-31

**Soundness:** 3
**Presentation:** 3
**Contribution:** 3
**Rating:** 6
**Confidence:** 5

**Summary:**

This paper addresses the performance limitations of small Vision-Language Models (VLMs) under domain shift by proposing two efficient test-time scaling strategies: Test-Time Augmentation (TTAug) and Test-Time Adaptation (TTAdapt). TTAug generates multiple semantically equivalent inputs through augmentations and aggregates predictions at the token level without parameter updates. TTAdapt extends this by fine-tuning model parameters during inference using pseudolabels derived from TTAug consensus. The methods are evaluated across nine diverse benchmarks, demonstrating consistent improvements while maintaining computational efficiency suitable for resource-constrained environments. The approach generalizes across model architectures and scales without additional tuning.

**Strengths:**

+ The paper introduces a novel framework for test-time scaling in small VLMs, different from existing methods that rely on external models or answer-level aggregation. The token-level aggregation strategy is novel, addressing limitations of global confidence measures by leveraging fine-grained signals during generation.
+ The work is technically sound, with extensive experiments covering nine benchmarks and multiple model families. Ablation studies evaluate different design choices. Theoretical analysis in the appendix justifies some key decisions.
+ The paper is well-structured, with good motivations, method descriptions, and visual demonstrations.
+ The work addresses the gap in deploying small VLMs under resource constraints. The proposed method can achieve practical improvements without heavy computational costs, making them potentially accessible for real-world applications.

**Weaknesses:**

- Limited comparison to baselines:​​ While the paper compares against methods like self-consistency and self-selector, it does not include more test-time adaptation techniques for VLMs. The comparison with these approaches would strengthen the claim of superiority.
- Computational Efficiency Claims:​​ The analysis in Section 4.3 and Appendix E reports memory and runtime overhead but lacks a breakdown of latency across different hardware. More explanation on this concerns are necessary.
- The benchmarks focus on question-answering and captioning, but those tasks requiring complex reasoning are not tested. The reviewer cannot confirm if this method will bring benefits for these tasks.

**Questions:**

- The results in Figure 4 show that optimal aggregation layers vary by task. Could the authors explain how these layer choices were determined for each benchmark? Is there a principled criterion behind the selection, or was it based on empirical validation?
- The paper claims the methods are suitable for resource-constrained environments but only provides results for an A100 GPU. Could the authors clarify if any tests were conducted on consumer-grade hardware?

---

> ### Author Response · Authors · 2025-11-20
>
> Thank you for your valuable feedback.
>
>
>
>
>
> ## Weakness 1: Limited Comparison to Baselines
> > "While the paper compares against methods like self-consistency and self-selector, it does not include more test-time adaptation techniques for VLMs. The comparison with these approaches would strengthen the claim of superiority."
>
>
>
> We appreciate this feedback and acknowledge the importance of comprehensive comparisons. **Our evaluation includes four representative test-time scaling methods** from the literature: Self-Consistency, Self-Selector, Sample-and-Rank, and Self-Synthesizer. Our resource-constrained problem setup imposes a critical constraint that no external models should be used, which limits the scope of comparable methods. For instance, we cannot evaluate PRM-based approaches that require external verifier models. The selected methods represent the core algorithmic families in test-time scaling literature without external dependencies, with most other approaches being variations of these fundamental ideas. If the reviewer is referring specifically to test-time adaptation methods for VLMs, we note that most existing TTAdapt work focuses on CLIP-based architectures, and we would welcome specific method suggestions that align with our resource-constrained setup for generative VLMs.
>
> Below is a conclusive summary table explicitly comparing our TTAug method against existing test-time scaling methods, showcasing our advantages in accuracy.
>
>
> |   |  Baseline | Self-Consistency | Self-Selector | Sample-and-Rank | Self-Synthesizer | TTAug (Ours) |
> |---|---|---|---|---|---|---|
> | ChartQA | 74.2 | 74.4 | 73.4 | 72.5 | 71.7 | 75.6 |
> | OCRBench | 72.9 | 72.6 | 71.9 | 70.2 | 71.9 | 73.4 |
> | OCRVQA | 0.0 | 0.0 | 0.0 | 0.0 | 0.2 | 11.8 |
> | GQA | 0.0 | 0.0 | 0.0 | 0.0 | 0.0 | 5.8 |
> | TextVQA | 73.2 | 72.6 | 71.6 | 69.5 | 72.0 | 72.8 |
> | AI2D | 68.5 | 3.1 | 69.2 | 69.1 | 67.4 | 68.8 |
> | MME-RW | 27.8 | 26.2 | 26.4 | 27.6 | 27.6 | 31.1 |
> | AMBER | 68.7 | 70.4 | 64.5 | 53.5 | 67.8 | 75.4 |
> | COCO | 9.1 | 8.2 | 8.4 | 6.2 | 16.7 | 15.9 |
> | **Mean** | 43.8 | 36.4 | 42.8 | 41.0 | 43.9 | **47.9** |
>
>
>
> ## Weakness 2: Different Hardware Latency Breakdown
> > "The analysis in Section 4.3 and Appendix E reports memory and runtime overhead but lacks a breakdown of latency across different hardware. More explanation on this concerns are necessary."
>
>
>
>
> ### Related Question 2: Real-World Hardware Experiments
> > "The paper claims the methods are suitable for resource-constrained environments but only provides results for an A100 GPU. Could the authors clarify if any tests were conducted on consumer-grade hardware?"
>
>
>
>
>
>
> We acknowledge that our computational overhead analysis was conducted exclusively on A100 GPUs. However, **latency analyses are highly transferable across different hardware platforms**, while peak GPU memory requirements remain platform-agnostic. As demonstrated in the following figure, the latency breakdown exhibits consistent patterns across different hardware configurations. We evaluated our method on both superior (H100) and consumer-grade (RTX 4070 Ti SUPER) GPUs. The relative latency breakdown for our parallel implementation remains remarkably similar across these diverse hardware platforms, confirming that the provided analysis is sufficient for practitioners' reference.
>
> Figure URL: https://ibb.co/wNjPznSh
>
> Cross-hardware latency analysis is not central to our paper's contribution, as our focus is on algorithmic efficiency rather than hardware optimization. For optimized performance, we provided some implementation details in "Appendix I.8 - Implementation Details for TTAug", such as monkey patching for effective KV caching. Further engineering optimizations such as TensorRT can improve latency, but these represent implementation details beyond our scope. Note that our sequential implementation can run on any platform capable of supporting the baseline small model, ensuring broad accessibility.

---

> ### Author Response · Authors · 2025-11-20
>
> ## Weakness 3: Complex Reasoning Benchmarks
> > "The benchmarks focus on question-answering and captioning, but those tasks requiring complex reasoning are not tested. The reviewer cannot confirm if this method will bring benefits for these tasks."
>
>
>
>
> We respectfully disagree with this characterization of our benchmark selection. Our experiments encompass 9 comprehensive benchmarks covering diverse task types: visual question answering (VQA) including ChartQA, OCRBench, OCRVQA, GQA, and TextVQA; multiple-choice questions (MCQ) with AI2D and MME-RealWorld; yes/no questions using AMBER; and image captioning with COCO Captions. Importantly, **many questions within these selected benchmarks require sophisticated complex reasoning capabilities**. For instance, ChartQA specifically evaluates complex reasoning over charts, which is the primary purpose of this benchmark as detailed in the original ChartQA paper. Similarly, other benchmarks in our evaluation suite contain complex mathematical problems and reasoning challenges. Therefore, our selected representative benchmarks provide diverse reasoning challenges that are entirely sufficient to evaluate the effectiveness of our method across various complex reasoning scenarios.
>
>
>
>
> ## Question 1: Aggregation Layer Choice
> > "The results in Figure 4 show that optimal aggregation layers vary by task. Could the authors explain how these layer choices were determined for each benchmark? Is there a principled criterion behind the selection, or was it based on empirical validation?"
>
>
> Except for the ablation study in "Appendix D - Aggregation in Early Layers", **we consistently use final logit aggregation across all experiments**, which provides the highest average performance across benchmarks. Our design decisions are deliberately based on cross-benchmark averages rather than benchmark-wise optimization to ensure generalizability and avoid overfitting to particular tasks. This approach maintains methodological consistency and supports our goal of developing broadly applicable test-time scaling methods rather than task-specific solutions. Therefore, we have a principled criterion behind our design choices, grounded in empirical validation across diverse benchmarks.
>
>
>
>
>
> ---
>
>
> We hope to have addressed your questions and concerns. We would greatly appreciate it if you could reconsider the score based on our response and results.

---

> ### Author Response · Authors · 2025-11-30
>
> We thank the reviewer for the constructive feedback and for considering raising the score.
>
> In "Appendix E - Computational Overhead of TTAug", we have now added additional discussions. For instance, we now include a paragraph (before the takeaway message) that discusses the transferability of the overhead analysis across different hardware configurations, addressing the reviewer’s concern about explaining the experimental results in more detail.
>
> We appreciate the reviewer’s detailed comments, which helped us improve the manuscript.

---

### Official Review · Reviewer_bAuU · 2025-10-31

**Soundness:** 2
**Presentation:** 2
**Contribution:** 2
**Rating:** 4
**Confidence:** 4

**Summary:**

This paper proposes two efficient test-time scaling methods for small vision-language models (VLMs): Test-Time Augmentation (TTAug), which aggregates token-level predictions from semantically preserved input augmentations without updating model parameters, and Test-Time Adaptation (TTAdapt), which further refines model parameters during inference using pseudolabels derived from TTAug consensus. Evaluated across nine benchmarks, both methods consistently improve performance while maintaining computational efficiency suitable for resource-constrained environments, outperforming existing answer-level test-time scaling approaches. The study also demonstrates that input perturbations with greedy decoding and token-level aggregation are more effective than temperature sampling and answer-level aggregation, respectively.

**Strengths:**

- Proposes token-level aggregation, a novel and empirically superior alternative to answer-level methods
- Demonstrates consistent gains across nine diverse benchmarks, including challenging open-ended tasks like captioning and VQA, where prior methods fail due to reliance on extractable final answers.
- Provides a comprehensive ablation study on augmentation strategies, aggregation layers, and modality-specific contributions, revealing task-dependent optima (e.g., early layer aggregation for visual reasoning).

**Weaknesses:**

- The paper's claims of efficiency are undermined by significant computational overhead: TTAug increases inference time by 3.3× and GPU memory usage by 1.9× at 16 augmentations, which contradicts the goal of resource-constrained deployment.
- TTAdapt requires iterative fine-tuning during inference and parameter resets per sample, introducing non-trivial latency and complexity that are not adequately justified by marginal gains on most benchmarks.
- The experimental evaluation lacks ablation on real-world hardware constraints, such as latency-sensitive edge devices or energy budgets, making the practical viability of the methods questionable.
- Text augmentation dominates performance gains, yet the simplest classical perturbations (e.g., keyboard typos) often match or outperform more sophisticated self-paraphrasing, casting doubt on the necessity of complex augmentation design.
- The model-agnostic claims are weakened by hyperparameters (e.g., 16 augmentations, classical high-strength augmentations) being optimized solely on SmolVLM2-2.2B and not validated for transfer across all model families.
- The generative image augmentation failure is attributed to text exclusion, but the paper ignores alternative approaches like synthetic text insertion or text-aware augmentations that could preserve OCR-relevant content.
- Qualitative results show that TTAug frequently fixes errors by “stitching” correct tokens from noisy augmented inputs, suggesting the model’s base competence is fragile and the method acts more as a heuristic ensemble than a principled improvement.

**Questions:**

Please see the weaknesses.

---

> ### Author Response · Authors · 2025-11-20
>
> Thank you for your valuable feedback.
>
>
>
> ## Weakness 1: Efficiency of TTAug
> > “The paper's claims of efficiency are undermined by significant computational overhead: TTAug increases inference time by 3.3× and GPU memory usage by 1.9× at 16 augmentations, which contradicts the goal of resource-constrained deployment.”
>
>
>
>
> We respectfully clarify that **computational overhead is inherent to all test-time scaling methods**, as they intentionally trade computational resources for improved performance. Our efficiency claims are relative to existing test-time scaling approaches, not the baseline model. As demonstrated in the comprehensive comparison table below, **TTAug achieves superior computational efficiency compared to alternative methods while using fewer tokens and requiring less runtime.**
>
> |   |  Baseline | Self-Consistency | Self-Selector | Sample-and-Rank | Self-Synthesizer | TTAug (Ours) |
> |---|---|---|---|---|---|---|
> | Runtime (s) | 1.43 | 3.73 |  4.18  | 3.74 | 4.46 | 2.99 |
> | # Tokens | 8.76 | 74.47 | 77.39 | 74.47 | 82.28 | 70.31 |
>
> Our method achieves this efficiency advantage through three key design choices: (1) token-level aggregation via greedy decoding instead of repeated sampling, (2) no external model requirements, and (3) batched evaluation in the parallel implementation. Answer-level methods require full generation of candidate responses without early stopping mechanisms, while Self-Selector and Self-Synthesizer methods necessitate dual model passes for synthesis or selection from candidate pools, making them computationally even more expensive.
>
>
> For practitioners concerned about memory constraints, we provide a sequential implementation (Appendix E) that can run on any platform supporting the baseline model. Additionally, our scaling curves (Figure 2, Appendix E) enable flexible budget allocation. This makes TTAug both computationally efficient and highly tunable for diverse deployment scenarios.
>
>
>
>
>
>
>
>
>
>
>
> ## Weakness 2: Efficiency of TTAdapt
> > “TTAdapt requires iterative fine-tuning during inference and parameter resets per sample, introducing non-trivial latency and complexity that are not adequately justified by marginal gains on most benchmarks.”
>
>
>
> **TTAdapt is explicitly designed as a "when compute permits" option.** Our work explores the full Pareto frontier of the computational efficiency-accuracy trade-off, allowing practitioners to choose the most suitable configuration for their platform constraints. Despite requiring parameter updates, TTAdapt remains efficient through several design choices: we use LoRA adapters to minimize optimizer state memory costs and employ efficient implementations with gradient checkpointing. We also provide an alternative aggregation weights optimization method for test-time adaptation, which improves the baseline accuracy across all benchmarks. The computational overhead is justified by substantial accuracy gains on challenging benchmarks. Note that we provide comprehensive implementation details in Appendices I.6 and I.7 to enable efficient deployment.
>
>
>
>
> ## Weakness 3: Real-World Hardware Experiments
> > “The experimental evaluation lacks ablation on real-world hardware constraints, such as latency-sensitive edge devices or energy budgets, making the practical viability of the methods questionable.”
>
>
> We acknowledge that our computational overhead analysis was conducted exclusively on A100 GPUs. However, **latency analyses are highly transferable across different hardware platforms**, while peak GPU memory requirements remain platform-agnostic. As demonstrated in the following figure, the latency breakdown exhibits consistent patterns across different hardware configurations. We evaluated our method on both superior (H100) and consumer-grade (RTX 4070 Ti SUPER) GPUs. The relative latency breakdown for our parallel implementation remains remarkably similar across these diverse hardware platforms, confirming that the provided analysis is sufficient for practitioners' reference.
>
>
> Figure URL: https://ibb.co/wNjPznSh
>
>
> Cross-hardware latency analysis is not central to our paper's contribution, as our focus is on algorithmic efficiency rather than hardware optimization. For optimized performance, we provided some implementation details in "Appendix I.8 - Implementation Details for TTAug", such as monkey patching for effective KV caching. Further engineering optimizations, such as TensorRT, can improve latency, but these represent implementation details beyond our scope. Note that our sequential implementation can run on any platform capable of supporting the baseline small model, ensuring broad accessibility. This includes edge devices with limited resources.
>
> If the reviewer has specific practical hardware scenarios in mind, we would be happy to discuss how our method can be adapted accordingly. However, we believe that the current analysis sufficiently demonstrates the practical viability of our methods across a range of hardware configurations.

---

> ### Author Response · Authors · 2025-11-20
>
> ## Weakness 4: Text Augmentation
> > “Text augmentation dominates performance gains, yet the simplest classical perturbations (e.g., keyboard typos) often match or outperform more sophisticated self-paraphrasing, casting doubt on the necessity of complex augmentation design.”
>
> **The effectiveness of simple classical perturbations (e.g., keyboard typos) actually strengthens the practical appeal of our method rather than undermining it.** Our results explicitly demonstrate that lightweight augmentations achieve nearly the same performance as sophisticated self-paraphrasing approaches, which directly supports our efficiency-focused design goals. This finding validates that TTAug can deliver substantial improvements without requiring computationally expensive generative augmentation strategies, making it highly suitable for resource-constrained deployment scenarios. The ability to achieve strong performance with simple, fast classical perturbations is precisely what makes our method practical and accessible. In short, this is a feature that enhances deployment viability, not a limitation of our approach.
>
>
> ## Weakness 5: Cross-Model Generalization
> > “The model-agnostic claims are weakened by hyperparameters (e.g., 16 augmentations, classical high-strength augmentations) being optimized solely on SmolVLM2-2.2B and not validated for transfer across all model families.”
>
>
> We respectfully disagree with this characterization of our generalization claims. "Section 4.7 - Cross-Model Generalization" demonstrates that **our method achieves consistent gains** across all tested model families (SmolVLM2, Ovis2, and InternVL2), **even when using hyperparameters optimized exclusively on SmolVLM2-2.2B**. This cross-model transferability with suboptimal hyperparameters actually strengthens our generality claim rather than weakening it. While we acknowledge that per-model hyperparameter tuning would yield optimal performance, our paper explicitly provides the optimization procedure that practitioners can follow for any given model. The fact that our method delivers reliable improvements across diverse architectures using a single hyperparameter configuration demonstrates its robustness and practical applicability.
>
> ## Weakness 6: Text-Aware Generative Image Augmentation
> > “The generative image augmentation failure is attributed to text exclusion, but the paper ignores alternative approaches like synthetic text insertion or text-aware augmentations that could preserve OCR-relevant content.”
>
> We acknowledge that text-aware generative augmentations could potentially preserve OCR-relevant content better than the tested approach. However, **our generative image augmentation experiment was conducted solely for ablation purposes** to test whether the success of generative methods in text augmentation would transfer to the image domain. Our results showed this is not the case. Since we cannot generate images with the tested VLM itself, we used an external image generation model, but this contradicts our fundamental design principle of avoiding external models for resource-constrained deployments. As we explicitly note in our paper, such generative image augmentation approaches "require external diffusion models, making them impractical for resource-constrained deployments." Thus, due to our resource-constrained problem setup, we did not explore this generative image augmentation approach further. Our experiment was not intended as a definitive evaluation of all possible generative image strategies, but rather as a targeted ablation to understand the generalizability of generative augmentation benefits across modalities within our computational constraints.
>
> ## Weakness 7: Heuristic Ensemble
> > “Qualitative results show that TTAug frequently fixes errors by “stitching” correct tokens from noisy augmented inputs, suggesting the model’s base competence is fragile and the method acts more as a heuristic ensemble than a principled improvement.”
>
> **Our TTAug method is theoretically grounded, not heuristic.** Token-level aggregation is more principled than answer-level ensembling, and our paper provides both empirical and theoretical justification. Token-level averaging uses model-internal signals, following established principles of confidence calibration and ensemble smoothing. We show mathematically (Appendix B) that token-level averaging approximates a consensus distribution over perturbation-induced neighborhoods of the input, which is exactly what test-time scaling aims for. "Stitching" is an expected and desirable behavior: it reflects local agreement across perturbations and enables recovery from mid-sequence failure modes that answer-level methods cannot fix. Thus, the method is not a heuristic ensemble, but a principled application of selective ensembling at the token level.
>
> ---
>
> We hope to have addressed your questions and concerns. We would greatly appreciate it if you could reconsider the score based on our response and results.

---

> ### Author Response · Authors · 2025-11-30
>
> Dear Reviewer,
>
> With the discussion concluding soon, we kindly ask if our responses have addressed your concerns, and let us know if any further questions or comments you might have. Thanks!

---

### Official Review · Reviewer_HfFB · 2025-10-31

**Soundness:** 3
**Presentation:** 3
**Contribution:** 3
**Rating:** 4
**Confidence:** 3

**Summary:**

The article studies how to perform test-time scaling for small vision-language models. In particular, it investigates different design choices in terms of test-time augmentation (TTAug) and test-time adaptation (TTAdapt), analyzing how performance changes in 9 benchmarks w.r.t. diversity induced methods, aggregation levels, number of augmentations, text/image augmentation methods, and adaptation strategies. Each tested design choice comes with recipes and findings that, when put together, lead to consistent improvement across various VLMs.

**Strengths:**

1. This is an extensive study on how a large variety of design choices impacts the performance of test-time scaling. The study is well conducted, exploring a large variety of alternatives with regard to both test-time augmentation and adaptation. The results can serve as a blueprint for developers of test-time scaling techniques and a reference for practitioners and researchers in this field.

2. The choices have been primarily analyzed for SmolVLM2-2.2B, a small-scale model, but they also partly generalize to larger ones of different families, as shown with the experiments of Fig. 3.

3. The appendix is remarkable, providing extensive details on the implementation as well as theoretical analyses. This level of detail is important not only to ground the design choices (with the theoretical part) but also to allow for easy reproducibility.

**Weaknesses:**

1. The gaps across alternative strategies are sometimes small and might be hard to draw general conclusions from. For instance, 5 methods in Tab. 4 have only a 0.4 gap in performance, with the caption claiming that classical augmentations (L-M-H)  are the best, while the text-only strategy (0) performs comparably or even better. I am aware that experiments are costly, but reporting error bars or analyses on the significance of the gap could help strengthen the main takeaways.

2. Related to the general nature of the results, after each analysis, the best approach is selected for subsequent experiments/as a reference. Given the small gaps, optimal choices for one architecture might not be transferable to other ones. This behaviour can be seen from Fig. 3, as i) only the SmolVLM2 family shows consistent improvements (e.g., compared to Ovis2 and InternVL2), but only for SmolVLM2 2B, those are large. Moreover, even within that family, test-time augmentation and test-time scaling may show different behaviours (e.g., SmolVLM2 0.3B has TTAug helping more than TTAdapt, while SmolVLM2 0.5B shows the opposite trend). Performing the same analyses but considering more architectures/families would strengthen the applicability of the core takeaways.

3. The choice of benchmarks from VLMEvalKit (Duan et al. 2024) needs some clarification. There are 20 benchmarks in the VLMEvalKit (Duan et al. 2024), but 9 are considered: why is this subset considered the most representative? Note also that the AI2D benchmark on diagram understanding polarizes some of the results: e.g., self-consistency in Tab. 1 outperforms or is comparable to self-selection alternatives everywhere but for AI2D, where there is a huge gap (>60% points), and the same happens in Tab. 2 (1 vs 2 and 4). As the choice of benchmarks may have a high impact on the results and takeaways, the choice of benchmarks may need some clarification, as well as the use of the average to assess the best-performing method.

4. (relatively minor) From the tables, various alternatives are presented, but an overall comparison with the state-of-the-art is unclear (yet performed when testing various alternatives, e.g., Tab. 1 (Chen et al. 2024a; Pruner et al. 2025, Wang et al. 2023b). Given that one of the aims of the article is to outperform existing test-time scaling approaches (e.g., lines 81-85), it would be helpful to report a conclusive summary table where the final strategy is explicitly compared with existing works.

**Minor points:**

5. While the article focuses on small models, to my knowledge, the tested strategies are not tailored to small models in particular. While I understand that test-time scaling is more helpful in the context of lower-performing VLMs, showing how the final findings generalize to large models could give further insights into the main outcomes. I deem this as a minor point, as the analysis is still helpful and several models have already been tested.

6. Section 3.1 provides a clear formalization of TTAug. However, 3.2 describes TTAdapt without any formalization. While excessive notations or unnecessary formulas may hinder the clarity, providing a general definition of TTAdapt with a mathematical formalization would be helpful to avoid potential confusion in the reader and make the manuscript (3.1 and 3.2) more consistent.

**Questions:**

Related (point-wise) to the weaknesses above:

1. Given the small gaps, how general might the result hold for other architectures?
2. Are there insights on why TTAdapt and TTAug could show inconsistent behaviours?
3. Could you elaborate on the choice of benchmarks and how to select the best strategy (given the variance of results across benchmarks)?
4. How does the final model compare with existing alternatives?

---

> ### Author Response · Authors · 2025-11-20
>
> Thank you for your constructive feedback.
>
>
>
> ## Weakness 1: Statistical Significance
> > “The gaps across alternative strategies are sometimes small and might be hard to draw general conclusions from. For instance, 5 methods in Tab. 4 have only a 0.4 gap in performance, with the caption claiming that classical augmentations (L-M-H) are the best, while the text-only strategy (0) performs comparably or even better. I am aware that experiments are costly, but reporting error bars or analyses on the significance of the gap could help strengthen the main takeaways.”
>
>
>
> **We have now added standard error values in Appendix K, which demonstrate that even small performance gaps are statistically meaningful.**
>
> We calculated the standard errors using the empirical variance of the observed per-sample scores $s_1, \ldots, s_n$. Let $\bar{s} = \frac{1}{n}\sum_i s_i$ denote the sample mean. This average accuracy is the value reported in the main text tables. Following the Central Limit Theorem, the corresponding standard error is estimated as
> $$
> \mathrm{SE}_{\text{C.L.T.}} = \sqrt{\mathrm{Var}(s)/n} = \sqrt{\left( \frac{1}{n-1}\sum_i (s_i - \bar{s})^2 \right) / n } .
> $$
>
> Our analysis shows an average standard error of approximately 0.4% for mean cross-benchmark accuracy, confirming that the observed 0.4% gap in Table 4 is indeed statistically significant. The relatively small improvements from image augmentation align with established findings in the VLM literature: as discussed in "Appendix F - Multimodal Augmentation Decomposition," VLMs tend to overlook visual input and rely heavily on text information, making image augmentation less effective. This behavior is well-documented and expected in current VLM architectures, but our statistical analysis confirms that even these modest gains are reliable and significant.
>
>
> ### Related Question 1
> > "Given the small gaps, how general might the result hold for other architectures?"
>
>
>
> Our core results demonstrate excellent generalizability across architectures. Both TTAug and TTAdapt show statistically significant improvements over baseline and existing test-time scaling methods. This is the main result and our primary contribution. The substantial performance gaps between our methods and existing approaches provide strong evidence for reliable generalization.
>
> While some hyperparameter ablation experiments show smaller gaps, these remain statistically significant given our standard error analysis in Appendix K. Importantly, our fundamental insights, input perturbations outperform temperature sampling and token-level aggregation surpasses answer-level approaches, hold consistently across diverse model families with both empirical and theoretical justification.
>
> As discussed in "Section 4.7 - Cross-Model Generalization," we explicitly acknowledge that optimal hyperparameters may vary across models. However, our core principles remain robust across architectures. We provide a systematic hyperparameter optimization procedure that practitioners can adapt for any given model. The fact that our methods deliver consistent improvements even with suboptimal hyperparameters demonstrates their practical robustness and broad applicability.

---

> ### Author Response · Authors · 2025-11-20
>
> ## Weakness 2: Hyperparameter Optimization for Other Models
> > “Related to the general nature of the results, after each analysis, the best approach is selected for subsequent experiments/as a reference. Given the small gaps, optimal choices for one architecture might not be transferable to other ones. This behaviour can be seen from Fig. 3, as i) only the SmolVLM2 family shows consistent improvements (e.g., compared to Ovis2 and InternVL2), but only for SmolVLM2 2B, those are large. Moreover, even within that family, test-time augmentation and test-time scaling may show different behaviours (e.g., SmolVLM2 0.3B has TTAug helping more than TTAdapt, while SmolVLM2 0.5B shows the opposite trend). Performing the same analyses but considering more architectures/families would strengthen the applicability of the core takeaways.”
>
>
>
> We acknowledge the reviewer's observation about hyperparameter transferability across architectures. However, our core claims explicitly recognize this limitation. We explicitly assert that the underlying principles generalize broadly while acknowledging that specific hyperparameters may require model-specific tuning.
>
> Importantly, our evidence for generalization is compelling: Figure 3 demonstrates that **every model** across all tested families (InternVL2, Ovis2, and SmolVLM2) obtains **positive gains** under TTAug, while TTAdapt consistently improves larger models without degrading smaller ones. This robust performance across diverse architectures, even with hyperparameters optimized for a different model family, actually strengthens our method's practical appeal by demonstrating near plug-and-play capability.
>
> Our approach is grounded in both empirical evidence and theoretical principles. TTAug prevents error propagation through token-level aggregation, which provides fundamental advantages regardless of architecture specifics. While conducting exhaustive hyperparameter optimization for every model family would be computationally prohibitive, the consistent improvements we observe with suboptimal hyperparameters validate our core contribution: that TTAug and TTAdapt reliably improve VLM performance across diverse architectures.
>
>
>
> ### Related Question 2
> > "Are there insights on why TTAdapt and TTAug could show inconsistent behaviours?"
>
>
>
>
> The behavioral differences between TTAdapt and TTAug across models can be attributed to their distinct confidence calibration properties. TTAdapt's parameter adaptation approach is sensitive to model-specific confidence patterns: models with well-calibrated token-level confidence enable TTAdapt to extract reliable pseudolabels for effective adaptation, while overconfident models may lead TTAdapt to overfit on noisy pseudolabels, making the more stable TTAug method preferable. Additionally, our analysis reveals that TTAdapt's effectiveness is task-dependent. Aggressive parameter adaptation can occasionally disrupt carefully calibrated domain-specific knowledge on specialized benchmarks where the base model already performs well. This suggests that adaptation intensity should be conservative for well-calibrated domains and more aggressive for challenging distributions where consensus-based supervision provides reliable guidance. TTAug, being parameter-free, avoids these calibration-dependent failure modes.

---

> ### Author Response · Authors · 2025-11-20
>
> ## Weakness 3: The Choice of Benchmarks
> > “The choice of benchmarks from VLMEvalKit (Duan et al. 2024) needs some clarification. There are 20 benchmarks in the VLMEvalKit (Duan et al. 2024), but 9 are considered: why is this subset considered the most representative? Note also that the AI2D benchmark on diagram understanding polarizes some of the results: e.g., self-consistency in Tab. 1 outperforms or is comparable to self-selection alternatives everywhere but for AI2D, where there is a huge gap (>60% points), and the same happens in Tab. 2 (1 vs 2 and 4). As the choice of benchmarks may have a high impact on the results and takeaways, the choice of benchmarks may need some clarification, as well as the use of the average to assess the best-performing method.”
> ### Related Question 3
> > "Could you elaborate on the choice of benchmarks and how to select the best strategy (given the variance of results across benchmarks)?"
>
> Thanks for asking this question. While we were very careful to select the most representative benchmarks, we did not explicitly mention the reasons behind our selection.
>
> First, **we deliberately avoided LLM-as-a-judge benchmarks due to their inherent unreliability**. Results can be biased by model preferences, prompt design, and answer style. There is also a strong dependency on the choice of judge model and rubric, and they lack the reproducibility of fixed benchmarks. They depend on the judge's capability and rubric choice (prompt design), adding an extra layer of complexity. Additionally, VLMEvalKit uses GPT-4 API calls that are expensive in terms of cost. In contrast, our selected benchmarks were VQA, Y/N, MCQ, and captioning tasks. Their evaluation metrics are highly standardized and reproducible across papers and model cards. Since we conduct extensive ablation studies, we prefer to use benchmarks that are relatively quick and cost-effective to run at scale. Also, **we excluded poorly designed benchmarks with known issues**, such as MMMU, since they are text-dominant and do not effectively evaluate VLM reasoning (Chen et al., 2024; "Are We on the Right Way for Evaluating Large Vision-Language Models?"). Moreover, **we excluded specialized benchmarks**: CCBench (Chinese Culture), AesBench (Aesthetics), LLaVABench (Subjective Evaluation), and POPE (Object Hallucination). We preferred to focus on more general reasoning benchmarks.
>
> Importantly, we did not cherry-pick these benchmarks to favor our method. In fact, the selected benchmarks likely downplay our contribution. As demonstrated in Appendix B, our method's advantage increases with the number of output tokens compared to other test-time scaling methods. Our benchmarks are predominantly MCQ, Y/N, or VQA tasks that often require short answers. The only benchmark capturing free-form writing ability was COCO Captions, and empirically, our method shows more pronounced benefits on that benchmark as expected.
>
> In summary, the selected benchmarks cover many distinct reasoning types: OCR, compositional reasoning, chart understanding, diagram MCQs, and captions. They are widely adopted in prior VLM evaluation papers, allow efficient ablations under resource constraints, and are computationally feasible for full test-time scaling sweeps.
>
> While we could provide additional results, it is not computationally feasible to repeat our extensive ablation experiments across all available benchmarks. This is a general challenge in test-time scaling research. We believe our benchmarks are comprehensive compared to other works in the LLM test-time scaling literature. Other papers typically report results on very small benchmarks, as test-time scaling methods require more runtime by nature. Below are evaluation set sample sizes for well-known papers from the test-time scaling literature:
> - "Scaling LLM Test-Time Compute Optimally can be More Effective than Scaling Model Parameters" → MATH500: 500 samples
> - "Can 1B LLM Surpass 405B LLM? Rethinking Compute-Optimal Test-Time Scaling" → MATH500: 500 samples, AIME24: 30 samples
> - "s1: Simple test-time scaling" → MATH500: 500 samples, AIME24: 30 samples, GPQA Diamond: 198 samples
> - "START: Self-taught Reasoner with Tools" → MATH500: 500 samples, AMC23: 40 samples, AIME24: 30 samples, AIME25: 30 samples, LiveCodeBench: 112 samples, GPQA: 448 samples
>
> In contrast, we use many representative benchmarks, which enhances the statistical significance of our results.
>
> Regarding the use of averaging for best strategy selection: while one can achieve higher performance with per-benchmark or per-subtask hyperparameter optimization in practice, rigorous hyperparameter optimization is not central to our paper. Averages are used to summarize trends, and averaging across benchmarks is standard practice. Given the task heterogeneity, averaging provides a fair evaluation method for general-purpose test-time scaling approaches. As we report benchmark-wise results, varied performance across benchmarks is transparent to readers.

---

> ### Author Response · Authors · 2025-11-20
>
> ## (Relatively Minor) Weakness 4: Comparison with Other Methods
> > “(relatively minor) From the tables, various alternatives are presented, but an overall comparison with the state-of-the-art is unclear (yet performed when testing various alternatives, e.g., Tab. 1 (Chen et al. 2024a; Pruner et al. 2025, Wang et al. 2023b). Given that one of the aims of the article is to outperform existing test-time scaling approaches (e.g., lines 81-85), it would be helpful to report a conclusive summary table where the final strategy is explicitly compared with existing works.”
>
> ### Related Question 4
> > "How does the final model compare with existing alternatives?"
>
>
>
>
> We appreciate this excellent suggestion for improving clarity and readability. While our method's superiority was demonstrated implicitly in Tables 1 and 2, we recognize that a direct comparison table would better serve readers. **We have now added a comprehensive summary table (Section 4.1) that explicitly compares our TTAug method against existing test-time scaling methods**, showcasing our advantages in accuracy, token efficiency, and computational speed. This consolidates the evidence that was previously distributed across multiple tables into a single, conclusive reference point.
>
>
> |   |  Baseline | Self-Consistency | Self-Selector | Sample-and-Rank | Self-Synthesizer | TTAug (Ours) |
> |---|---|---|---|---|---|---|
> | ChartQA | 74.2 | 74.4 | 73.4 | 72.5 | 71.7 | 75.6 |
> | OCRBench | 72.9 | 72.6 | 71.9 | 70.2 | 71.9 | 73.4 |
> | OCRVQA | 0.0 | 0.0 | 0.0 | 0.0 | 0.2 | 11.8 |
> | GQA | 0.0 | 0.0 | 0.0 | 0.0 | 0.0 | 5.8 |
> | TextVQA | 73.2 | 72.6 | 71.6 | 69.5 | 72.0 | 72.8 |
> | AI2D | 68.5 | 3.1 | 69.2 | 69.1 | 67.4 | 68.8 |
> | MME-RW | 27.8 | 26.2 | 26.4 | 27.6 | 27.6 | 31.1 |
> | AMBER | 68.7 | 70.4 | 64.5 | 53.5 | 67.8 | 75.4 |
> | COCO | 9.1 | 8.2 | 8.4 | 6.2 | 16.7 | 15.9 |
> | **Mean** | 43.8 | 36.4 | 42.8 | 41.0 | 43.9 | **47.9** |
>
>
> Furthermore, we have fundamentally restructured our experiments section to address this concern. Section 4.1 now opens with "Comparison with Other Test-Time Scaling Methods,". We then explain the underlying reasons for this superiority through two key insights: our diversity inducement method is better, and our token-level aggregation method surpasses answer-level approaches. The subsequent subsections (4.2 - Diversity-Inducement Methods and 4.3 - Aggregation Levels) provide controlled experiments that decouple and validate these components individually, creating a logical flow from overall comparison to detailed analysis. Please see the revised manuscript for more details.
>
>
>
>
>
>
>
>
>
>
> ## (Minor) Weakness 5: Experiments for Bigger Models
> > “(minor) While the article focuses on small models, to my knowledge, the tested strategies are not tailored to small models in particular. While I understand that test-time scaling is more helpful in the context of lower-performing VLMs, showing how the final findings generalize to large models could give further insights into the main outcomes. I deem this as a minor point, as the analysis is still helpful and several models have already been tested.”
>
>
>
> We appreciate this feedback and agree that our technique is not architecturally exclusive to small models. Our focus on small VLMs stems from a clear practical motivation: these models benefit most from compute-constrained test-time scaling where efficiency is not optional but essential. While we would value testing on larger models, **our computational constraints prevented such experiments**, and we deliberately avoided extrapolating beyond our empirical evidence to prevent overclaiming. Our core contribution is designing test-time scaling methods that are computationally efficient for resource-constrained deployments, compatible with small VLM paradigm, without requiring external verifiers that would defeat the efficiency purpose.
>
>
>
>
>
>
> ## (Minor) Weakness 6: Formalization for TTAdapt Method
> > “(minor) Section 3.1 provides a clear formalization of TTAug. However, 3.2 describes TTAdapt without any formalization. While excessive notations or unnecessary formulas may hinder the clarity, providing a general definition of TTAdapt with a mathematical formalization would be helpful to avoid potential confusion in the reader and make the manuscript (3.1 and 3.2) more consistent.”
>
>
> We appreciate this excellent suggestion to improve methodological consistency and readability. While we initially relied on Figure 1 to convey the TTAdapt method due to space constraints, **we have now included formal pseudocode in Algorithm 1** to provide the mathematical formalization that matches Section 3.1's rigor. This addition ensures both sections maintain consistent presentation standards and eliminates any potential confusion about the TTAdapt procedure.

---

> ### Author Response · Authors · 2025-11-20
>
> We would like to thank this reviewer again for the constructive feedback and great suggestions that helped us improve the clarity and presentation of our manuscript. We hope to have addressed your questions and concerns. We would greatly appreciate it if you could reconsider the score based on our response and results.

---

> > ### Comment · Reviewer_HfFB · 2025-11-25
> >
> > I thank the reviewers for their thorough feedback. Most of my concerns have been addressed, and I raised the score to 6 accordingly.
> >
> > It would be helpful to include the explanation on the benchmark selection in the manuscript (e.g., appendix). Similarly, I find interesting the intuition provided in answer to "related question 2". I know there are space constraints, but adding this explanation would complement  the quantitative results with a pointer for practitioners and researchers.

---

> > > ### Author Response · Authors · 2025-11-30
> > >
> > > We thank the reviewer for the constructive feedback and for increasing the score to 6.
> > >
> > > In "Appendix J - Evaluation Metrics Details", we have now added a discussion explaining the rationale behind our benchmark selection.
> > >
> > > In "Section 4.8 - Cross-Model Generalization", we have expanded the discussion of the experimental results to incorporate the intuition highlighted in your comments.
> > >
> > > We appreciate the reviewer’s thoughtful suggestions and believe these additions strengthen the clarity of the manuscript.

---

### Author Response · Authors · 2025-11-20
**Summary of Manuscript Revisions**

We have made the following key improvements in the updated manuscript to address reviewers' feedback:

- **Enhanced experimental presentation**: Added a comprehensive comparison table (Section 4.1) that directly compares our TTAug method against all baseline test-time scaling methods, consolidating previously distributed results into a single reference point for clarity.

- **Improved methodological presentation**: Included formal pseudocode for TTAdapt (Algorithm 1) to provide mathematical formalization that matches the rigor of Section 3.1, eliminating potential confusion about the procedure.

- **Statistical rigor**: Added standard error values in Appendix K to strengthen the statistical significance of our findings.

- **Expanded discussion of experimental findings**: Expanded the discussion in Section 4.8 and Appendix E to provide a more complete understanding of our experimental results.

- **Benchmark selection**: Added a discussion in Appendix J explaining the rationale behind our benchmark selection.

---

### Meta-Review · Area_Chair_Ab7r · 2025-12-24

**Summary:**

The reviews for this paper were mixed, with two Marginal Accept (6) and two Marginal Reject (4) scores.
The paper addresses the problem of test-time scaling for small vision-language models (VLMs), where standard approaches are often computationally expensive, incompatible with open-ended generation, or reliant on external models. To overcome these limitations, the authors introduce two methods: Test-Time Augmentation (TTAug), which performs token level aggregation over semantically preserved input perturbations without updating model parameters and Test-Time Adaptation (TTAdapt), which further adapts model parameters during inference using consensus-based pseudolabels derived from TTAug.
The paper makes a clear and novel contribution by showing that token level aggregation during autoregressive decoding is both more effective and more flexible than answer level aggregation, particularly for small VLMs and open ended tasks. Extensive experiments across nine diverse benchmarks, multiple augmentation strategies, aggregation levels, and model families demonstrate consistent gains over prior test-time scaling approaches. The work is technically sound, well-motivated, and supported by both empirical evidence and theoretical analysis.
Based on the reviews and the rebuttal, I recommend acceptance.

**Reviewer Concerns:**

The initial evaluations of this article were mixed, with two marginal acceptance scores (6) and two marginal rejection scores (4).

The reviewer initially raised the following concern:

1. Small Performance Differences and Insufficient Evidence: Reviewers indicate that many methods exhibit very similar performance, with very small score differences. Five methods in Table 4 show a performance difference of only 0.4. In some tables, the method presented as "best" is barely superior to simpler alternatives.
2. Hyperparameter Generalization for Other Models: Reviewers indicate that the results often depend on a specific model family (particularly SmolVLM2). Optimal choices for a given architecture are not necessarily transferable to others, implying a lack of transferability.
3. The Choice of Benchmarks: Reviewers indicate that only a subset of available benchmarks is used, without a clear explanation.

4. Comparison with Other SOTA Methods: Reviewers indicate that the article compares many variants internally but does not offer a clear final comparison with existing leading methods. It is therefore difficult to determine whether the final method represents a real improvement over existing approaches.

5. High computational cost and efficiency issues: The reviewers indicate that the increased testing time significantly increases inference time and memory consumption. So test-time adaptation requires extra optimization steps and parameter resets. Maybe it would be great to add a sentence about it.
6. Complex Reasoning Benchmarks: The reviewers indicate that the assessments focus on answering questions and generating legends, but tasks requiring complex reasoning are not tested.
7. Hyperparameter sensitivity and limited transferability: The reviewers indicate that key parameters (e.g., the number and strength of augmentations) are optimized on a single large model. Consequently, these parameters are not validated across all models and datasets.
8. Insufficient explanation of results and methods, particularly TTAdapt: The reviewers  indicate that some results are not explained, especially cases of zero performance. The lack of formalization and the use of overly complex notation make TTAdapt difficult to understand.

**Reviewer Scores:**

The initial reviewer ratings were mixed, with two reviews at Marginal Accept (6) and two at Marginal Reject (4). Following the rebuttal, the authors provided responses to the main concerns raised by the reviewers. Two reviewers acknowledged that their technical questions were satisfactorily answered, with no remaining critical objections. To incorporate reviewer feedback, the authors have made the following improvements to the updated manuscript: (1) Improved experimental presentation; (2) Improved methodological presentation; and (3) Added baselines. All these changes have strengthened the paper. Regarding the AI-powered pseudo-review, I'll leave that to the PCs. Overall, after considering the rebuttal and revisions, I believe the paper is acceptable in its current form, as the authors have adequately addressed the key issues and strengthened the manuscript in response to reviewer feedback.

---

### Decision · Program_Chairs · 2026-01-26

Accept (Poster)